# Kanade: Disentangled Linguistically Rich Tokens for Speech Modeling

## Abstract

A good language model starts with a good tokenizer. Tokenization is especially important for speech modeling, which must handle noisy continuous speech recordings. A speech tokenizer should produce compact, linguistically rich representations while still enabling high-quality synthesis. We present Kanade, a tokenizer that realizes this ideal. Kanade separates out acoustic constants like speaker identity from the signal to create a single-stream discrete representation of speech that captures linguistic content, including suprasegmental features. Experiments show that Kanade achieves state-of-the-art speaker disentanglement and linguistic availability while maintaining competitive reconstruction quality.

## 1 Introduction

In the past decade, natural language processing has made tremendous progress. This was enabled by the advent of language models pretrained using self-supervised learning. The power of this approach was demonstrated by encoders such as BERT (Devlin et al., 2019) and later next-token prediction models like GPT, which could perform various tasks without explicit training (Brown et al., 2020). Spoken language processing has followed a similar path. Supervised models are still popular for tasks like Automatic Speech Recognition (ASR), but for others the state-of-the-art (SOTA) often uses pretrained self-supervised models within a larger task-specific architecture (Mohamed et al., 2022).

The next-token prediction framework has also been applied to speech in pure spoken language models (SLMs) (Lakhotia et al., 2021), TTS (Chen et al., 2025), and speech-to-speech translation (Lee et al., 2022). In text language models (LMs), the tokenizer splits text into subword units. In autoregressive speech models, the speech encoder plays a similar, but more demanding role. In contrast to text, which is already a semantically dense discrete representation of human language, recordings of speech are continuous waveforms that also include other acoustic information such as background noise and speaker identity. This makes extracting meaningful representations a difficult task.

We often want encoded representations to be discrete (Mousavi et al., 2025). These are called *speech tokens* and they align with our intuition that linguistic units such as syllables and words are discrete. They are convenient because they allow us to use the architectures of the text LMs that have been so successful. Speech tokens also naturally mix with text tokens, making it convenient to build multi-modal LMs or initialize training with a pretrained text LM (Hassid et al., 2023).

For spoken language modeling, an ideal speech tokenizer should:

**Surface linguistic information** Just like text, good representations should surface the basic units of language (Borsos et al., 2023; Guo et al., 2025). This includes phonetic and prosodic (intonation, stress, and rhythm) information (Kharitonov et al., 2022). Perhaps the best reason to prefer SLMs over text LM cascades is that an SLM can understand and output prosodic features. In human discourse, prosody is used to segment speech (Mehler et al., 1981), distinguish words, parse ambiguous sentences (Kjelgaard & Speer, 1999), draw attention to specific information (Bolinger, 1972), indicate forward references (Gernsbacher & Jescheniak, 1995), and indicate turn-taking (Cutler & Pearson, 1986), among other uses. For more on the role of prosody in human language, see Cutler et al. (1997) and Dahan (2015). When representations are rich in low-level phonetic and prosodic information, we can recover higher-level features like morphology or syntax.

**Suppress non-linguistic information** Importantly, speech tokens should be similar regardless of speaker or background conditions: any acoustic instance of /a/ should be recognized as belonging to the class /a/, regardless of the situation in which it is spoken. This is similar to how image encoders are often optimized to produce representations that encode the identity of the pictured object rather than channel or environment properties like orientation, lighting, and camera characteristics.

The neural networks used in downstream models can learn more efficiently if we provide them with representations that contain only relevant information (Tishby & Zaslavsky, 2015). Just as it is wasteful to learn models of images at the pixel level, which is correlated and noisy (van den Oord et al., 2017), language modeling on verbose representations may be wasteful. We also need to be careful to create representations with short sequence lengths, since the transformers (Vaswani et al., 2017) often used in language models perform poorly on long sequence lengths (Tay et al., 2020).

**Enable high-quality reconstruction** For SLMs to output high quality speech, they must produce representations that can be turned into a high-fidelity waveform (Borsos et al., 2023; Guo et al., 2025).

These goals are often in conflict. It can be difficult to (1) provide only relevant (i.e., linguistic) information, and (2) preserve environment and speaker characteristics for reconstruction. However, disentanglement sidesteps this dilemma: ideal disentanglement would perfectly separate speech into linguistic and non-linguistic content. The former could be used for linguistic tasks like ASR or TTS, and the latter can be used only when necessary for speaker-related tasks or speech synthesis. Disentangled representations have been shown to make downstream models easier to train and require less data to generalize well (Higgins et al., 2017; van Steenkiste et al.). They are also interpretable and allow for more control (e.g., disentangling speaker identity allows for voice conversion).

It is common to separate speech into time-varying content and acoustic invariants. Since many linguistically irrelevant features like speaker identity and microphone characteristics are constant, this allows the content stream to contain easily-accessible linguistic information, while relegating information necessary for reconstruction to a separate representation. Martín-Cortinas et al. (2024) have shown that using only the content stream can improve the performance of downstream language modeling. The authors hypothesize that this is because the model learns the content distribution, rather than a more complicated joint distribution of speaker information and content. That is, we can avoid the tradeoff described above by preserving reconstruction information, but *not feeding it to models that don't need it*.

In this work, we present Kanade, a disentangled single-layer speech tokenizer. Kanade uses WavLM features to produce a stream of discrete tokens for time-varying content and a global embedding for acoustic invariants. Among speech codecs, Kanade achieves high reconstruction quality SOTA metrics on 1) **lexical availability** as measured by downstream ASR and TTS tasks, 2) **paralinguistic availability** as measured by speaker and emotion discrimination, and 3) **speaker disentanglement** as measured by voice-conversion performance and speaker discrimination tasks, all despite a low marginal bitrate.

At inference time, it usually suffices to use only the content branch. Kanade's excellent disentanglement ensures that the global embedding does not encode content, so it does not need to be calculated for linguistic discriminative tasks like ASR or intent classification. Generative models like SLMs only need to generate a content stream, which can then be decoded to speech using a single baked-in global embedding. This fixed embedding might be considered the "voice" of the model.

Our contributions:
- We build a simple and lightweight speech tokenizer that achieves best-of-class disentanglement only by restricting the flow of information rather than auxiliary methods.
- To measure the suitability of speech tokenizers for speech modeling, we assemble a suite of metrics measuring reconstruction, ease of language modeling, and performance on downstream tasks. We calculate these metrics on a wide variety recent open-source speech tokenizers, including ours.
- We demonstrate that a single-layer speech codec can have competitive SLM performance.
- Along the way, we document the approaches we considered, trade-offs we made, and a rich set of ablations. While it is our hope that Kanade is useful as it is, we also want to create a strong foundation for future work. (Code: `https://anonymous.4open.science/r/kanade-code`, Audio samples: `https://anonymous-speech-research.github.io/demo2/`)

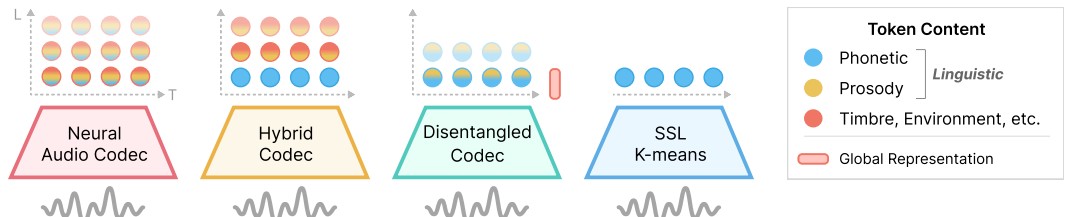

Figure 1: **Comparison of information distribution in major classes of speech tokenizers.** Color gradients represent mixed content. Adapted from SpeechTokenizer (Zhang et al., 2024). Kanade is a single-layer disentangled codec.

## 2 RELATED WORK

Self-supervised representations such as those from wav2vec 2.0 (Baevski et al., 2020), Hu-BERT (Hsu et al., 2021), and WavLM (Chen et al., 2022a) contain readily available phonetic (Pasad et al., 2021) and prosodic information, as well as easily separable speaker information (Kamper et al., 2025). The earliest SLMs used these representations by discretizing them using k-means clustering (Lakhotia et al., 2021). Unfortunately, k-means tokens from layers selected for phonetic information largely drop speaker and prosodic information, making them unsuitable for prosody modeling and resynthesis (Kharitonov et al., 2022; Polyak et al., 2021; Sicherman & Adi, 2023).

To mitigate this issue, AudioLM (Borsos et al., 2023) uses SSL-based tokens in combination with a neural audio codec (NAC). It generates SSL tokens which are then converted to NAC tokens and then to speech. This design allows the main language model to focus on modeling phonetically-rich tokens, but then uses a different model to fill in the acoustic details. This suffers from an information bottleneck: the SLM cannot pass information about how to vocalize the prosody-poor SSL tokens it generates. SpeechTokenizer (Zhang et al., 2024) is a hybrid codec that distills its first RVQ (Gray, 1984) layer from HuBERT representations. While this removes the need for a separate SSL encoder, SLMs using it still require an AudioLM-like complex multi-stage generation process.

Ye et al. (2025b) present a single-layer codec with FSQ. This has the potential to reduce complexity in autoregressive models, since there is no need for an additional step to produce finer tokens, and also allows models to better attend to suprasegmental features. However, these tokens are not disentangled, and so require downstream models to learn a more high-entropy distribution.

RepCodec (Huang et al., 2024) uses a VQ-VAE architecture to quantize SSL features, efficiently capturing semantic information. A unit-based vocoder is then trained for speech reconstruction. Kanade's content branch is inspired by this work, but shows end-to-end training with speech reconstruction can improve prosody and speech quality.

Most disentangled speech tokenizers use a multi-branch architecture along with at least one additional method to encourage disentanglement, such as time invariance (Ren et al., 2024), data augmentation (Guo et al., 2024), supervision (Ju et al., 2024), or using pretrained models (Zheng et al., 2024). Conversely, Kanade achieves disentanglement using only a two-branch architecture. To our knowledge, only BiCodec (Wang et al., 2025a) has attempted this. However, it has a more complicated global branch and our work demonstrates that it does not achieve good disentanglement.

For more details, see Appendix A. Figure 1 illustrates each of the main speech tokenizer types.

## 3 METHOD

The major components of our model are illustrated in Figure 2. First, we use an SSL encoder to extract SSL features from various layers. Features from deep layers, associated with phonetic content (Pasad et al., 2023), go into a *content* branch (top gray box, Section 3.1.1) which further encodes the speech and then quantizes it into tokens (green circles). Features from shallow layers, associated with speaker characteristics (Chen et al., 2022b), go into a *global* branch (bottom gray box, Section 3.1.2) which produces a single continuous embedding (red square). The decoder (right side of Figure 2, Section 3.1.3) reconstructs the waveform from the content tokens and global

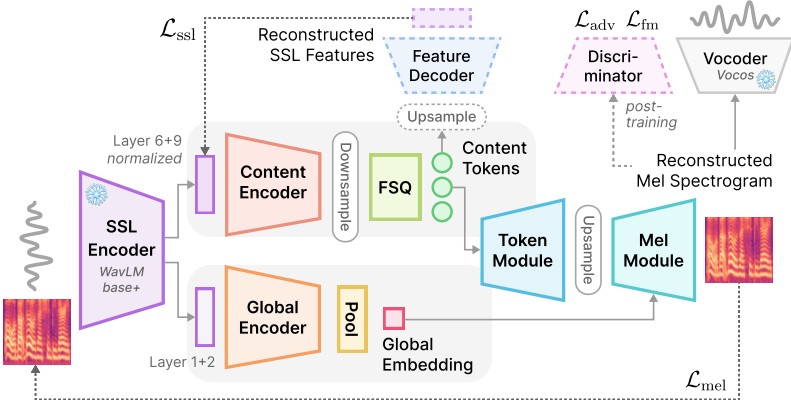

Figure 2: **Model architecture of Kanade**

embedding. We train using SSL feature and mel spectrogram reconstructions losses (Section 3.2.1), then post-train using adversarial losses (Section 3.2.2). To summarize our approach:

**SSL reconstruction loss** on content-rich SSL features emphasizes phonetic information.

**Mel reconstruction loss** is sensitive to supresegmental features, so encourages the content branch to include them.

**A global branch** provides a path for non-linguistic information to flow through. Feature reconstruction loss is relatively insensitive to this information, so the bitrate-constrained content encoder is encouraged to drop it.

### 3.1 ARCHITECTURE

**SSL encoder** SSL features already contain the information that we would like to extract from speech, including not only lingusistic, but also easily separable reconstruction-related information (Kamper et al., 2025). Therefore, it is easier to reconfigure these than start with the raw audio or mel spectrogram. See Appendices C.1 and C.2 for layer selection ablations.

#### 3.1.1 CONTENT BRANCH

**Content encoder** We average the content layers' representations and normalize each dimension to zero mean and unit variance. We pass these features through a transformer encoder, selected for its strong modeling ability (see Table 9 for an ablation). We use local window attention in all our transformers to bias the model towards encoding information near its source and because it is cheaper to calculate. The encoder outputs are temporally downsampled via a strided convolution.

**Vector quantization (FSQ)** We use a VQ-VAE (van den Oord et al., 2017) architecture for extracting discrete tokens given its success in prior work (Défossez et al., 2023; Huang et al., 2024). Unfortunately, the vector quantization method used by van den Oord et al. (2017) is sensitive to initialization, prone to codebook collapse, and can have difficulty keeping up with constantly moving encoder outputs (Łańcucki et al., 2020). Though previous work uses residual vector quantization (RVQ) (Gray, 1984) to alleviate these problems, we wanted to produce one token per timestep so opted to use Finite Scalar Quantization (FSQ) (Mentzer et al., 2024) to quantize encoder outputs. FSQ is simple and avoids many of the problems caused by a dynamic codebook.

To obtain tokens, representations from the content encoder are projected to the FSQ dimension, quantized, and represented by their indices in the implied codebook.

#### 3.1.2 GLOBAL BRANCH

The goal of the global branch is to capture information about the audio that does not change over time. Hence, we produce only one global embedding for the entire utterance. Since linguistic information can only be conveyed by features that can change, nearly all of it is forced into the content branch. Ablation confirms this (see Table 6).

The global branch architecture is inspired by NeXt-TDNN (Heo et al., 2024), which uses the ECAPA-TDNN (Desplanques et al., 2020) architecture with modified ConvNeXt (Liu et al., 2022) blocks. We did not use a transformer for the global branch because there is no long-range structure that we would like to capture. The global embedding is not discretized because we don't expect it to be used in autoregressive modeling. Li et al. (2024) showed that discretizing it may be detrimental. Furthermore, we show in Appendix D.3 that the continuous representation can be freely manipulated to condition the decoder.

**Global encoder** The shallow SSL representations for the global branch are averaged, but not normalized. They are then passed to the global encoder, which is a stack of standard ConvNeXt blocks.

**Attentive stats pool** To obtain one embedding for the entire sequence, we use an attentive stats pool (Okabe et al., 2018), following ECAPA-TDNN. An ablation using average pooling instead shows this slightly improves reconstruction quality (see Table 8).

### 3.1.3 DECODER

The first step of decoding is to convert the content tokens back into their codes by performing a lookup in the codebook.

These are passed through two transformer-based decoder modules: the **Token Module** and **Mel Module**. This two-module design is inspired by TTS systems (Ren et al., 2021), where phonemes are put through a transformer, expanded to the spectrogram length with a duration predictor, and then put through another transformer. Since our tokens are produced at a constant rate, we use transposed strided convolution to upsample features before feeding them to the mel module instead of a duration predictor.

The mel module's role is to produce a final mel spectrogram. It is conditioned by the global embedding using adaLN-Zero (Peebles & Xie, 2023). All timesteps receive the same conditioning. We choose adaptive layer normalization based on its success in AdaSpeech (Wu et al., 2022) and use the zero variant because it has better training characteristics. However, ablations in Table 8 show that our architecture is not very sensitive to the way decoding is conditioned. A convolutional post-net is applied at the end to refine the generated spectrogram.

We target a mel spectrogram rather than a waveform mainly to make model training easier. The focus of our work is token quality and we found it sufficient to use Vocos (Siuzdak, 2024) as a final step to convert the mel spectrogram to a waveform.

## 3.2 TRAINING OBJECTIVES

### 3.2.1 MAIN TRAINING PHASE

**Feature reconstruction** Since the SSL representations that the content branch uses surface useful linguistic information, we use a feature reconstruction loss to preserve that information in our tokens. Ablation shows this is very important, as seen in Table 6. To compute this, we convert the tokens back into their codes and upsample with a transposed strided convolution to the SSL frame rate. We then pass these to the transformer-based **Feature Decoder** to reconstruct the input. We compare the results with the input to the content encoder and compute the L2 loss $\mathcal{L}_{\text{ssl}}$, as was done by RepCodec (Huang et al., 2024). The feature decoder is used in training only.

**Mel reconstruction** We compute L1 loss from the reconstructed mel spectrogram to obtain $\mathcal{L}_{\text{mel}}$, following convention (Kim et al., 2021).

We combine these two losses to obtain $\mathcal{L} = \mathcal{L}_{\text{mel}} + \alpha \mathcal{L}_{\text{ssl}}$ in the main training phase. We also tried splitting this into two stages, with only SSL loss at first, then switching to mel reconstruction loss. However, we found this caused the encoder to ignore some prosodic features (see Table 6). This is similar to how k-means (which is also computed using L2 distances in SSL feature space) loses prosodic information before phonetic information (Kharitonov et al., 2022; Onda et al., 2025), so we suspect that distances in phonetically rich SSL layers are not very sensitive to prosodic features.

### 3.2.2 GAN POST-TRAINING

With only the main training phase, the model produces intelligible speech (see ablations in Table 10), but the spectrogram is blurry, degrading audio quality. Wu et al. (2023) show that introducing GAN (Goodfellow et al., 2014) post-training on the decoder can restore finer details.

To avoid passing gradients through the vocoder, we compare the mel spectrograms rather than the waveforms. The discriminator splits the mel spectrum into frequency bands and feeds each to a stack of convolutional layers. The per-band results are concatenated back together, a final convolution is applied, and the results are downsampled by 2D average pooling in the time and frequency dimensions, producing the final discriminator output. This formulation was originally proposed in DAC (Kumar et al., 2023). We use adversarial loss $\mathcal{L}_{adv}$ and feature matching loss $\mathcal{L}_{fm}$ as described in Vocos (Siuzdak, 2024). During post-training, only the global branch and the decoder are updated.

The post-training objective is $\mathcal{L}_{post} = \mathcal{L}_{mel} + \beta\mathcal{L}_{adv} + \gamma\mathcal{L}_{fm}$.

## 4 EXPERIMENTS

### 4.1 TRAINING SETUP

Details on our model and training configurations can be found in Appendix E.1. We train our models using all training sets of LibriTTS (Zen et al., 2019), a multi-speaker English corpus containing 586 hours of audiobook speech sampled at 24kHz. LibriTTS is derived from the same materials as the LibriSpeech (Panayotov et al., 2015) corpus.

### 4.2 BASELINES

We compare Kanade with a variety of SOTA speech codecs, including hybrid codecs, single-stream codecs, and disentangled codecs. See Appendix E.5 for more details. SpeechTokenizer (Zhang et al., 2024) is abbreviated as ST.

We also train several reference models that change the way content is encoded. We train k-means reference models (KM) that use the same SSL representations used by the content encoder (see Section 3.1.1—normalizing before clustering is consistent with prior work (Borsos et al., 2023)). These features are downsampled with average pooling and clustered using k-means, which is trained on the LibriTTS train subsets. A separate continuous reference model (Cont.) is trained by replacing both encoding branches with full-resolution (50Hz) continuous SSL features. Since we remove the global branch, these are an average of all four layers used in our main models.

### 4.3 EVALUATION

We evaluate generated speech according to: (1) **intelligibility**: word/character error rate (WER/CER) using Parakeet [1]; (2) **quality**: MUSHRA[2], UTMOS (Saeki et al., 2022), ViSQOL (Chinen et al., 2020), and Mel L1; (3) **speaker identity**: speaker embedding cosine similarity (SIM) using WavLM Base+ for Speaker Verification[3] (WavLM-SV) and mel cepstral distortion (MCD); and (4) **prosody**: log F0 Pearson correlation (F0Corr) and root mean square error (F0RMSE), extracted by SWIPE (Camacho & Harris, 2008). Evaluation code is largely adapted from VERSA (Shi et al., 2025).

We also evaluate our models and baselines using various downstream tasks. The relevant metrics will be introduced along with their results.

---

[1] https://huggingface.co/nvidia/parakeet-tdt-0.6b-v3

[2] More details about the listening test are in Appendix E.4.

[3] https://huggingface.co/microsoft/wavlm-base-plus-sv

Table 1: **Speech reconstruction results.** The top group includes reference metrics. Only models that are best in some metric are included. The bold numbers are the best in their group. For all results, see Table 19 in the appendix.

| Model | Token Rate | Intelligibility | | Quality | | | | Speaker | | Prosody | |
|---|---|---|---|---|---|---|---|---|---|---|---|
| | | WER↓ | CER↓ | MUSHRA↑ | UTMOS↑ | ViSQOL↑ | Mel L1↓ | SIM↑ | MCD↓ | F0Corr↑ | F0RMSE↓ |
| Ground Truth | – | 1.9 | 0.6 | 78.0 | 4.07 | 5.00 | – | – | – | – | – |
| Cont. 50Hz | – | 2.0 | 0.6 | 72.1 | 3.90 | 4.54 | 0.74 | 0.99 | 3.91 | 0.94 | 0.04 |
| KM 12.5Hz | 12.5 | 3.0 | 1.1 | 72.1 | 4.04 | 3.33 | 1.44 | 0.96 | 7.45 | 0.66 | 0.15 |
| KM 25Hz | 25 | 2.7 | 1.0 | 72.4 | 4.07 | 3.40 | 1.30 | 0.96 | 6.76 | 0.67 | 0.15 |
| *Multi-layer* | | | | | | | | | | | |
| FACodec | 480 | **2.1** | **0.7** | 81.4 | 4.11 | 4.27 | 0.76 | 0.98 | 5.17 | 0.94 | **0.04** |
| PAST | 400 | **2.1** | **0.7** | **82.4** | **4.18** | **4.32** | 0.72 | **0.99** | 4.42 | 0.92 | **0.04** |
| ST | 400 | **2.1** | **0.7** | 76.0 | 3.90 | 4.26 | 0.72 | 0.98 | 4.72 | 0.92 | 0.05 |
| DualCodec | 100 | **2.1** | **0.7** | 75.6 | 4.12 | 4.28 | **0.66** | 0.98 | **4.08** | **0.95** | **0.04** |
| *Single-layer* | | | | | | | | | | | |
| X-Codec 2 | 50 | 2.5 | 0.9 | 77.0 | 4.13 | **4.12** | **0.77** | **0.98** | 4.92 | 0.90 | 0.06 |
| BiCodec | 50 | 2.5 | 0.9 | 75.0 | 4.18 | 4.09 | 0.94 | **0.98** | 5.22 | **0.91** | 0.05 |
| WavTokenizer | 40 | 9.4 | 4.7 | 72.1 | 3.57 | 3.55 | 1.00 | 0.92 | 6.17 | **0.91** | 0.07 |
| StableCodec | 25 | 5.7 | 2.6 | **79.3** | **4.31** | 3.50 | 1.28 | 0.93 | 7.29 | **0.91** | **0.05** |
| Kanade 12.5Hz | 12.5 | 3.3 | 1.3 | 74.6 | 4.17 | 3.69 | 1.25 | 0.97 | 6.82 | 0.85 | 0.10 |
| Kanade 25Hz | 25 | **2.4** | **0.8** | 75.0 | 4.16 | 3.86 | 1.02 | 0.97 | 5.67 | 0.88 | 0.07 |

## 4.4 RESULTS

### 4.4.1 RECONSTRUCTION

We evaluate speech reconstruction on LibriSpeech `test-clean`. The results are shown in Table 1. Kanade maintains high speech quality and achieves the best WER among single-layer codecs and even approaches the heaviest RVQ models. The k-means reference models have significantly degraded audio quality and prosody preservation (0.68 KM 25Hz vs. 0.88 Kanade 25Hz on F0Corr), even when conditioned by the global embedding. This indicates that our content tokens capture prosodic information better than k-means tokens, which is further confirmed by probing (see Appendix D.1).

For MUSHRA confidence intervals, see Table 21. For results on out-of-distribution data, see Appendix D.6. For reconstruction metrics on longer samples, see Appendix D.8.

### 4.4.2 DISCRIMINATIVE DOWNSTREAM TASKS

**ASR** To measure the availability of lexical information in tokens, we train decoder-only ASR models following (Huang et al., 2024). The models are trained to predict SentencePiece (Kudo & Richardson, 2018) text tokens conditioned on speech tokens. The models are trained on tokens extracted from the LibriSpeech training sets. We use all the token layers, but exclude global representations. These models are evaluated on LibriSpeech `test-clean`. Appendix E.3 contains further details.

The results are shown in Table 2. Kanade 25Hz achieves the lowest WER (7.1%), drawing nearer to the performance of k-means tokens. Evaluation on spontaneous speech yields similar results (see Appendix D.7). This indicates that our single stream captures rich and easily-accessible lexical information. For metrics of phonetic information such as ABX and PNMI, see Appendix D.2. For a correlation analysis between lexical and phonetic metrics, see Appendix D.5.

**Speaker and emotion recognition** We train discriminative models for each tokenizer using all layers for RVQ tokenizers. For disentangled tokenizers, we report results using only the global representation or content stream. We evaluate two speaker tasks: **speaker identification** (SID) and **automatic speaker verification** (ASV). Following Jung et al. (2022), we train ECAPA-TDNN (Desplanques et al., 2020) with AAM-softmax loss (Deng et al., 2019) on representations extracted from VoxCeleb1 (Nagrani et al., 2020). We report accuracy (Acc) and equal error rate (EER) for SID and ASV, respectively. For emotion recognition (ER), we use an identical backbone with cross-entropy loss. Following Yang et al. (2024), we perform 5-fold cross-validation across the five sessions of IEMOCAP and report the unweighted average accuracy (Acc) on the four most common classes (angry, sad, neutral, and happy/excited). Appendix E.3 contains further details.

Table 3: **Text-to-speech results**

| Model | LibriTTS test-clean | | | | | Seed-TTS-eval | |
|---|---|---|---|---|---|---|---|
| | WER↓ | SIM↑ | UTMOS↑ | Quality↑ | Prosody↑ | WER↓ | SIM↑ |
| Ground Truth | 2.3% | – | 4.13 | 74.9 | 80.9 | 1.9% | – |
| KM 12.5Hz | 4.6% | 0.95 | 3.96 | 72.0 | 67.0 | 5.4% | 0.93 |
| KM 25Hz | 4.3% | 0.95 | 4.05 | 74.9 | 75.9 | 4.9% | 0.93 |
| CosyVoice 2 | 1.8% | 0.96 | 4.42 | 77.1 | 83.0 | 2.1% | 0.95 |
| ST | 9.7% | 0.95 | 3.95 | 75.0 | 79.0 | 11.2% | **0.94** |
| Mimi | 6.6% | 0.95 | 3.48 | 74.9 | 73.9 | 6.0% | 0.93 |
| DualCodec | 10.0% | **0.96** | 3.68 | 73.0 | 80.0 | 5.5% | **0.94** |
| PAST | 8.0% | 0.95 | 4.14 | 74.9 | 78.4 | 9.0% | **0.94** |
| TiCodec | 11.5% | 0.94 | 3.86 | 73.8 | 72.9 | 12.9% | 0.92 |
| StableCodec | 9.0% | 0.91 | 3.78 | 71.0 | 66.0 | 10.9% | 0.87 |
| BiCodec | 7.8% | 0.95 | 4.12 | 73.8 | 78.9 | 7.5% | 0.94 |
| X-Codec 2 | 6.5% | 0.95 | **4.21** | 72.0 | 78.0 | 7.2% | **0.94** |
| WavTokenizer | 13.9% | 0.92 | 3.76 | 74.5 | 77.0 | 15.6% | 0.91 |
| Kanade 12.5Hz | 5.9% | 0.95 | 4.13 | **77.1** | 77.9 | 5.7% | **0.94** |
| Kanade 25Hz | **4.2%** | 0.95 | 4.18 | 73.0 | **81.0** | **4.0%** | **0.94** |

Results are shown in Table 2. Kanade achieves SOTA performance when using only its global embedding, nearly matching the performance of WavLM features (Cont. 50Hz) on ASV EER. We see that BiCodec also performs better on these tasks when using only the global embedding, while TiCodec fails to capture speaker information in its global representation. Kanade fails on both speaker tasks when using content tokens, suggesting good disentanglement.

### 4.4.3 GENERATIVE DOWNSTREAM TASKS

**Text-to-speech (TTS)** To test text-conditioned generative modeling, we train an autoregressive phoneme-based TTS model for each tokenizer on the LibriTTS training sets. Following CosyVoice (Du et al., 2024a), speaker ID is conditioned by prepending the input with WavLM-SV speaker embeddings from the reference. Global embeddings for synthesis after TTS modeling are also extracted from the reference. To synthesize speech for RVQ-based decoders, we need to generate multiple layers of dependent tokens. Autoregressive modeling produces the highest quality results when tokens are flattened into a single token stream (Copet et al., 2023). While some works cite scalability and performance as reasons to use additional modeling step instead (Chen et al., 2025; Défossez et al., 2024), we chose to standardize on flattening for its topline synthesis quality and simplicity. Details are in Appendix E.3.

Table 2: **Downstream task results (%)**. For context, includes SOTA metrics from specialized models for ASR (Rekesh et al., 2023), SID (Saritha et al., 2024), ASV (Heo et al., 2024) and ER (Cao et al., 2025). N.C. denotes not converged.

| Model | Lexical | | Speaker | | Emotion |
|---|---|---|---|---|---|
| | WER↓ | CER↓ | SID Acc↑ | ASV EER↓ | ER Acc↑ |
| Cont. 50Hz | 4.3 | 1.9 | 92.5 | 6.2 | 73.9 |
| KM 12.5Hz | 5.8 | 2.9 | 2.9 | 38.7 | 59.7 |
| KM 25Hz | 5.8 | 3.1 | 8.4 | 28.1 | 63.0 |
| SOTA | 1.4 | – | 99.3 | 0.8 | 79.9 |
| DualCodec | 9.8 | 5.3 | 22.9 | 18.8 | 54.8 |
| ST | 8.2 | 4.2 | 60.0 | 11.7 | 60.0 |
| Mimi | 10.4 | 5.5 | 33.4 | 17.4 | 54.4 |
| PAST | 7.9 | 3.9 | 74.1 | 9.8 | 55.3 |
| X-Codec 2 | 11.0 | 6.0 | 0.2 | 39.0 | 45.8 |
| StableCodec | 11.8 | 6.3 | 0.1 | 45.0 | 42.4 |
| WavTokenizer | 18.1 | 10.3 | 13.1 | 27.4 | 48.1 |
| FACodec | – | – | 64.7 | 11.8 | 58.5 |
|   content only | 8.2 | 4.2 | 76.8 | 8.9 | 54.3 |
|   global only | – | – | N.C. | N.C. | 40.8 |
| TiCodec | – | – | 23.9 | 20.4 | 48.8 |
|   content only | 9.4 | 4.8 | 56.2 | 13.4 | 51.7 |
|   global only | – | – | 4.3 | 43.0 | 45.7 |
| BiCodec | – | – | 17.6 | 31.8 | 46.6 |
|   content only | 100.1 | 71.4 | 0.5 | 38.7 | 46.0 |
|   global only | – | – | 27.0 | 19.7 | 49.7 |
| Kanade 12.5Hz | – | – | 69.6 | 13.7 | 59.1 |
|   content only | 8.1 | 4.0 | 0.2 | 44.1 | 42.3 |
|   global only | – | – | **78.8** | **6.6** | 54.3 |
| Kanade 25Hz | – | – | 71.0 | 11.8 | **60.2** |
|   content only | **7.1** | **3.8** | 0.3 | 36.2 | 45.8 |
|   global only | – | – | 78.6 | 7.0 | 53.0 |

We randomly select 1,000 samples (4-10 seconds) from LibriTTS (Zen et al., 2019) `test-clean` and condition each with 3 reference samples from the same speaker. Quality and prosody are evaluated using MUSHRA-like listening tests. See Appendix E.4 for details. We also report Seed-TTS-eval results for comparison with other work.

The results are shown in Table 3. Kanade achieves SOTA intelligibility (4.2%, 5.9% WER) with excellent quality and prosody. This finding aligns with the ASR metrics discussed earlier: the

Table 4: **Voice conversion results.** Bold numbers are the best among tokenizers.

| Model | Lexical Content | | Quality | Speaker Timbre | | Prosody |
|---|---|---|---|---|---|---|
| | WER↓ | CER↓ | UTMOS↑ | EER↑ | Similarity↑ | S-F0Corr↑ |
| Ground Truth | 0.0% | 0.0% | 4.08 | – | – | – |
| KM 12.5Hz | 1.5% | 0.6% | 4.22 | 29.8% | 74.0 | 0.55 |
| LinearVC | 0.6% | 0.2% | 3.94 | 29.7% | 73.4 | 0.62 |
| FreeVC | 0.6% | 0.3% | 3.99 | 29.0% | 74.5 | 0.67 |
| CosyVoice 2 | 1.1% | 0.5% | 4.11 | 31.0% | 76.0 | 0.64 |
| FACodec | 0.8% | 0.4% | 3.45 | 18.6% | 62.6 | 0.66 |
| BiCodec | 1.2% | 0.6% | 3.84 | 18.5% | 71.4 | 0.61 |
| DualCodec | 21.5% | 12.9% | 2.51 | 6.8% | 52.0 | 0.54 |
| ST | 74.7% | 61.7% | 1.54 | 10.6% | 35.0 | 0.20 |
| Mimi | 120.3% | 86.8% | 3.09 | **38.5%** | **81.7** | 0.21 |
| PAST | 22.9% | 15.1% | 1.84 | 8.2% | 23.3 | 0.20 |
| TiCodec | **0.5%** | **0.2%** | 3.32 | 5.4% | 68.0 | **0.77** |
| Kanade 12.5Hz | 1.6% | 0.7% | **4.17** | 32.0% | 77.6 | 0.64 |
| Kanade 25Hz | 0.7% | 0.3% | 4.16 | 30.7% | 77.1 | 0.71 |

Table 5: **Spoken language modeling results.** Chance level is 50%.

| Model | Token rate | Vocab. size | sWUGGY↑ | sBLIMP↑ | sSC↑ | tSC↑ |
|---|---|---|---|---|---|---|
| KM 12.5Hz | 12.5 | 12 800 | 75.8 | **57.5** | 51.8 | 66.7 |
| KM 25Hz | 25 | 12 800 | 68.1 | 53.5 | 51.1 | 63.5 |
| ST | 50 | 1024 | 75.8 | 54.9 | 52.0 | 64.4 |
| PAST | 50 | 1024 | 76.8 | 53.6 | 51.8 | 59.5 |
| Mimi | 12.5 | 2048 | **77.6** | 56.1 | 52.0 | **67.8** |
| Kanade 12.5Hz | 12.5 | 12 800 | 76.6 | 55.2 | **52.1** | 65.3 |
| Kanade 25Hz | 25 | 12 800 | 69.7 | 52.4 | 51.3 | 60.0 |

stronger lexical availability in our content tokens provides easier text alignment for downstream tasks. For MUSHRA confidence intervals, see Tables 23 and 24.

**Voice Conversion (VC)** To measure disentanglement in hybrid and disentangling speech tokenizers, we combine content tokens (usually RVQ layer 1) extracted from *source* utterances with remaining tokens and embeddings extracted from *reference* (or *target*) utterances and then resynthesize. This is done for 1,000 gender-balanced (source, reference) pairs from VCTK (Yamagishi et al., 2019). We randomly select 20 source speakers, 10 target speakers, and 5 source sentences.

If the phonetic and prosodic content matches the source and the timbre matches the reference, this indicates good disentanglement between content and speaker characteristics. Lingustic content is measured using WER and prosodic correlation (S-F0Corr) is with respect to the *source*. We calculate equal error rate (EER, higher is better) following Das et al. (2020) using WavLM Base+ for Speaker Verification. We additionally conduct MUSHRA-like listening tests to subjectively evaluate speaker similarity (see Appendix E.4). Specialized VC models were included as baselines: LinearVC (Kamper et al., 2025), FreeVC (Li et al., 2023), and CosyVoice 2 (Du et al., 2024b).

Results are shown in Table 4. We observe catastrophic content degradation in the converted speech of SpeechTokenizer and Mimi, suggesting leakage of linguistic content into higher layers. We also observe poor timbre transfer in all other tokenizers. Kanade is the only speech codec that both preserves content (WER, F0Corr) and achieves high speaker similarity (EER, Similarity). Moreover, our performance matches or even surpasses specialized VC models, demonstrating that our simple architecture achieves excellent disentanglement. For the full results, see Table 20 and 22.

**Spoken language modeling (SLM)** We use the Slam (Maimon et al., 2025a) recipe to train a warm-start SLM based on Qwen-2.5-0.5B on one epoch of LibriLight (Kahn et al., 2020). Only first-layer RVQ tokens are used. We evaluate in-vocabulary sWUGGY, sBLIMP (Dunbar et al., 2021), sStoryCloze (sSC), and tStoryCloze (tSC) (Hassid et al., 2023), all of which measure accuracy in assigning higher probability to linguistically plausible inputs. Since we keep constant the SLM architecture, these metrics indirectly measure whether a tokenizer makes available the information necessary to learn higher-level linguistic structure. Only baselines that performed well in preliminary testing (see Appendix D.4) are included here. For these, only the semantic layer is

Table 6: **Ablation results.** Based on Kanade 12.5Hz without post-training.

| Model | Reconstruction | | | | | | Discriminative Downstream | | | |
|---|---|---|---|---|---|---|---|---|---|---|
| | WER↓ | MUSHRA↑ | UTMOS↑ | Mel L1↓ | SIM↑ | F0Corr↑ | WER↓ | SID Acc↑ | ASV EER↓ | ER Acc↑ |
| Kanade 12.5Hz | 3.5% | 69.0 | 4.10 | 1.27 | 0.96 | 0.84 | 8.1% | 69.6% | 13.7% | 59.1% |
| w/o Dual-branch | 6.1% | 24.0 | 2.93 | 1.66 | 0.88 | 0.66 | 10.4% | 1.3% | 35.9% | 50.3% |
| w/o Feature recon. | 8.0% | 68.5 | 4.08 | 1.25 | 0.96 | 0.84 | 14.9% | 66.8% | 14.0% | 58.6% |
| w/o End-to-end | 3.3% | 60.7 | 3.97 | 1.34 | 0.96 | 0.76 | 7.7% | 67.1% | 13.3% | 58.9% |
| w/o FSQ | 25.8% | 43.7 | 3.37 | 1.44 | 0.95 | 0.69 | 18.6% | 62.0% | 14.0% | 61.1% |

used. The resuls in Table 5 show that k-means, distilled tokens, and our tokens all have similar performance (though 25 Hz variants underperform).

## 4.5 ABLATION STUDIES

We conduct a rich set of ablation studies to verify the effectiveness of our design choices. Some results are shown here in Table 6. See Appendix C for more.

**Dual-branch design** We train a model without a global branch, using only content tokens to reconstruct both SSL features and a mel spectrogram. This model shows heavy degradation on every metric. Despite its simplicity, the global embedding is indispensable, capturing constant acoustic information and allowing the content branch to focus on linguistic content.

**SSL feature reconstruction loss** In the model trained without SSL features reconstruction loss, reconstruction and downstream ASR WERs are significantly higher. This suggests that the SSL feature reconstruction loss encourages the content branch to encode lexical information.

**End-to-end training** In this setting, we (1) train the content FSQ-VAE with only SSL feature reconstruction loss, freeze it, and then (2) train the other components with only mel spectrogram reconstruction loss. This is similar to the approach used by Huang et al. (2024). While this 2-stage method has a slightly lower WER, the speech quality, in particular prosody, degrades. This demonstrates that end-to-end training with dual objectives can extract more prosodic information without losing much lexical information.

**FSQ** We replace FSQ with ordinary VQ (van den Oord et al., 2017), using exponential moving average (EMA) codebook (decay 0.8), k-means initialization, and random restart for dead codes (Dhariwal et al., 2020). The results show a serious degradation on nearly every metric, linguistic information (WER, F0Corr). This observation aligns with findings reported by Mentzer et al. (2024). FSQ yields better results and removes the need to tune extra hyperparameters.

## 5 CONCLUSION

We introduced Kanade, a speech tokenizer that extracts compact, linguistically rich single-stream tokens suitable for both generative and discriminative modeling. Kanade draws closer to the ideal speech tokenizer, with excellent information preservation and linguistic availability. It starts with SSL features that already expose relevant information, allowing training with only 600 hours of data and 120M unfrozen parameters. It then cleanly disentangles time-invariant features and linguistic content only by restricting the flow of information. This allows downstream models using Kanade tokens to achieve better results than baselines on various speech tasks, including ASR and TTS. Despite the simplicity of Kanade's disentanglement approach, speaker recognition and voice conversion results reveal the best disentanglement among tested tokenizers. Furthermore, competitive SLM results show that simple autoregressive language modeling with a single stream of tokens is possible without giving up the speech quality benefits of a reconstruction-oriented codec.

While Kanade does not introduce any new components, it shows that a simple architecture with simple losses is enough to create a high-performing speech tokenizer.

## ETHICS STATEMENT

We recognize the potential for abuse using our models, especially when used for voice conversion. However, during GAN post-training the discriminator was very strong and we had to hobble it

severely, indicating that the audio generated by our model can easily be detected. We acknowledge that the pretrained SSL encoder and our training data have biases and encourage anyone using our architecture to use debiasing techniques or train with a larger, more diverse dataset, as we also plan to do in the future.

## REPRODUCIBILITY STATEMENT

All training data (Section 4.1) and evaluation data (described alongside each metric) we use, with the exception of TIMIT (Garofolo et al., 1993), is freely available. TIMIT is available from the Linguistic Data Consortium for a fee. Baselines (detailed in Section E.5) are tested using their official open-source implementations and checkpoints. We made our best effort to provide model architecture and training details (see Sections E.1 and E.3). We are committed to the integrity of our work and will answer any questions regarding it by email. Some audio samples produced by our models can be found on our demo page: `https://anonymous-speech-research.github.io/demo2/`. We will publicly release our training code, evaluation code, and checkpoints before submitting the camera-ready paper.

## LLM USAGE

We used LLMs for coding assistance, literature discovery (to summarize paper contents and retrieve related works), and some research ideation (to discuss experiment design, etc.), but did not use them to draft or review this paper (with an exception for writing scripts to format LaTeX tables).

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

## A  SURVEY OF RELATED WORK

Speech tokenizers of recent years generally fall into two categories: neural audio codec (NAC)-based and SSL-based. NACs (Zeghidour et al., 2022; Défossez et al., 2023) are designed to compress audio and are mainly trained with reconstruction objectives (Défossez et al., 2023). SSL representations are designed to capture the structure of speech and are trained using contrastive loss (Baevski et al., 2020) or masked prediction (Hsu et al., 2021). Using k-means, SSL representations can be clustered into tokens that expose high levels of phonetic information (Hsu et al., 2021; Choi et al., 2024). Both token types have been used in spoken language models (SLMs) (Lakhotia et al., 2021; Borsos et al., 2023).

Efforts to improve speech tokenization have followed two main streams of work:

**Unification**  In speech contexts, we often want both the information-preserving properties of reconstruction-oriented NACs and the linguistic availability of SSL tokens. We need the former to generate high-quality speech and the latter to understand speech and produce coherent outputs (Borsos et al., 2023). To this end, we have seen the development of NACs with more linguistic availability (Zhang et al., 2024) and SSL discritization methods that improve the quality of generated speech (Huang et al., 2024).

**Disentanglement**  It is trivial to achieve a high level of phonetic availability (by representing speech as text) or acoustic preservation (by representing it as a waveform). The difficulty lies in doing both at the same time, so disentanglement has been a recurring theme in speech tokenization efforts. This can help (1) make phonetic content more available (Qian et al., 2022); (2) achieve flexible control of individual aspects of speech (Choi et al., 2023); and (3) reduce information redundancy (Ren et al., 2024).

In the following paragraphs, we select a few works with at least one of these goals.

**Hybrid speech codecs**  These codecs are usually based on NAC architectures, but are designed to enhance lexical information by: (1) *knowledge distillation*, typically using SSL encoders as teacher models (Zhang et al., 2024; Défossez et al., 2024; Jiang et al., 2024; Khurana et al., 2025); (2) *directly using SSL features* as input (Ye et al., 2025a; Wang et al., 2025a; Li et al., 2025; Hussein et al., 2025; Liu et al., 2024); (3) *text conditioning* of the encoder and/or decoder (Tseng et al., 2025; Wang et al., 2025b); or (4) *supervision*, typically using phonemes (Ju et al., 2024) or text (Har-Tuv et al., 2025).

Though largely effective, these methods have at least one of the following limitations: (1) relying on multi-layer token structure, which complicates the architecture of downstream models; (2) having parallel encoders, resulting in architectural redundancy; or (3) needing extra annotations and limiting scalability. Following this line of research, Kanade improves on these by using only SSL features that are already rich and learning single-layer linguistically meaningful tokens in an unsupervised way, with a simple and elegant architecture.

**SSL tokens**  K-means tokenization, popularized by HuBERT (Hsu et al., 2021), is the prevalent method of discretizing SSL speech representations. It is simple and extracts phonetic information suitable for SLMs (Lakhotia et al., 2021; Borsos et al., 2023). However, performing k-means on the layers typically associated with high phonetic content eliminates too much information that is required for faithful resynthesis (Sicherman & Adi, 2023).

To preserve additional information, Shi et al. (2024) propose applying k-means iteratively on the residuals, creating a multi-layer token structure reminiscent of multi-layer NACs.

Others (Huang et al., 2024; Zhang et al., 2025) have used VQ-VAE to learn a discrete latent space. The structure of our content branch (Section 3.1.1) was based on these methods. Auxiliary losses are sometimes added to improve robustness (Gat et al., 2023; Chang et al., 2023; Messica & Adi, 2024) or integration with SLMs (Turetzky & Adi, 2024).

Some methods extract syllable-like units, producing extremely low-bitrate coarse tokens. This significantly improves SLM performance (Baade et al., 2025; Cho et al., 2025).

UniWav (Liu et al., 2025) adds discriminative and generative objectives in a unified pre-training framework. The k-means tokens obtained from their representation exhibit improved reconstruc-

tion quality. Since pretraining requires significant computational resources, our work explores the possibility of extracting better speech tokens from existing pretrained SSL models.

**Disentangled speech representations** Here we outline techniques for disentangling different aspects (phonetic content, prosody, speaker timbre, style, etc.) of speech. We include techniques from research on voice conversion for completeness. (1) *information bottlenecks*, including vanilla autoencoders (Qian et al., 2019), k-means quantization (Polyak et al., 2021; Huang et al., 2022), and VQ-VAE (Chorowski et al., 2019; Wu et al., 2020; Tjandra et al., 2021; Zhang et al., 2025); (2) *structured priors* that separate local and global information (Hsu et al., 2017; Wang et al., 2018; Yin et al., 2022); (3) *contrastive learning* (Qian et al., 2022; Tang et al., 2022) and *invariance learning* (Chan et al., 2022; Choi et al., 2023; Chang et al., 2023; Ren et al., 2024); (4) *adversarial learning* techniques such as GAN (Kameoka et al., 2018) and gradient reversal (Ju et al., 2024; Łajszczak et al., 2024); and (5) *direct supervision* (Hussain et al., 2023; Ju et al., 2024; Har-Tuv et al., 2025). Some less popular techniques include *mutual information loss* (Wang et al., 2021; Yang et al., 2022; Lian et al., 2022), *instance normalization* (Chou et al., 2019; Lin et al., 2021), and *linear transformations in the feature space* (Kamper et al., 2025). Some models use more than one technique. For example, TiCodec (Ren et al., 2024) uses vector quantization, structured priors, and invariance learning.

Following previous work, Kanade uses VQ-VAE as an information bottleneck. It assumes that acoustic constants by including a global branch to provide a path for them to be preserved outside of the main token stream, disentangling them from linguistic content. Critically, it does not introduce the complexity of extra training objectives found in some of the methods above.

## B    LIMITATIONS AND FUTURE WORK

Since the SSL encoder we use is based on a bidirectional transformer, our tokens are not streamable, requiring audio chunking and limiting applicability in some scenarios. Since the effective receptive field of SSL encoders is limited (Meng et al., 2025), this can be solved by distilling a streamable encoder (Choi et al., 2025) and modifying our architecture to a streaming design, as done by Défossez et al. (2024).

Our content tokens are produced at a constant rate, which may lead to information redundancy and reduce alignment with linguistic categories. We hope to adopt approaches pioneeered by Baade et al. (2025) and Cho et al. (2025) to enable variable-rate tokenization, mitigating these issues. Although we achieve excellent separation of dynamic content and acoustic constants, currently it is still not possible to further disentangle the content. As shown by the Gigaspeech experiments (see Appendix D.6), our current approach is sensitive to dynamic background noise such as music. Furthermore, it could be useful to further separate linguistic content into phonetic and prosodic features for better flexibility.

Since the focus of this paper is to improve linguistic availability and information preservation in discrete speech tokens, we did not experiment with any vocoding settings other than targeting a mel spectrogram and using Vocos to generate a waveform. To improve audio quality, we might consider using a more advanced decoder.

For more limitations regarding out-of-distribution data, see Appendix D.6.

# C    ADDITIONAL ABLATION STUDIES

## C.1    CONTENT BRANCH

Table 7: **Content branch ablation results**

| Model | Reconstruction | | | | | Downstream |
|---|---|---|---|---|---|---|
| | WER↓ | UTMOS↑ | Mel L1↓ | SIM↑ | F0Corr↑ | WER↓ |
| Kanade 12.5Hz | 3.5% | 4.10 | 1.27 | 0.96 | 0.84 | 8.1% |
| Token rate 6.25Hz | 14.0% | 3.55 | 1.73 | 0.95 | 0.65 | 15.8% |
| Codebook size 3125 | 4.9% | 4.05 | 1.33 | 0.96 | 0.79 | 10.0% |
| Layer 6 | 4.2% | 4.09 | 1.29 | 0.96 | 0.82 | 12.5% |
| Layer 9 | 3.5% | 4.07 | 1.28 | 0.96 | 0.81 | 7.5% |
| Layer 12 | 3.5% | 4.04 | 1.29 | 0.96 | 0.80 | 7.8% |
| Layer 9+12 | 3.6% | 4.08 | 1.29 | 0.96 | 0.81 | 7.4% |
| Layer 1–12 weighted-sum | 3.5% | 4.07 | 1.30 | 0.96 | 0.80 | 8.7% |

Table 8: **Global branch ablation results**

| Model | Reconstruction | | | | | Downstream | | |
|---|---|---|---|---|---|---|---|---|
| | WER↓ | UTMOS↑ | Mel L1↓ | SIM↑ | F0Corr↑ | SID Acc↑ | ASV EER↓ | ER Acc↑ |
| Kanade 12.5Hz | 3.5% | 4.10 | 1.27 | 0.96 | 0.84 | 69.6% | 13.7% | 59.1% |
| Layer 6+9 | 3.6% | 4.06 | 1.46 | 0.94 | 0.79 | 71.7% | 13.8% | 64.1% |
| Layer 1–4 weighted-sum | 3.7% | 4.09 | 1.28 | 0.97 | 0.82 | 75.4% | 10.9% | 60.4% |
| Mel | 3.6% | 3.81 | 1.23 | 0.93 | 0.81 | 46.3% | 20.0% | 56.0% |
| Avg pooling | 3.7% | 4.10 | 1.29 | 0.96 | 0.81 | 70.3% | 12.6% | 59.5% |
| Conditioning: full decoder | 3.8% | 4.09 | 1.27 | 0.96 | 0.82 | 70.9% | 11.8% | 59.5% |
| Conditioning: addition | 3.8% | 4.09 | 1.25 | 0.97 | 0.83 | 82.6% | 12.7% | 59.5% |

Results of ablation on the content branch are shown in Table 7.

We tried decreasing the **token rate** and **effective codebook size**. When the token rate is halved (85bps), the linguistic content and speech quality is unacceptable. On the other hand, the codebook size has more moderate effect on information capacity, since the bitrate decreases logarithmically with codebook size. In the model with 3,125 codes ($\sim 1/4$ of the original codebook size, 145bps), WER and F0Corr mildly degrade.

We also study **SSL feature layer selection** for the content branch input. We observe a pattern consistent with Pasad et al. (2023): shallow layers provide more acoustic information that benefits audio quality and prosody preservation; deep layers offer more phonetic information. We find the 9th layer ($3/4$ the way through) is a good balance point. Zhang et al. (2025) observed a similar result for HuBERT-large. Adding layer 6 to layer 9 improves speech quality and prosody, without losing much lexical availability (+0.6% downstream WER), so we stick to this combination. We also experimented with a learnable weighted-sum of all layers, with suboptimal results. Interestingly this model distributes over 80% of the weight to the deepest layer.

## C.2    GLOBAL BRANCH

Results of ablation on the global branch are shown in Table 8.

For **SSL feature layer selection**, we experiment with using the same combination of SSL layers as our content branch (layers 6 and 9). This improves emotion recognition performance (64.1% ER Acc) but Mel L1 and prosody metrics are worse. This suggests that deeper SSL features offer more paralinguistic semantics but less prosodic information.

In a model with a learnable weighted-sum of layers 1–4, we notice increased speaker recognition performance (75.4% SID Acc) but slightly worse intelligibility. Other metrics remain similar. During training, we find the model distributes over 50% of the weight to layer 1, indicating that the global branch prefers information from earlier layers. For simplicity, we stick to a combination of layers 1 and 2 for better intelligibility while maintaining reasonably high downstream performance.

Table 9: **Backbone and SSL encoder ablation results**

| Model | Reconstruction | | | | | Downstream | | | |
|---|---|---|---|---|---|---|---|---|---|
| | WER↓ | UTMOS↑ | Mel L1↓ | SIM↑ | F0Corr↑ | WER↓ | SID Acc↑ | ASV EER↓ | ER Acc↑ |
| Kanade 12.5Hz | 3.5% | 4.10 | 1.27 | 0.96 | 0.84 | 8.1% | 69.6% | 13.7% | 59.1% |
| ConvNeXt | 4.0% | 4.04 | 1.29 | 0.96 | 0.82 | 8.9% | 73.1% | 10.0% | 59.4% |
| HuBERT | 3.7% | 4.09 | 1.26 | 0.96 | 0.82 | 9.2% | 65.7% | 10.5% | 59.5% |

Table 10: **GAN post-training ablation results**

| Model | Reconstruction | | | | | |
|---|---|---|---|---|---|---|
| | WER↓ | MUSHRA↑ | UTMOS↑ | Mel L1↓ | SIM↑ | F0Corr↑ |
| Kanade 12.5Hz | 3.4% | 74.6 | 4.17 | 1.25 | 0.97 | 0.85 |
| w/o GAN | 3.5% | 69.0 | 4.10 | 1.27 | 0.96 | 0.84 |
| Kanade 25Hz | 2.4% | 75.0 | 4.16 | 1.02 | 0.97 | 0.88 |
| w/o GAN | 2.3% | 70.3 | 4.13 | 1.03 | 0.97 | 0.88 |

We also experiment with using mel spectrograms as input for global branch instead of SSL features. This worsened all metrics other than Mel L1. This indicates that SSL features provide more useful and structured information on speaker identity and paralinguistic details, benefiting both reconstruction and downstream performance. This result motivated us to build a tokenizer fully based on SSL features.

Moreover, we study the effect of **pooling and conditioning** in the global branch. Compared to average pooling, our main model with attentive statistical pooling (Okabe et al., 2018) has slightly better intelligibility and prosody. For conditioning mechanism ablation, we train (1) a variant where global embeddings apply adaLN-Zero (Peebles & Xie, 2023) conditioning to both the token module and mel module in our decoder instead of just the latter (noted as Conditioning: full decoder), and (2) a variant using simple addition instead of adaLN-Zero (noted as Conditioning: addition). Both of them exhibit slightly worse intelligibility and prosodic correlation, though the model with addition conditioning achieves remarkable SID accuracy (82.6%). We stick to adaLN-Zero conditioning only mel module, as this seems to better preserve linguistic information.

## C.3 ARCHITECTURE

We train a model with all transformers replaced with ConvNeXt (Liu et al., 2022) backbones with a matching parameter count. The results are in Table 9. The model shows similar results except mildly worse linguistic content metrics (+0.5% reconstruction WER and +0.8% downstream WER). This indicates that the stronger sequence modeling ability of transformers can help the model better preserve and surface linguistic information.

We also try replacing WavLM Base+ with HuBERT-base, which shows similar results in Table 9. This validates the effectiveness of our method across SSL models.

Table 10 shows reconstruction results without GAN post-training. Based on these ablations, post-training slightly improves audio quality (higher MUSHRA, UTMOS and lower Mel L1) without heavily affecting other metrics.

# D ANALYSIS

## D.1 PROSODIC INFORMATION PROBING

Table 11: **Probing results on fundamental frequency (F0)**

| Model | Corr↑ | RMSE↓ |
|---|---|---|
| KM 12.5Hz | 0.50 | 0.86 |
| KM 25Hz | 0.53 | 0.84 |
| DualCodec | 0.78 | 0.62 |
| SpeechTokenizer | 0.57 | 0.82 |
| Mimi | 0.46 | 0.88 |
| FACodec | 0.64 | 0.76 |
| X-Codec 2 | 0.55 | 0.83 |
| StableCodec | 0.35 | 0.93 |
| WavTokenizer | 0.78 | 0.63 |
| BiCodec | 0.50 | 0.86 |
| PAST | 0.54 | 0.83 |
| TiCodec | 0.68 | 0.73 |
| Kanade 12.5Hz | 0.68 | 0.73 |
| Kanade 25Hz | 0.75 | 0.65 |

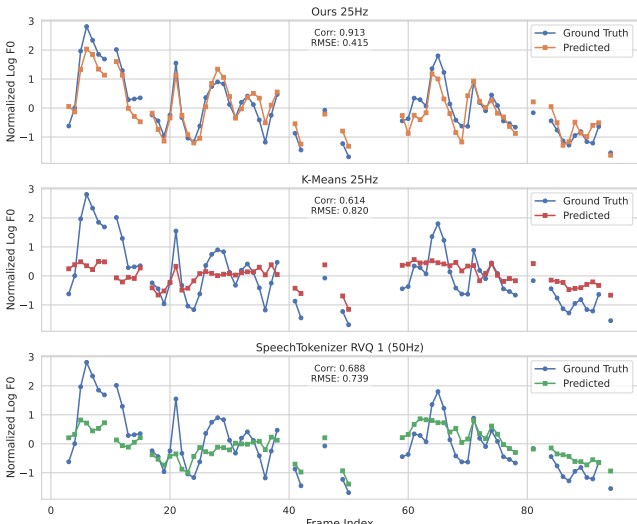

Figure 3: **Comparison of F0 probing predictions**

To measure the availability of prosodic information within speech tokens, we conduct a probing analysis on fundamental frequency (F0), which humans perceive as pitch. We train 7M-parameter 2-layer 512-dim bidirectional transformers with a linear head to predict log F0. The models are optimized with MSE loss for 50k steps, using AdamW (Loshchilov & Hutter, 2019) (learning rate 1e-3, $\beta_1 = 0.9, \beta_2 = 0.999$, weight decay 1e-2). We use LibriSpeech `train-clean-100` for training and `test-clean` for testing. F0 extraction settings match those in our reconstruction experiments (Section 4.4.1). Since our main focus is to investigate the usefulness of different speech tokens for prosody modeling in SLMs, we use tokens from the most linguistically-related RVQ layer (RVQ 1, or the first content layer in FACodec) for multi-layer codecs. The log F0 values are normalized for each instance, as only relative pitch is linguistically relevant. We report Pearson correlation coefficient (Corr) and root mean squared error (RMSE).

Results are shown in Table 11. Kanade models achieve better F0 probing performance than most of the baselines and k-means tokens (Kanade 25Hz 0.75 vs. KM 25Hz 0.53 on Corr). We display probing results for one sample in Figure 3: predictions from our content tokens are more aligned with the ground truth than those from k-means or SpeechTokenizer. These results verify that our tokens make prosody easily accessible.

## D.2 PHONETIC INFORMATION ANALYSIS

Table 12: **Phonetic information metrics**

| Model | ABX↓ | | PNMI↑ |
|---|---|---|---|
| | within | across | |
| KM 12.5Hz | 4.4% | 5.1% | 0.79 |
| KM 25Hz | 3.5% | 4.2% | 0.81 |
| DualCodec | 16.0% | 19.1% | 0.56 |
| ST | 3.6% | 4.5% | 0.69 |
| Mimi | 6.6% | 7.8% | 0.63 |
| FACodec | 4.4% | 5.9% | 0.53 |
| X-Codec 2 | 15.4% | 22.4% | 0.44 |
| StableCodec | 21.9% | 25.0% | 0.55 |
| WavTokenizer | 25.6% | 31.5% | 0.17 |
| BiCodec | 24.5% | 34.3% | 0.22 |
| Kanade 12.5Hz | 22.7% | 24.3% | 0.58 |
| Kanade 25Hz | 19.0% | 21.6% | 0.49 |

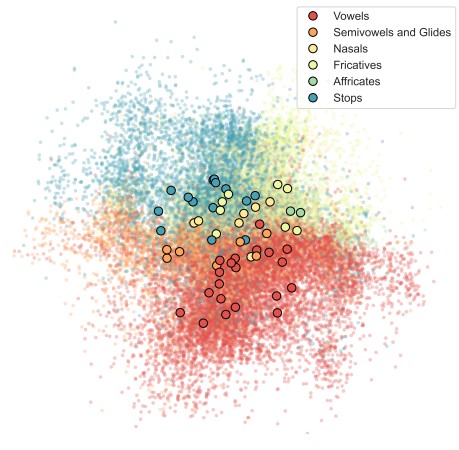

Figure 4: **PCA visualization of our content embedding.** Points are colored by category. Larger markers represent per-phoneme average embeddings.

First, we visualize the distribution of phones in the continuous content embedding space of our 12.5Hz model. We encode the TIMIT dataset (Garofolo et al., 1993) using the content encoder (768-dim) and find the average embedding for each phoneme. We perform Principal Component Analysis (PCA) with two components on these average embeddings, then project all the collected embeddings onto the learned PCA space. The result is shown in Figure 4. We observe a clear phonetic configuration of the embedding space.

To numerically evaluate the phonemic information in our content tokens, we measure **ABX** phoneme discriminability (Schatz et al., 2013) and phone-normalized mutual information (**PNMI**) (Hsu et al., 2021). In the literature on speech representations, the phone/phoneme terminology is not well-respected. We use terms as used in the original definitions of these metrics. Technically, both of them measure phonemic information, but hierarchical clustering shows that SSL representations are mostly phonetic (van Niekerk et al., 2023).

**ABX** measures the extent to which phonemic categories are localized in feature space. It starts with a minimal pair of triphones like "bag" and "beg". The model is presented with $A$, an instance of the first, $B$, an instance of the second, and $X$, another instance of one of the two triphones. A and B always come from the same speaker. X either comes from the same speaker (*within*) or a different speaker (*across*).

We choose a distance measure $d(x, y)$ and calculate both $d(X, A)$ and $d(X, B)$. In a well-configured embedding, $X$ should be closer to the sample from the same class. For example, if $A$ is an instance of "bag", $B$ is instance of "beg", and $X$ is another instance of "bag", then we expect $d(X, A) < d(X, B)$. The ABX score is the error rate: lower ABX scores indicate better phonemic discriminability directly in the representation space. We evaluate ABX on Libri-light (Kahn et al., 2020) test-clean, using the fastabx library (Poli et al., 2025). We use cosine similarity as the distance measure following convention (Dunbar et al., 2021).

**PNMI** calculates the mutual information between phones and tokens $I(\text{phones}; \text{tokens})$, normalized by phone entropy $H(\text{phones})$. It measures the amount of uncertainty about the phone identity that is eliminated by observing the token. Higher PNMI score indicates stronger correspondence between tokens and phones. We evaluate on the TIMIT dataset (Garofolo et al., 1993).

The results are shown in Table 12. K-means tokens achieve the best performance on these metrics, indicating a strong relationship with phonetic categories. SpeechTokenizer and Mimi, which use knowledge distillation, and FACodec, which uses phoneme labels, exhibit comparable performance. DualCodec, X-Codec 2, and Kanade, which use VQ-VAE, perform similarly. Notably, Kanade 12.5Hz is ranked the third on PNMI score among codecs.

In Figure 5, we visualize the relationship between speech tokens and phonemes. PNMI is a measure of the strength of this relationship. Though noisier than k-means tokens, ours show recognizable

correspondence to TIMIT phonemes. Curiously, all tokenizers other than BiCodec, FACodec, and ours have a significant token space that is unrelated to encoding this information.

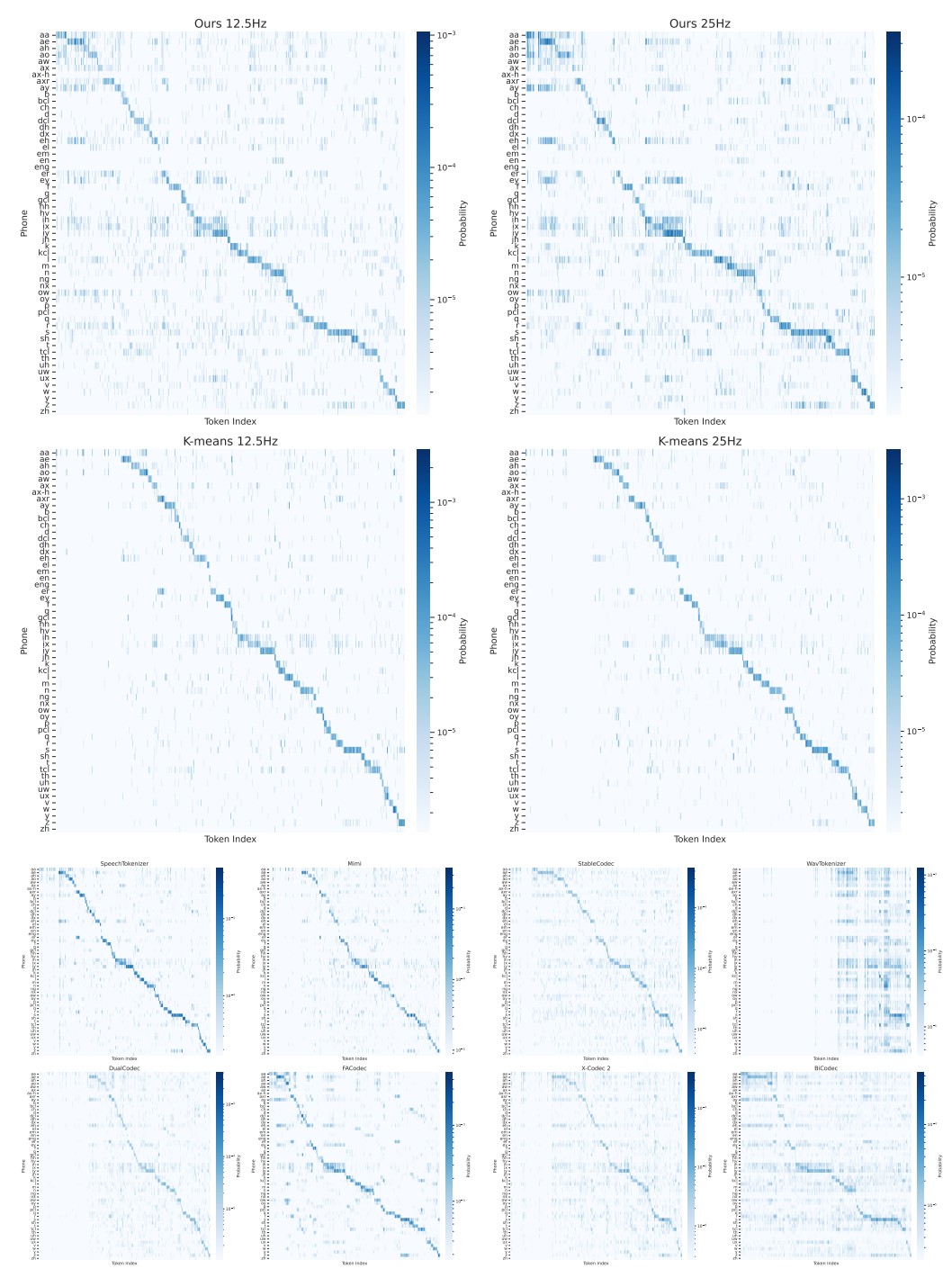

Figure 5: **Joint probability distributions on speech tokens and TIMIT phones.** The token indices are sorted for better visualization.

### D.3 GLOBAL EMBEDDING PCA ANALYSIS

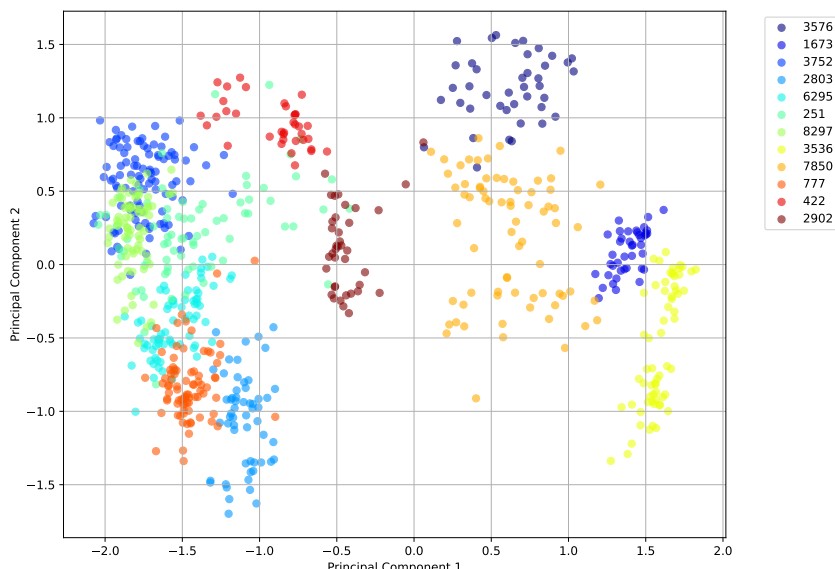

Figure 6: **PCA of global embeddings.** Colored by LibriSpeech speaker ID.

We perform PCA on the global embeddings from LibriSpeech `dev-clean` and plot a subset of the utterances in Figure 6.

To get a sense of what these components represent, we took utterances, tokenized them, and reconstructed them using a perturbed global embedding. Subjectively, the first principal component seems related to speaker gender. The second and third are harder to characterize without further analysis. Samples of these perturbations are available on the demo page[4].

---

[4]https://anonymous-speech-research.github.io/demo2/

## D.4 PRELIMINARY SLM RESULTS

Table 13: **Preliminary SLM results**

| Model | Token rate | Vocab. size | sWUGGY↑ | sBLIMP↑ |
|---|---|---|---|---|
| KM 12.5Hz | 12.5 | 12 800 | 69.8% | 54.0% |
| KM 25Hz | 25 | 12 800 | 66.8% | 53.3% |
| SSL-distilled Codecs | | | | |
| ST | 50 | 1024 | **71.0%** | 52.1% |
| Mimi | 12.5 | 2048 | 68.3% | **53.7%** |
| Other Codecs | | | | |
| DualCodec | 12.5 | 16 384 | 56.5% | 50.2% |
| FACodec | 80 | 1024 | 57.2% | 50.1% |
| X-Codec 2 | 50 | 65 536 | 52.4% | 50.0% |
| StableCodec | 25 | 46 656 | 57.1% | 51.1% |
| WavTokenizer | 40 | 4096 | 52.7% | 50.8% |
| BiCodec | 50 | 8192 | 54.1% | 50.2% |
| Kanade 12.5Hz | 12.5 | 12 800 | 65.6% | 51.8% |
| Kanade 25Hz | 25 | 12 800 | 61.5% | 51.2% |

Before training the SLMs described in main text, we trained weaker SLMs for each tokenizer using the training subset of LibriSpeech. For multi-layer tokenizers, tokens are extracted from the first RVQ layer (for FACodec, the first content layer), as those layers are meant to contain linguistic information for language modeling. Each training sequence is randomly cropped to 20.48 seconds. An autoregressive transformer is trained for 200k steps, with a batch size of 16. We use the last checkpoint for evaluation. Other transformer details are consistent with the descriptions in Appendix E.3.

Results are shown in Table 13. SSL-distilled codecs and k-means, both of which are phonetically dense perform the best. Kanade exposes more suprasegmental information (see Appendix D.1) in its one token stream, which may make learning more difficult, but as shown in the main text, using more powerful models can erase the gap.

### D.5 METRIC CORRELATION ANALYSIS

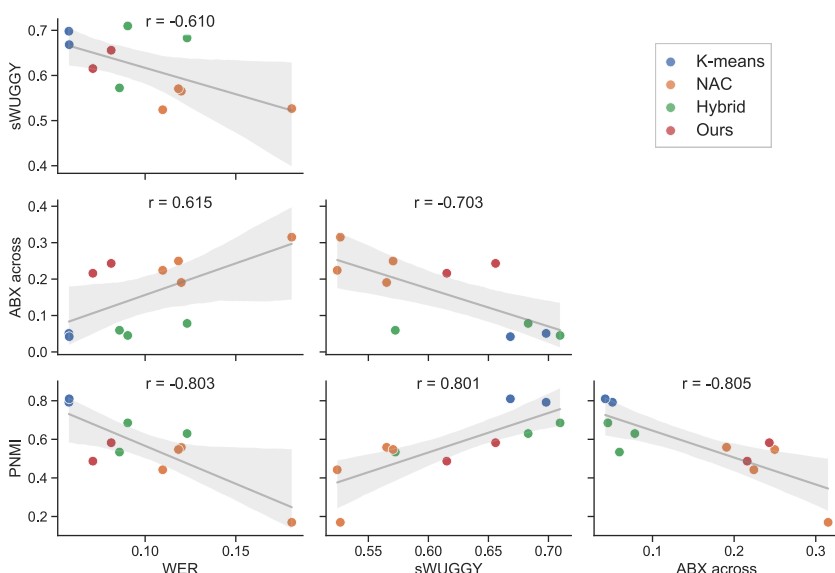

Figure 7: **Correlation among metrics of lexical and phonetic performance.** Lexical metrics include downstream ASR WER (Table 2) and sWUGGY in spoken language modeling (Table 13). Phonetic metrics includes ABX across and PNMI (Table 12). Coarse model groupings are included for readability.

**High phonetic discriminability is not a necessary condition for high lexical availability.** Although Kanade models do not get the best phonetic metrics (as seen in Table 12), they still achieve the SOTA performance on downstream ASR (as seen in Table 2). This observation lead us to further investigate the correlation between different linguistic metrics.

The results are shown in Figure 7, where we observe correlation (also reported by Chang et al. (2024)) between downstream WER and phonetic metrics. However, the relationship is not perfect. Notably, in the ABX-WER plot (second row, first column):

- Our models (red dots) are significantly higher than the regression line, which means they are better at providing lexical information than the models with similar phonetic performance.

- Hybrid codec models (green dots) are significantly lower than the regression line, which means they fail to achieve word error rates typical of models with similar ABX scores (k-means).

The ABX-sWUGGY plot (second row, second column) also shows our models achieve noticeably better sWUGGY scores than NACs, despite having similar ABX scores. Huang et al. (2024) also report that the relationship between phonetic discriminability and downstream performance is not strict: they recorded these scores during training and observed that PNMI scores peaked early then decreased in parallel with downstream WER.

These results suggest that ABX and PNMI, originally designed for acoustic unit discovery, may not be sufficient to measure token quality for downstream modeling. Kanade tokens perform similarly to NAC tokens on these metrics, but perform similarly to k-means tokens on lexical tasks. We hypothesize that Kanade tokens may contain more non-phonetic linguistic information that can help identify words or might have a less well-behaved continuous embedding space. However, without further investigation, we cannot make a decisive conclusion.

## D.6 Out-of-distribution reconstruction

Table 14: **OOD reconstruction results**. Evaluation on various out-of-distribution (OOD) datasets. † indicates models trained on relevant data (e.g., noisy data or Japanese). Includes only the best models from the reconstruction results. F0RMSE is not included for voice conversion results. For all results see Tables 25 and 26.

| Model | Intelligibility | | Quality | | Speaker | | Prosody | |
|---|---|---|---|---|---|---|---|---|
| | WER↓ | CER↓ | UTMOS↑ | Mel L1↓ | SIM↑ | MCD↓ | F0Corr↑ | F0RMSE↓ |
| Gigaspeech (Chen et al., 2021) *(noisy speech)* | | | | | | | | |
| Ground Truth | 9.7 | 5.1 | 2.84 | – | – | – | – | – |
| X-Codec 2† | 11.5 | 6.3 | 2.99 | 0.89 | 0.97 | 5.53 | 0.87 | 0.08 |
| BiCodec† | 11.9 | 6.6 | 3.07 | 1.28 | 0.96 | 6.36 | 0.87 | 0.08 |
| PAST 1:8 | 10.9 | 6.0 | 3.09 | 0.86 | 0.98 | 5.19 | 0.89 | 0.07 |
| DualCodec 1:8† | 11.0 | 6.0 | 3.11 | 0.76 | 0.98 | 4.62 | 0.84 | 0.08 |
| WavTokenizer | 33.9 | 21.9 | 2.64 | 1.19 | 0.88 | 6.97 | 0.82 | 0.10 |
| StableCodec† | 27.1 | 16.3 | 3.51 | 1.65 | 0.90 | 8.70 | 0.84 | 0.09 |
| Kanade 12.5Hz | 16.2 | 9.3 | 3.25 | 1.44 | 0.95 | 7.63 | 0.74 | 0.13 |
| Kanade 25Hz | 11.3 | 6.2 | 3.27 | 1.21 | 0.96 | 6.61 | 0.81 | 0.09 |
| Salmon Sentiment Consistency (Maimon et al., 2025b) *(emotional)* | | | | | | | | |
| Ground Truth | 2.9 | 1.0 | 3.79 | – | – | – | – | – |
| w/ change | 4.9 | 1.6 | 3.62 | – | – | – | – | – |
| X-Codec 2† | 3.8 | 1.2 | 3.77 | 0.84 | 0.97 | 5.40 | 0.85 | 0.09 |
| w/ change | 5.7 | 2.2 | 3.67 | 0.85 | 0.97 | 5.45 | 0.89 | 0.11 |
| BiCodec† | 5.4 | 1.7 | 3.84 | 1.21 | 0.98 | 6.00 | 0.81 | 0.10 |
| w/ change | 6.0 | 2.6 | 3.73 | 1.24 | 0.97 | 6.11 | 0.90 | 0.11 |
| PAST 1:8 | 3.0 | 1.0 | 3.91 | 0.78 | 0.99 | 5.10 | 0.85 | 0.09 |
| w/ change | 4.2 | 1.7 | 3.77 | 0.78 | 0.98 | 5.09 | 0.90 | 0.08 |
| DualCodec 1:8† | 3.6 | 1.1 | 3.91 | 0.71 | 0.98 | 4.45 | 0.88 | 0.08 |
| w/ change | 4.4 | 1.8 | 3.76 | 0.71 | 0.98 | 4.41 | 0.90 | 0.10 |
| WavTokenizer | 14.5 | 7.7 | 3.21 | 1.12 | 0.90 | 6.70 | 0.74 | 0.12 |
| w/ change | 17.5 | 9.7 | 3.13 | 1.13 | 0.90 | 6.93 | 0.82 | 0.16 |
| StableCodec† | 14.8 | 7.2 | 4.08 | 1.45 | 0.93 | 7.87 | 0.81 | 0.12 |
| w/ change | 18.0 | 9.3 | 4.03 | 1.49 | 0.92 | 7.98 | 0.84 | 0.12 |
| Kanade 12.5Hz | 6.4 | 2.3 | 3.83 | 1.38 | 0.95 | 7.72 | 0.66 | 0.19 |
| w/ change | 7.0 | 3.1 | 3.83 | 1.50 | 0.94 | 8.22 | 0.67 | 0.22 |
| Kanade 25Hz | 4.4 | 1.5 | 3.85 | 1.12 | 0.96 | 6.57 | 0.73 | 0.16 |
| w/ change | 4.7 | 1.9 | 3.88 | 1.21 | 0.96 | 6.80 | 0.75 | 0.18 |
| Japanese Versatile Speech (Takamichi et al., 2019) *(unseen language speech)* | | | | | | | | |
| Ground Truth | 4.6 | 2.5 | 3.63 | – | – | – | – | – |
| X-Codec 2† | 5.4 | 2.9 | 3.59 | 0.76 | 0.98 | 5.28 | 0.89 | 0.10 |
| BiCodec | 5.7 | 3.1 | 3.73 | 1.62 | 0.98 | 7.67 | 0.86 | 0.10 |
| PAST 1:8 | 5.2 | 2.8 | 3.62 | 0.84 | 0.98 | 5.98 | 0.88 | 0.09 |
| DualCodec 1:8† | 5.0 | 2.8 | 3.67 | 0.64 | 0.99 | 4.46 | 0.81 | 0.09 |
| WavTokenizer | 18.2 | 11.3 | 2.92 | 1.01 | 0.88 | 6.92 | 0.82 | 0.14 |
| StableCodec | 25.0 | 16.5 | 3.83 | 1.99 | 0.91 | 10.36 | 0.90 | 0.10 |
| Kanade 12.5Hz | 12.2 | 7.2 | 3.77 | 1.30 | 0.94 | 8.15 | 0.70 | 0.21 |
| Kanade 25Hz | 5.6 | 3.0 | 3.72 | 1.03 | 0.97 | 6.55 | 0.84 | 0.17 |
| English Read by Japanese (Nakagawa, 2007) *(accented speech)* | | | | | | | | |
| Ground Truth | 14.9 | 8.0 | 3.73 | – | – | – | – | – |
| X-Codec 2† | 20.7 | 11.3 | 3.69 | 0.78 | 0.97 | 5.16 | 0.86 | 0.08 |
| BiCodec | 21.4 | 11.7 | 3.76 | 2.25 | 0.97 | 8.75 | 0.86 | 0.07 |
| PAST 1:8 | 25.3 | 14.1 | 3.65 | 0.90 | 0.97 | 5.40 | 0.85 | 0.07 |
| DualCodec 1:8† | 17.1 | 9.4 | 3.71 | 0.66 | 0.98 | 4.27 | 0.86 | 0.07 |
| WavTokenizer | 51.7 | 31.6 | 3.06 | 1.06 | 0.91 | 6.45 | 0.82 | 0.08 |
| StableCodec | 51.4 | 29.3 | 4.03 | 2.52 | 0.91 | 10.76 | 0.87 | 0.06 |
| Kanade 12.5Hz | 33.8 | 18.6 | 3.78 | 1.28 | 0.95 | 6.80 | 0.80 | 0.09 |
| Kanade 25Hz | 22.9 | 12.3 | 3.75 | 1.05 | 0.96 | 5.85 | 0.86 | 0.07 |

We reconstructed randomly sampled utterances from out-of-distribution datasets. Objective metrics are shown in Table 14. We also included the best baselines from Table 1. In all datasets, we listened to Kanade samples with the poorest reconstruction quality. We found phone substitution errors to be common. Phone deletion also occurred with some frequency.

We tested noisy speech by sampling utterances with at least two words from Gigaspeech (Chen et al., 2021). Transcripts were preprocessed to remove punctuation and other tags before computing WER. Listening to the reconstructions, we found that background music and noise was partially captured

by the global embedding, as expected. Even though Kanade has only seen read English speech, it maintains some of the best WERs in this condition.

We tested emotional speech using the sentiment consistency subset of Salmon (Maimon et al., 2025b). Whispered samples were excluded. Samples are originally from Expresso (Nguyen et al., 2023). Each track has a consistent version (only one speech style / emotion) and an inconsistent version (speech style / emotion changes within the utterance). This dataset was chosen to test how Kanade encodes large changes in speaking style across and within utterances, which is not seen in the read English audio it was trained on. We report results for each version separately (*w/ change* indicates results for the inconsistent track). Subjectively, reconstructions of consistent samples were good. Inconsistent samples had some leakage of style into the global embedding, causing them to become more uniform upon resynthesis. Nearly all metrics are degraded in the inconsistent case. Interestingly, even speech tokenizers without disentanglement also suffered under this condition.

We also tested on Japanese, which was not seen during training. Transcripts and ASR results were normalized to phonological script before comparison. While the 25Hz variant is quite good (22% relative increase in WER, in line with results on English speech), the 12.5Hz variant performs poorly (165% relative increase in WER). Subjectively, it sounds slightly accented. Interestingly, the only other tokenizer that was not trained on Japanese data but did realatively well is BiCodec, which has a similar design to Kanade.

Finally, we reconstructed Japanese-accented English speech using sentence samples from ERJ (Nakagawa, 2007). Since segmentals in this dataset are not always clearly in an English phonetic category, we suspect that our discretization step may incorrectly categorize them and eliminate the ambiguity that would normally allow an ASR model to recover using its language modeling capabilities. No speech tokenizer did well on these utterances.

These experiments show that Kanade performs competitively in various scenarios despite being trained on very little data. The consistency experiment shows that large changes in vocal quality do not have a large detrimental effect on intelligibility.

These reconstructions can be found on our demo page[5].

## D.7 Out-of-distribution ASR

To validate that our results generalize to domains other than read speech, we trained ASR models on all tokenizers on the Switchboard (Godfrey & Holliman, 1993) corpus of telephone conversations. Results in Table 15 show similar relative rankings to in-domain speech.

Table 15: **OOD ASR results on Switchboard** (Godfrey & Holliman, 1993)

| Model | WER↓ | CER↓ |
|---|---|---|
| KM 12.5Hz | 17.3% | 11.2% |
| KM 25Hz | 15.0% | 9.5% |
| DualCodec | 28.2% | 18.1% |
| ST | 29.4% | 19.1% |
| Mimi | 30.6% | 20.1% |
| FACodec | 25.5% | 16.4% |
| X-Codec 2 | 103.3% | 75.6% |
| StableCodec | 45.0% | 30.2% |
| WavTokenizer | 67.2% | 46.8% |
| BiCodec | 108.8% | 78.4% |
| PAST | 28.9% | 18.8% |
| TiCodec | 29.1% | 18.9% |
| Kanade 12.5Hz | 24.6% | 15.9% |
| Kanade 25Hz | **18.6%** | **11.7%** |

---

[5]https://anonymous-speech-research.github.io/demo2/

D.8 LENGTH GENERALIZATION

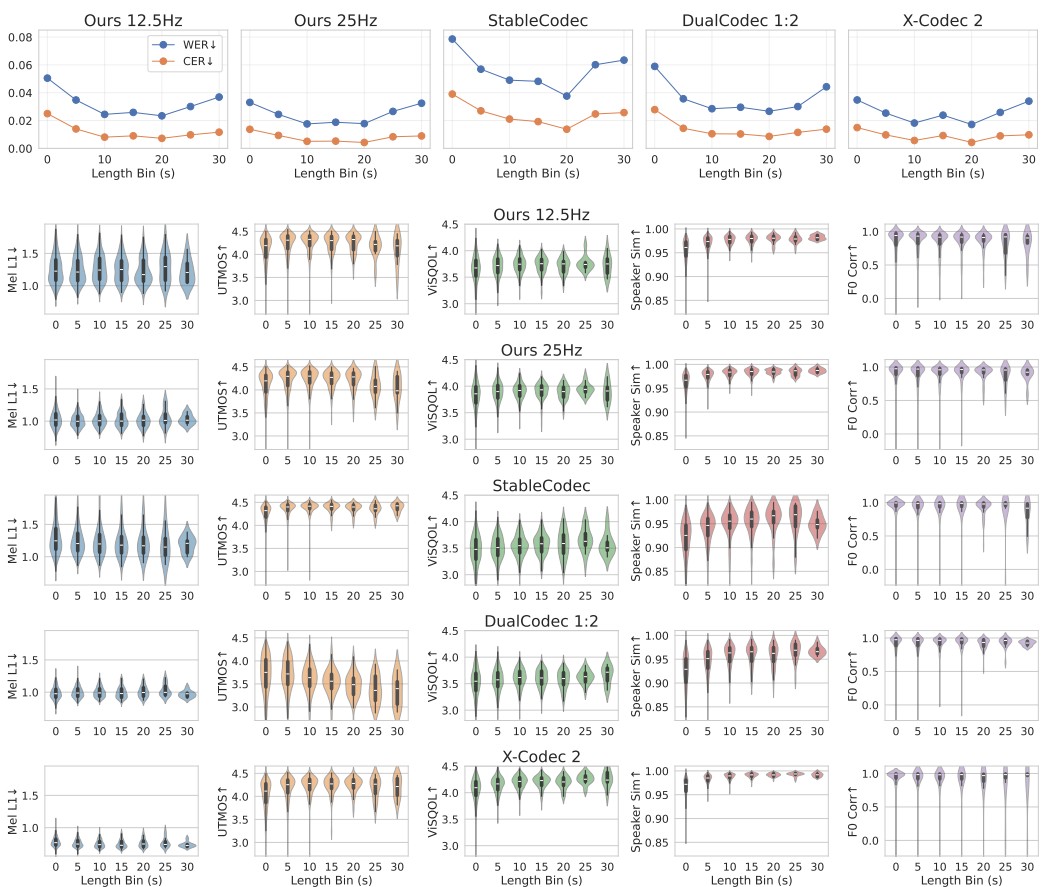

Figure 8: **Reconstruction metrics on different audio length bins**

A good speech tokenizer should work on audio that is longer than the sequences it was trained on. We test the length generalization performance of several high-performing baselines and Kanade on LibriSpeech `test-clean` using reconstruction metrics, binned by audio length. The bin width is 5 seconds, with the final bin including all samples more than 30 seconds long. The results are shown in Figure 8. Most models perform well, though audio quality in DualCodec degrades as the length increases. Kanade models (trained on 5.76s segments) show consistent performance on every metric even at 6x the audio length, indicating excellent length generalization.

### D.9 Chunked encoding and decoding

#### D.9.1 Chunked resynthesis

To show that it is possible to encode audio of arbitrary length with Kanade, we report metrics for a chunked resynthesis experiment. We randomly select 100 samples (25-35 minutes per sample) from Libri-Light (Kahn et al., 2020). Each is encoded into 5.76-second segments that overlap for 1.44 seconds. We decode with either (1) simple mean over all global embeddings of the chunks, or (2) an exponential moving average ($\alpha = 0.8$). We then combine the tracks using a 10ms crossfade.

To evaluate, we select 1000 randomly-selected LibriHeavy (Kang et al., 2024) segments that occur within the 100 LibriLight samples. The corresponding segments are cut out from the resynthesized audio and used to compute the metrics in Table 16.

Table 16: **Chunked Resynthesis Results**. Includes resynthesis using the average global embedding for all chunks or an exponential moving average ($\alpha = 0.8$)

| Model | Intelligibility | | Quality | | Speaker | | Prosody | |
|---|---|---|---|---|---|---|---|---|
| | WER↓ | CER↓ | UTMOS↑ | Mel L1↓ | SIM↑ | MCD↓ | F0Corr↑ | F0RMSE↓ |
| Ground Truth | 4.0 | 1.7 | 3.89 | – | – | – | – | – |
| Kanade 12.5Hz | | | | | | | | |
|   Simple mean | 5.7 | 2.6 | 3.98 | 1.68 | 0.97 | 8.57 | 0.75 | 0.13 |
|   EMA | 5.5 | 2.5 | 3.97 | 1.56 | 0.98 | 8.21 | 0.84 | 0.09 |
| Kanade 25Hz | | | | | | | | |
|   Simple mean | 4.7 | 2.2 | 4.00 | 1.64 | 0.98 | 8.31 | 0.78 | 0.12 |
|   EMA | 4.6 | 2.1 | 3.99 | 1.51 | 0.98 | 7.97 | 0.85 | 0.08 |

This experiment shows that a single global embedding (the simple mean) is enough to encode large amounts of audio with high fidelity.

#### D.9.2 Chunked streaming for SLMs

For an interactive speech language model, it is not necessary to compute any global embeddings as they are not used as input and the output speech is synthesized using a constant global embedding.

**Rough latency estimate** As Table E.2 shows, Kanade is extremely fast. Therefore, input latency is dominated by the amount of padding necessary on the end of the audio input to get a reasonable representation. According to work by Meng et al. (2025), we can estimate that we need a 400ms lookahead and 2 seconds of history to get SSL features that are reasonably accurate (and in turn, good tokens). For synthesis of the SLM output, it is not clear how much lookahead is necessary, but we conservatively estimate that it is the same as the input, 400ms. Therefore, the minimum theoretical latency is 800ms plus Kanade encoding time (2.4s times Kanade's encoding-decoding RTF of 0.0011 is 3ms) and SLM latency. A streaming variant would decrease the necessary lookahead and decrease latency substantially.

## D.10 CODEBOOK UTILIZATION

Table 17: **Normalized entropy of different speech tokenizers**

| DualCodec | ST | Mimi | FACodec | X-Codec 2 | StableCodec | WavTokenizer | BiCodec | Kanade 12.5Hz | Kanade 25Hz |
|---|---|---|---|---|---|---|---|---|---|
| 0.923 | 0.984 | 0.914 | 0.953 | 0.965 | 0.962 | 0.885 | 0.995 | 0.976 | 0.974 |

To test the codebook utilization of baselines and Kanade, we calculate normalized entropy as:

$$\text{Normalized Entropy} = -\frac{1}{\log N} \sum_{x=1}^{N} p(x) \log(p(x)),$$

where $N$ is the codebook size and $p(x)$ denotes the probability distribution of extracted codes at codebook index $x$. Values are between 0 and 1. Higher values indicates better codebook utilization. Note that for codebook-free models such as StableCodec, X-Codec 2 and Kanade, the codebook here refers to the effective codebook produced by FSQ indices. We estimate this using the tokens extracted from each tokenizer on LibriSpeech test-clean. The results are shown in Table 17. Nearly every tested model has good codebook utilization. The normalized entropy values of Kanade content tokens are over 97%, indicating excellent coding efficiency.

# E EXPERIMENT DETAILS

## E.1 MODEL AND TRAINING

**Model Details** The content encoder and feature decoder are 6-layer, 12-head, 768-dim LLaMA (Grattafiori et al., 2024)-style transformers with rotary position embeddings (RoPE) (Su et al., 2024), 2048-dim SwiGLU (Shazeer, 2020) feed-forward networks, and local attention (window size 125).

The FSQ (Mentzer et al., 2024)[6] module uses 5 dimensions with levels of $[8, 8, 8, 5, 5]$, equivalent to a codebook of 12,800 tokens. This results in bitrates of 171bps and 341bps for the 12.5Hz and 25Hz models, respectively.

The global branch uses a 4-layer, 384-dim ConvNeXt (Liu et al., 2022) encoder with attentive statistics pooling (Okabe et al., 2018)[7] to produce a 128-dim global embedding.

The token module is a 6-layer, 12-head, 768-dim transformer (window size 31/65 for 12.5/25Hz model), and the mel module is a 6-layer, 8-head, 512-dim transformer (window size 65) with adaLN-Zero (Peebles & Xie, 2023) conditioning. The post-net consists of 5 convolutional layers with a kernel size of 7 and 256 channels. The Mel spectrograms use 100 bins, 1024-point FFT, and 256 hop length, consistent with Vocos (Siuzdak, 2024).

The discriminator used in post-training is a multi-band spectrogram discriminator directly applied on our generated mel spectrogram, adapted from DAC (Kumar et al., 2023). It splits the mel bins into 5 bands, and processed each band using 5 convolution layers with kernel size of $[3, 3]$ and 64 channels. For a higher-level overview, see Section 3.2.2.

The resulting 12.5Hz model has 120M training parameters and 207M total parameters (containing 73M from WavLM Base+ and 13.5M from Vocos). The 25Hz variant has 118M training paramters and 205M total parameters.

**Training Details** We train the models for 150k steps with a batch size of 128 using randomly chunked 5.76-second audio segments. The SSL feature and mel-spectrogram reconstruction losses are weighted equally ($\alpha = 1$). We optimize with AdamW (Loshchilov & Hutter, 2019) ($\beta_1 = 0.9, \beta_2 = 0.99$, weight decay 1e-4) and a cosine learning rate schedule with a peak of 2e-4 and a 10% warmup.

In the GAN post-training phase, the weights for adversarial loss and feature matching loss are $\beta = 1/30$ and $\gamma = 1/3$, respectively. We use a constant learning rate of 4e-5 and select the final checkpoint based on validation Mel L1 loss and subjective quality.

All models are trained with `bfloat16` mixed precision and FlashAttention 2 for efficiency. Training takes approximately 32 hours on one NVIDIA 5090 GPU in total.

## E.2 TRAINING BUDGET AND INFERENCE EFFICIENCY

We estimate each model's training FLOPs by the reported training steps, batch size, and sample length combined with testing using official checkpoints, so they may be slightly inaccurate. Some values are unavailable due to undisclosed usage or failure to obtain good measurements.

One benefit of using SSL features to train a speech tokenizer is efficiency of data and computation. We use much less data (0.6k vs. X-Codec 2's 150k hours) and computation (6.1 vs. Mimi's 67.3 exaFLOPs) than comparable models. Kanade models are relatively lightweight, with one fifth the

---

[6]FSQ typically works in a very low-dimensional space, and partitions it using a simple fixed grid. To perform FSQ on a vector, we first project it into that lower-dimensional space. Then for each dimension we 1) squash it using a scaled $\tanh$ such it lies in a bounded range $(a, b)$ of the reals; and 2) round it to the nearest integer. There are a finite number of integers between $a$ and $b$ and these correspond to our quantization levels. Since the scaling factor of the squashing function can be chosen, we can freely chose the number of levels for each dimension.

[7]Attentive stats pooling passes the input with $d$ features to a simple convolutional network to weight each element of the input. The mean and standard deviation are then computed over the time dimension for each feature, producing one vector of dimension $2d$ for the entire sequence. The result is passed through a linear layer to obtain the final dimension of the global embedding and then layer normalized.

Table 18: **Training budget and inference efficiency.** We measure the number of FLOPs for forward passes only, as the actual training procedures of different models vary significantly. Real time factor (RTF) indicates ratios of processing time for either encoding (En) or decoding (De) to input audio length, measured on a NVIDIA A6000. Relative efficiency is calculated on full passes.

| Model | Params | Dataset | Sample Rate | Data Size (hours) | FLOPs ($\times 10^{18}$) | RTF (En) ↓ | RTF (De) ↓ | Relative Efficiency ↑ |
|---|---|---|---|---|---|---|---|---|
| Mimi | 79M | – | 24 kHz | – | 67.3 | 0.0007 | 0.0006 | 0.77x |
| DualCodec | 84M | Emilia | 24 kHz | 100k | 27.4 | 0.0078 | 0.0011 | 0.11x |
| StableCodec | 953M | Libri-Light + MLS | 16 kHz | 105k | 24.5 | 0.0028 | 0.0028 | 0.18x |
| FACodec | 102M | Libri-Light | 16 kHz | 60k | 6.8 | 0.0035 | 0.0067 | 0.10x |
| ST | 104M | LibriSpeech | 16 kHz | 1k | 1.0 | 0.0010 | 0.0008 | 0.59x |
| X-Codec 2 | 823M | Emilia + MLS | 16 kHz | 150k | – | 0.0181 | 0.0016 | 0.05x |
| BiCodec | 156M | LibriSpeech + Emilia | 16 kHz | 3k | – | 0.0045 | 0.0030 | 0.13x |
| WavTokenizer | 81M | LibriTTS | 24 kHz | 0.6k | 6.7 | 0.0003 | 0.0003 | 1.67x |
| TiCodec | 63M | LibriTTS | 24 kHz | 0.6k | 0.6 | 0.0021 | 0.0028 | 0.21x |
| PAST | 184M | LibriSpeech + TIMIT | 16 kHz | 1.0k | 1.5 | 0.0012 | 0.0007 | 0.53x |
| Kanade 12.5Hz | 207M | LibriTTS | 24 kHz | 0.6k | 5.9 | 0.0009 | 0.0002 | 1.00x |
| Kanade 25Hz | 205M | LibriTTS | 24 kHz | 0.6k | 6.1 | 0.0009 | 0.0002 | 1.00x |

parameters of StableCodec, but the 25Hz edition still obtains a similar MUSHRA subjective quality score (73.57 vs. StableCodec's 73.81). Inference speed is also excellent, surpassing most baselines.

### E.3    DOWNSTREAM MODEL CONFIGURATIONS

Our transformer-based downstream models all share similar backbones to the ones used in our main model. They are 12-layer, 12-head, 768-dim LLaMA-style transformers with 85M parameters (excluding embedding and output projection layers). Downstream transformers are configured as decoder-only with causal attention. We use AdamW (Loshchilov & Hutter, 2019) ($\beta_1 = 0.9, \beta_2 = 0.999$, weight decay 1e-3) and a cosine learning rate schedule with a peak of 2e-4 and a 10% warmup.

**ASR**  Before training, we train a SentencePiece (Kudo & Richardson, 2018) text tokenizer on LibriSpeech transcripts with a vocabulary size of 5,000. The transformer model is trained for 100k steps on the LibriTTS training sets, with each batch of tokens extracted from 240 seconds of speech. The training sequences are in the format `<speech><BOS><text><EOS>`. Cross-entropy loss is calculated on text tokens only. To use both of the two content token layers of FACodec, we use two embedding layers, concatenate the resulting embeddings along the feature dimension, then project them back to the original dimension via a linear layer. For the continuous reference model, we use the average of layer 6 and 9 features as input. We use label smoothing of 0.1. After training, we select the best checkpoint with the lowest validation loss to test the final WER. During testing, we set beam size as 8, length penalty to 1.0, and patience factor to 2.0.

**TTS**  Before training, we run grapheme-to-phoneme on LibriTTS transcripts to get all phonemes using SoundChoice (Ploujnikov & Ravanelli, 2022). A transformer is trained on the LibriTTS training sets for 200k steps, each step with a batch of tokens extracted from 120 seconds of speech. The sequence format is `<speaker embedding><phonemes><BOS><speech><EOS>`, where the cross-entropy loss during training is calculated on speech tokens only. For RVQ models, we combine all code indices of the used RVQ codebooks to create the token vocabulary. For example, if a tokenizer uses two 1024-code codebooks, then the first codebook has indices $[0, 1023]$, and the second has indices $[1024, 2047]$, forming a final vocabulary of size 2048. We use the last checkpoint for evaluation. During inference, we set the temperature to 1.0 and use top-p sampling with $p = 0.9$. We omit FACodec as a baseline here as its token rate is too high (240Hz for 3 layers).

**Speaker and emotion discriminators**  For speaker and emotion related tasks, we use ECAPA-TDNN (Desplanques et al., 2020) backbones. Following RawNet3 (Jung et al., 2022), the token embedding dimension, hidden dimension, and final embedding dimension are 192, 1024, and 192, respectively. We use AdamW ($\beta_1 = 0.9, \beta_2 = 0.999$, weight decay 1e-3) with a constant learning rate 3e-4. We select the best checkpoint with the lowest validation loss for evaluation. For the speaker model, we train for 50k steps; the scale and margin in AAM-Softmax loss (Deng et al., 2019) are set to 30 and 0.3, respectively; the batch size is 64 and samples are randomly cropped to

3 seconds. For the emotion model, we train for 25k steps. The batch size is 32 and we use label smoothing of 0.3 to mitigate overfitting.

For RVQ models, the token embeddings from different token layers are concatenated and projected back to the original dimension via a linear layer. For models with a global embedding, they are projected to the token embedding dimension and added to the token embeddings at each time step. For BiCodec, which produces 32 global tokens, we allocate individual embedding layers for each token index and aggregate those embeddings. For the continuous reference model, we use the average of features from layers 1, 2, 6 and 9, which are used as the inputs of our main model.

As introduced in Section 4.4.2, we also evaluate these tasks only on our global embedding. To do this, we replace the ECAPA-TDNN with a 3-layer MLP. The hidden dimension is 768.

### E.4 SUBJECTIVE LISTENING TEST

We conduct Multiple Stimuli with Hidden Reference and Anchor (MUSHRA) subjective listening tests using webMUSHRA (Schoeffler et al., 2018).

For the reconstruction quality scores, we ask the subjects to judge *"unnatural or robotic-sounding speech; muffled or distorted sound; the rhythm and melody of the voice sounding unnatural; the speaker's voice sounding different; and incorrect words or slurred pronunciation"*. The ground truth is shown as reference. 10 audio samples of between 3–6 seconds are randomly selected from LibriSpeech `test-clean`.

For TTS, we prepare for two different tests. In the speech quality test, we ask the subjects to judge *"robotic artifacts, static noise, muffled sound; slurred pronunciation or unclear speech"* and ignore *"the speaker's emotion, rhythm, speed, pitch, or intonation"*. In the prosody naturalness test, we ask the subjects to judge *"the melody of the voice (intonation), correct stress on words, natural speed, and logical pauses (rhythm)"* and ignore *"audio quality issues such as static, robotic buzzing, or muffled sounds"*. There is no reference shown in the TTS tests. 10 audio samples are randomly selected from the LibriTTS `test-clean` subset.

For the VC speaker similarity test, we ask the subjects to judge *"if the sample sounds exactly like the same person as the reference"*. The reference speech from the target speaker is shown as reference. 10 audio samples are randomly selected from the VCTK subset.

For all tests, the ground truth is included as a hidden condition. Each sample is scored by at least 25 people. Since it is difficult for participants to score many models at once, we divide the models into groups with roughly balanced quality composition based on objective metrics. We removed outlier participants from the collected data and calibrated the groups by the mean reference scores among the groups. Lowpass-filtered anchors are not used.

We use bootstrapping (Mendonça & Delikaris-Manias, 2018) with 1000 iterations to estimate the median scores for each models and report 95% confidence intervals (see Section E.6).

### E.5 BASELINES

**SpeechTokenizer** (Zhang et al., 2024) A hybrid speech codec that distills HuBERT (Hsu et al., 2021) features into the first of 8 RVQ layers. By doing this, SpeechTokenizer makes its first layer more like HuBERT features, making them a suitable alternative to SSL k-means tokens for spoken language modeling. The rest of the token layers encode the rest of the information necessary for reconstruction. It is one of the earliest hybrid speech codecs. The token rate per layer is 50Hz and the codebook size is 1024. We use the `hubert_avg` checkpoint. SpeechTokenizer and other RVQ-based models introduced below support variable bitrates by using only the first $N$ token layers, thanks to random quantizer dropout training. [8]

**Mimi** (Défossez et al., 2024) A streaming hybrid speech codec that distills WavLM (Chen et al., 2022a) features into the first quantization layer, similar to SpeechTokenizer. The difference is Mimi uses a separate VQ layer for distillation alongside 7 normal RVQ layers. The token rate per layer is 12.5Hz and the codebook size is 2048. [9]

---

[8] `https://github.com/ZhangXInFD/SpeechTokenizer`

[9] `https://huggingface.co/kyutai/mimi`

**DualCodec** (Li et al., 2025) A hybrid speech codec that incorporates SSL features by compressing w2v-BERT 2.0 (Barrault et al., 2023) features with a ConvNeXt (Liu et al., 2022)-based VQ-VAE and using the quantized latents as RVQ 1. A separate encoder is applied to the waveform to produce an acoustic embedding. RVQ 1 is decoded and subtracted from the acoustic embedding. The remaining 7 RVQ layers quantize the residual. The token rate per layer is 12.5Hz and the codebook size is 16,384 for the first layer and 4,096 for the rest. [10]

**FACodec** (Ju et al., 2024) A hybrid speech codec that explicitly disentangles prosody, phonetic content, and speaker identity using supervision and gradient reversal layers. It produces 6 RVQ layers: 1 for prosody (supervised by F0), 2 for phonetic content (supervised using phonemes sequences) and 3 for residual details. It also produces a global speaker embedding learned by speaker supervision. Unlike codecs that distill from SSL features, FACodec enhances phonetic information in content tokens via direct supervision. The token rate per layer is 80Hz and the codebook size is 1024. Considering the high bitrate, all evaluations omit the 3 residual layers. [11]

**StableCodec** (Parker et al., 2024) A large transformer-based single-layer neural audio codec that uses a novel post-hoc residual formulation of FSQ (Mentzer et al., 2024). They show transformers' great scalability in speech coding and reach very low a bitrate of 400bps. It represents one of the earliest speech codecs with a transformer-based architecture. In their official repository, the authors further fine-tune the model using CTC loss on phonemes to enhance lexical information. Following their recommendation, we use this fine-tuned checkpoint `stable-codec-speech-16k`. The token rate is 25Hz and the codebook size is 46656. [12]

**WavTokenizer** (Ji et al., 2024) A single-layer neural audio codec uses several techniques to improve codebook utilization, such as k-means initialization and dead code random restart. It also uses a ConvNeXt (Liu et al., 2022) backbone and predicts Short-Time Fourier Transform magnitude and phase values instead of waveform. The token rate is 40Hz and the codebook size is 4096. We use the speech-only checkpoint `small-600-24k-4096`. [13]

**X-Codec 2** (Ye et al., 2025b) A single-layer neural audio codec that adds a parallel VQ-VAE for w2v-BERT 2.0 (Barrault et al., 2023) feature reconstruction alongside the original acoustic VQ-VAE. Frozen SSL and acoustic features are projected and concatenated into a shared space that is quantized using FSQ (Mentzer et al., 2024). The token rate is 50Hz and the codebook size is 65536. [14]

**BiCodec** (Wang et al., 2025a) A single-layer neural audio codec that uses wav2vec 2.0 (Baevski et al., 2020) features as main input and extracts global tokens from mel spectrogram to represent constant acoustic characteristics such as speaker timbre. It uses cross attention mechanism similar to Q-former on ECAPA-TDNN features to extract fixed-length global tokens, which are then quantized by FSQ. The decoder reconstructs both the waveform and SSL features. The token rate is 50Hz and the codebook size is 8192. [15]

We don't include the earlier codecs such as EnCodec (Défossez et al., 2023) and DAC (Kumar et al., 2023) because (1) they mainly focus on high-quality general audio coding, while we focus on speech-only tokenizers that have potential for speech language modeling; (2) they need more tokens to reconstruct good quality audio, with the lowest bitrates starting from 1.5kbps, which is impractical for speech LMs; and (3) their approaches are already well represented and improved on in later works such as SpeechTokenizer, Mimi and DualCodec.

---

[10] https://github.com/jiaqili3/dualcodec
[11] https://github.com/open-mmlab/Amphion/tree/main/models/codec/ns3_codec
[12] https://github.com/Stability-AI/stable-codec
[13] https://github.com/jishengpeng/WavTokenizer
[14] https://huggingface.co/HKUSTAudio/xcodec2
[15] https://github.com/SparkAudio/Spark-TTS

## E.6 Full Results

Table 19: **Full Speech reconstruction results.** Grouped by model family. Bold numbers indicate the best performance in that column.

| Model | Bitrate | Token Rate | Intelligibility | | Quality | | | | Speaker | | Prosody | |
|---|---|---|---|---|---|---|---|---|---|---|---|---|
| | | | WER↓ | CER↓ | MUSHRA↑ | UTMOS↑ | ViSQOL↑ | Mel L1↓ | SIM↑ | MCD↓ | F0Corr↑ | F0RMSE↓ |
| Ground Truth | – | – | 1.9 | 0.6 | 78.0 | 4.07 | 5.00 | – | – | – | – | – |
| Cont. 50Hz | – | – | 2.0 | 0.6 | 76.7 | 3.90 | 4.54 | 0.74 | 0.99 | 3.91 | 0.94 | 0.04 |
| KM 12.5Hz | 171 | 12.5 | 3.0 | 1.1 | 72.1 | 4.04 | 3.33 | 1.44 | 0.96 | 7.45 | 0.66 | 0.15 |
| KM 25Hz | 341 | 25 | 2.7 | 1.0 | 72.4 | 4.07 | 3.40 | 1.30 | 0.96 | 6.76 | 0.67 | 0.15 |
| FACodec* 1:6 | 4800 | 240 | **2.1** | **0.7** | 81.4 | 4.11 | 4.27 | 0.76 | 0.98 | 5.17 | 0.94 | **0.04** |
| FACodec* 1:3 | 2400 | 240 | 2.4 | 0.8 | – | 3.62 | 3.85 | 1.02 | 0.97 | 6.05 | 0.85 | 0.08 |
| PAST 1:8 | 4000 | 50 | **2.1** | **0.7** | 82.4 | 4.18 | **4.32** | 0.72 | **0.99** | 4.42 | 0.92 | **0.04** |
| PAST 1:4 | 2000 | 50 | 2.4 | 0.9 | – | 3.88 | 4.07 | 0.85 | 0.98 | 5.07 | 0.89 | 0.06 |
| PAST 1:2 | 1000 | 50 | 3.1 | 1.2 | – | 2.45 | 3.17 | 1.27 | 0.88 | 6.88 | 0.39 | 0.31 |
| ST 1:8 | 4000 | 50 | **2.1** | **0.7** | 76.0 | 3.90 | 4.26 | 0.72 | 0.98 | 4.72 | 0.92 | 0.05 |
| ST 1:4 | 2000 | 50 | 2.6 | 0.9 | 74.2 | 3.56 | 3.86 | 0.90 | 0.96 | 5.66 | 0.88 | 0.07 |
| ST 1:2 | 1000 | 100 | 3.6 | 1.4 | – | 2.28 | 3.15 | 1.25 | 0.90 | 7.29 | 0.78 | 0.11 |
| TiCodec* 1:4 | 3000 | 75 | 2.3 | 0.8 | – | 3.60 | 4.11 | 0.82 | 0.97 | 7.43 | 0.91 | 0.05 |
| TiCodec* 1:2 | 1500 | 75 | 3.7 | 1.6 | – | 3.43 | 3.77 | 0.97 | 0.94 | 7.99 | 0.88 | 0.07 |
| TiCodec* 1:1 | 750 | 75 | 9.3 | 4.8 | – | 3.17 | 3.44 | 1.09 | 0.91 | 6.55 | 0.85 | 0.08 |
| Mimi 1:8 | 1100 | 50 | 3.7 | 1.9 | – | 3.56 | 3.87 | 1.18 | 0.97 | 6.30 | 0.93 | 0.05 |
| Mimi 1:4 | 550 | 50 | 7.7 | 5.1 | – | 3.02 | 3.47 | 1.41 | 0.93 | 7.45 | 0.87 | 0.09 |
| Mimi 1:2 | 275 | 50 | 14.7 | 10.8 | – | 2.39 | 2.88 | 1.82 | 0.86 | 9.39 | 0.60 | 0.17 |
| DualCodec 1:8 | 925 | 50 | **2.1** | **0.7** | 75.6 | 4.12 | 4.28 | **0.66** | 0.98 | **4.08** | **0.95** | **0.04** |
| DualCodec 1:4 | 625 | 50 | 2.6 | 0.9 | – | 4.07 | 3.97 | 0.79 | 0.97 | 4.95 | 0.93 | 0.05 |
| DualCodec 1:2 | 325 | 25 | 3.7 | 1.5 | 72.4 | 3.67 | 3.56 | 0.99 | 0.94 | 6.11 | 0.91 | 0.07 |
| X-Codec 2 | 800 | 50 | 2.5 | 0.9 | 77.0 | 4.13 | 4.12 | 0.77 | 0.98 | 4.92 | 0.90 | 0.06 |
| BiCodec* | 650 | 50 | 2.5 | 0.9 | 75.0 | 4.18 | 4.09 | 0.94 | 0.98 | 5.22 | 0.91 | 0.05 |
| WavTokenizer | 480 | 40 | 9.4 | 4.7 | 72.1 | 3.57 | 3.55 | 1.00 | 0.92 | 6.17 | 0.91 | 0.07 |
| StableCodec | 388 | 25 | 5.7 | 2.6 | 79.3 | **4.31** | 3.50 | 1.28 | 0.93 | 7.29 | 0.91 | 0.05 |
| Kanade* 25Hz | 341 | 25 | 2.4 | 0.8 | 75.0 | 4.16 | 3.86 | 1.02 | 0.97 | 5.67 | 0.88 | 0.07 |
| Kanade* 12.5Hz | 171 | 12.5 | 3.3 | 1.3 | 74.6 | 4.17 | 3.69 | 1.25 | 0.97 | 6.82 | 0.85 | 0.10 |

Models marked with * also use a fixed-size representation for reconstruction. FACodec: 8192 bits (256-dim `fp32`), TiCodec: 80 bits (8 tokens), BiCodec: 384 bits (32 tokens), and Kanade: 4096 bits (128-dim `fp32`).

Table 20: **Full voice conversion results**

| Model | Intelligibility | | Quality | Speaker | Prosody |
|---|---|---|---|---|---|
| | WER↓ | CER↓ | UTMOS↑ | EER↑ | F0Corr↑ |
| Ground Truth | 0.0 | 0.0 | 4.08 | – | – |
| KM 12.5Hz | 1.8 | 0.8 | 4.11 | 27.0 | 0.53 |
| kNN-VC | 0.7 | 0.3 | 3.89 | 34.1 | 0.59 |
| LinearVC | 0.6 | 0.2 | 3.94 | 29.7 | 0.62 |
| FreeVC | 0.6 | 0.3 | 3.99 | 29.0 | 0.67 |
| CosyVoice 2 | 1.1 | 0.5 | 4.11 | 31.0 | 0.64 |
| PAST 1:8 | 22.9 | 15.1 | 1.84 | 8.2 | 0.20 |
| PAST 1:4 | 13.3 | 8.3 | 1.80 | 5.4 | 0.17 |
| PAST 1:2 | 6.6 | 3.8 | 1.69 | 3.9 | 0.17 |
| ST 1:8 | 74.7 | 61.7 | 1.54 | 10.6 | 0.19 |
| ST 1:4 | 35.2 | 26.1 | 1.62 | 8.9 | 0.19 |
| ST 1:2 | 10.6 | 6.0 | 1.52 | 5.8 | 0.22 |
| TiCodec 1:4 | **0.5** | **0.2** | 3.32 | 5.4 | **0.77** |
| TiCodec 1:2 | 3.4 | 1.9 | 3.13 | 5.7 | 0.74 |
| TiCodec 1:1 | 10.2 | 6.1 | 3.25 | 8.9 | 0.64 |
| Mimi 1:8 | 120.3 | 86.8 | 3.09 | **38.5** | 0.24 |
| Mimi 1:4 | 110.8 | 84.6 | 2.15 | 15.2 | 0.21 |
| Mimi 1:2 | 102.4 | 85.3 | 1.59 | 5.1 | 0.18 |
| DualCodec 1:8 | 21.5 | 12.9 | 2.50 | 6.8 | 0.54 |
| DualCodec 1:4 | 8.5 | 4.6 | 2.88 | 7.1 | 0.56 |
| DualCodec 1:2 | 4.4 | 2.3 | 3.07 | 5.8 | 0.62 |
| BiCodec | 1.2 | 0.6 | 3.84 | 18.5 | 0.61 |
| FACodec | 0.8 | 0.4 | 3.45 | 18.6 | 0.66 |
| Kanade 25Hz | 0.7 | 0.3 | 4.16 | 30.7 | 0.71 |
| Kanade 12.5Hz | 1.6 | 0.7 | **4.17** | 32.0 | 0.64 |

Table 21: **Full reconstruction MUSHRA results** with 95% confidence intervals.

| Model | − | Median | + |
|---|---|---|---|
| Ground Truth | 76.0 | 78.0 | 80.0 |
| Cont. 50Hz | 72.1 | 76.7 | 80.3 |
| KM 12.5Hz | 66.9 | 72.1 | 76.2 |
| KM 25Hz | 68.2 | 72.4 | 76.1 |
| ST 1:8 | 72.0 | 76.0 | 78.0 |
| ST 1:4 | 64.9 | 74.2 | 78.8 |
| DualCodec 1:8 | 73.5 | 75.6 | 80.9 |
| DualCodec 1:2 | 68.2 | 72.4 | 75.6 |
| FACodec | 77.8 | 81.4 | 83.4 |
| PAST 1:8 | 78.3 | 82.4 | 84.5 |
| StableCodec | 75.2 | 79.3 | 81.4 |
| X-Codec 2 | 74.0 | 77.0 | 80.0 |
| BiCodec | 72.0 | 75.0 | 79.0 |
| WavTokenizer | 65.9 | 72.1 | 76.2 |
| Kanade 12.5Hz | 70.3 | 74.5 | 77.7 |
| w/o GAN | 59.0 | 69.0 | 74.0 |
| w/o Dual-branch | 15.0 | 24.0 | 46.5 |
| w/o SSL Recon. | 57.5 | 68.5 | 75.5 |
| w/o End-to-End | 51.1 | 60.7 | 66.6 |
| w/o FSQ | 31.4 | 43.7 | 55.9 |
| Kanade 25Hz | 72.0 | 75.0 | 78.0 |
| w/o GAN | 66.0 | 70.3 | 75.6 |

Table 22: **Full voice conversion speaker similarity MUSHRA results** with 95% confidence intervals.

| Model | − | Median | + |
|---|---|---|---|
| Ground Truth | 72.0 | 74.5 | 77.0 |
| KM 12.5Hz | 71.0 | 74.0 | 76.0 |
| kNN-VC | 69.0 | 73.0 | 75.5 |
| LinearVC | 69.3 | 73.4 | 78.1 |
| FreeVC | 71.0 | 74.5 | 77.5 |
| CosyVoice 2 | 73.0 | 76.0 | 79.0 |
| ST | 25.0 | 35.0 | 47.5 |
| Mimi | 77.6 | 81.7 | 85.9 |
| DualCodec | 34.0 | 52.0 | 68.0 |
| FACodec | 51.7 | 62.6 | 69.3 |
| PAST | 15.5 | 23.3 | 50.7 |
| TiCodec | 57.0 | 68.0 | 73.0 |
| BiCodec | 66.7 | 71.4 | 75.5 |
| Kanade 12.5Hz | 72.4 | 77.6 | 81.7 |
| Kanade 25Hz | 73.4 | 77.1 | 80.7 |

Table 23: **Full TTS speech quality MUSHRA results** with 95% confidence intervals.

| Model | − | Median | + |
|---|---|---|---|
| Ground Truth | 72.0 | 74.9 | 77.1 |
| KM 25Hz | 71.5 | 74.9 | 79.3 |
| KM 12.5Hz | 67.0 | 72.0 | 78.5 |
| CosyVoice 2 | 74.9 | 77.1 | 79.3 |
| ST | 69.0 | 75.0 | 78.0 |
| Mimi | 71.5 | 74.9 | 78.2 |
| DualCodec | 69.0 | 73.0 | 78.0 |
| PAST | 70.4 | 74.9 | 79.3 |
| TiCodec | 71.5 | 73.8 | 77.1 |
| StableCodec | 64.0 | 71.0 | 77.0 |
| X-Codec 2 | 68.0 | 72.0 | 78.0 |
| BiCodec | 69.8 | 73.8 | 76.0 |
| WavTokenizer | 68.0 | 74.5 | 79.0 |
| Kanade 12.5Hz | 72.6 | 77.1 | 79.3 |
| Kanade 25Hz | 67.0 | 73.0 | 80.0 |

Table 24: **Full TTS prosody naturalness MUSHRA results** with 95% confidence intervals.

| Model | − | Median | + |
|---|---|---|---|
| Ground Truth | 78.9 | 80.9 | 83.0 |
| KM 12.5Hz | 60.0 | 67.0 | 73.0 |
| KM 25Hz | 69.8 | 75.9 | 78.9 |
| CosyVoice 2 | 80.9 | 83.0 | 85.5 |
| ST | 75.0 | 79.0 | 81.0 |
| Mimi | 66.8 | 73.9 | 78.4 |
| DualCodec | 74.0 | 80.0 | 83.0 |
| PAST | 72.9 | 78.4 | 81.5 |
| TiCodec | 65.8 | 72.9 | 78.9 |
| StableCodec | 58.0 | 66.0 | 74.5 |
| X-Codec 2 | 75.0 | 78.0 | 81.0 |
| BiCodec | 73.9 | 78.9 | 82.0 |
| WavTokenizer | 73.0 | 77.0 | 80.0 |
| Kanade 12.5Hz | 73.9 | 77.9 | 80.9 |
| Kanade 25Hz | 78.0 | 81.0 | 83.0 |

Table 25: **Full OOD reconstruction results (Part I).** Evaluation on noisy (Gigaspeech) and emotional (Salmon) speech. † indicates models trained on noisy data.

| Model | Intelligibility | | Quality | | Speaker | | Prosody | |
|---|---|---|---|---|---|---|---|---|
| | WER↓ | CER↓ | UTMOS↑ | Mel L1↓ | SIM↑ | MCD↓ | F0Corr↑ | F0RMSE↓ |
| *Gigaspeech (Chen et al., 2021) (noisy speech)* | | | | | | | | |
| Ground Truth | 9.7 | 5.1 | 2.84 | – | – | – | – | – |
| FACodec 1:6 | 11.3 | 6.3 | 2.85 | 0.88 | 0.97 | 5.32 | 0.88 | 0.07 |
| PAST 1:8 | 10.9 | 6.0 | 3.09 | 0.86 | 0.98 | 5.19 | 0.89 | 0.07 |
| PAST 1:4 | 12.6 | 7.1 | 2.70 | 0.99 | 0.96 | 5.95 | 0.81 | 0.11 |
| PAST 1:2 | 18.5 | 11.1 | 1.78 | 1.41 | 0.85 | 7.62 | 0.27 | 0.34 |
| ST 1:8 | 11.8 | 6.6 | 2.60 | 0.85 | 0.97 | 5.45 | 0.88 | 0.08 |
| ST 1:4 | 14.7 | 8.8 | 2.41 | 1.04 | 0.93 | 6.47 | 0.83 | 0.10 |
| ST 1:2 | 21.4 | 13.1 | 1.71 | 1.44 | 0.85 | 8.28 | 0.75 | 0.13 |
| TiCodec 1:4 | 12.4 | 7.0 | 2.45 | 0.91 | 0.95 | 6.49 | 0.86 | 0.08 |
| TiCodec 1:2 | 18.5 | 11.5 | 2.35 | 1.08 | 0.91 | 7.35 | 0.83 | 0.09 |
| TiCodec 1:1 | 31.4 | 21.0 | 2.25 | 1.21 | 0.88 | 8.05 | 0.74 | 0.13 |
| Mimi 1:8† | 12.3 | 7.0 | 2.71 | 1.23 | 0.96 | 6.66 | 0.85 | 0.09 |
| Mimi 1:4† | 16.0 | 9.6 | 2.37 | 1.45 | 0.93 | 7.68 | 0.79 | 0.11 |
| Mimi 1:2† | 22.6 | 14.2 | 1.98 | 1.81 | 0.85 | 9.35 | 0.58 | 0.17 |
| DualCodec 1:8† | 11.0 | 6.0 | 3.11 | 0.76 | 0.98 | 4.62 | 0.84 | 0.08 |
| DualCodec 1:4† | 12.3 | 7.0 | 3.07 | 0.91 | 0.96 | 5.54 | 0.83 | 0.09 |
| DualCodec 1:2† | 15.8 | 9.3 | 2.78 | 1.16 | 0.93 | 6.81 | 0.81 | 0.10 |
| X-Codec 2† | 11.5 | 6.3 | 2.99 | 0.89 | 0.97 | 5.53 | 0.87 | 0.08 |
| BiCodec† | 11.9 | 6.6 | 3.07 | 1.28 | 0.96 | 6.36 | 0.87 | 0.08 |
| WavTokenizer | 33.9 | 21.9 | 2.64 | 1.19 | 0.88 | 6.97 | 0.82 | 0.10 |
| StableCodec† | 27.1 | 16.3 | 3.51 | 1.65 | 0.90 | 8.70 | 0.84 | 0.09 |
| Kanade 12.5Hz | 16.2 | 9.3 | 3.25 | 1.44 | 0.95 | 7.63 | 0.74 | 0.13 |
| Kanade 25Hz | 11.3 | 6.2 | 3.27 | 1.21 | 0.96 | 6.61 | 0.81 | 0.09 |
| *Salmon Sentiment (Maimon et al., 2025b) (emotional)* | | | | | | | | |
| Ground Truth | 2.9 | 1.0 | 3.79 | – | – | – | – | – |
| w/ change | 4.9 | 1.6 | 3.62 | – | – | – | – | – |
| FACodec 1:6 | 3.8 | 1.2 | 3.87 | 0.77 | 0.98 | 4.92 | 0.92 | 0.08 |
| w/ change | 4.4 | 1.8 | 3.77 | 0.80 | 0.98 | 5.20 | 0.90 | 0.09 |
| FACodec 1:3 | 3.9 | 1.4 | 3.32 | 1.12 | 0.97 | 6.19 | 0.79 | 0.15 |
| w/ change | 5.9 | 2.2 | 3.34 | 1.18 | 0.96 | 6.42 | 0.78 | 0.17 |
| PAST 1:8 | 3.0 | 1.0 | 3.91 | 0.78 | 0.99 | 5.10 | 0.85 | 0.09 |
| w/ change | 4.2 | 1.7 | 3.77 | 0.78 | 0.98 | 5.09 | 0.90 | 0.08 |
| PAST 1:4 | 3.9 | 1.4 | 3.46 | 0.97 | 0.96 | 5.86 | 0.86 | 0.10 |
| w/ change | 5.4 | 2.0 | 3.32 | 0.98 | 0.95 | 6.09 | 0.85 | 0.13 |
| PAST 1:2 | 6.9 | 3.0 | 1.96 | 1.56 | 0.72 | 7.97 | 0.20 | 0.49 |
| w/ change | 6.1 | 2.8 | 1.92 | 1.56 | 0.68 | 7.97 | 0.10 | 0.49 |
| ST 1:8 | 3.9 | 1.2 | 3.53 | 0.82 | 0.97 | 5.58 | 0.86 | 0.10 |
| w/ change | 4.3 | 1.7 | 3.42 | 0.81 | 0.97 | 5.54 | 0.86 | 0.10 |
| ST 1:4 | 5.2 | 1.7 | 3.15 | 1.02 | 0.92 | 6.52 | 0.79 | 0.11 |
| w/ change | 7.4 | 3.7 | 3.11 | 1.02 | 0.91 | 6.59 | 0.88 | 0.12 |
| ST 1:2 | 9.4 | 3.8 | 2.30 | 1.39 | 0.82 | 8.28 | 0.72 | 0.16 |
| w/ change | 10.6 | 5.2 | 2.30 | 1.39 | 0.82 | 8.32 | 0.80 | 0.15 |
| TiCodec 1:4 | 3.9 | 1.3 | 3.44 | 0.86 | 0.96 | 6.09 | 0.92 | 0.10 |
| w/ change | 4.5 | 1.9 | 3.28 | 0.86 | 0.96 | 6.19 | 0.86 | 0.11 |
| TiCodec 1:2 | 6.0 | 2.7 | 3.07 | 1.05 | 0.92 | 7.12 | 0.88 | 0.10 |
| w/ change | 8.0 | 4.3 | 3.00 | 1.06 | 0.91 | 7.28 | 0.81 | 0.12 |
| TiCodec 1:1 | 16.7 | 9.3 | 2.98 | 1.18 | 0.88 | 7.70 | 0.75 | 0.16 |
| w/ change | 19.0 | 10.6 | 2.83 | 1.20 | 0.87 | 7.66 | 0.78 | 0.15 |
| Mimi 1:8† | 4.1 | 1.8 | 3.22 | 1.29 | 0.96 | 6.94 | 0.82 | 0.11 |
| w/ change | 5.9 | 3.0 | 3.09 | 1.28 | 0.95 | 7.09 | 0.79 | 0.14 |
| Mimi 1:4† | 6.1 | 2.9 | 2.75 | 1.56 | 0.91 | 8.07 | 0.75 | 0.14 |
| w/ change | 8.5 | 4.6 | 2.65 | 1.55 | 0.91 | 8.17 | 0.77 | 0.16 |
| Mimi 1:2† | 13.2 | 8.4 | 2.18 | 2.03 | 0.83 | 9.86 | 0.48 | 0.23 |
| w/ change | 14.7 | 9.1 | 2.18 | 2.00 | 0.82 | 9.80 | 0.53 | 0.24 |
| DualCodec 1:8† | 3.6 | 1.1 | 3.91 | 0.71 | 0.98 | 4.45 | 0.88 | 0.08 |
| w/ change | 4.4 | 1.8 | 3.76 | 0.71 | 0.98 | 4.41 | 0.90 | 0.10 |
| DualCodec 1:4† | 4.7 | 1.8 | 3.88 | 0.88 | 0.97 | 5.56 | 0.78 | 0.12 |
| w/ change | 5.3 | 2.4 | 3.77 | 0.88 | 0.97 | 5.36 | 0.91 | 0.10 |
| DualCodec 1:2† | 6.8 | 2.9 | 3.46 | 1.13 | 0.94 | 6.63 | 0.80 | 0.13 |
| w/ change | 7.0 | 3.4 | 3.41 | 1.12 | 0.94 | 6.74 | 0.81 | 0.15 |
| X-Codec 2† | 3.8 | 1.2 | 3.77 | 0.84 | 0.97 | 5.40 | 0.85 | 0.09 |
| w/ change | 5.7 | 2.2 | 3.67 | 0.85 | 0.97 | 5.45 | 0.89 | 0.11 |
| BiCodec† | 5.4 | 1.7 | 3.84 | 1.21 | 0.98 | 6.00 | 0.81 | 0.10 |
| w/ change | 6.0 | 2.6 | 3.73 | 1.24 | 0.97 | 6.11 | 0.90 | 0.11 |
| WavTokenizer | 14.5 | 7.7 | 3.21 | 1.12 | 0.90 | 6.70 | 0.74 | 0.12 |
| w/ change | 17.5 | 9.7 | 3.13 | 1.13 | 0.90 | 6.93 | 0.82 | 0.16 |
| StableCodec† | 14.8 | 7.2 | 4.08 | 1.45 | 0.93 | 7.87 | 0.81 | 0.12 |
| w/ change | 18.0 | 9.3 | 4.03 | 1.49 | 0.92 | 7.98 | 0.84 | 0.12 |
| Kanade 12.5Hz | 6.4 | 2.3 | 3.83 | 1.38 | 0.95 | 7.72 | 0.66 | 0.19 |
| w/ change | 7.0 | 3.1 | 3.83 | 1.50 | 0.94 | 8.22 | 0.67 | 0.22 |
| Kanade 25Hz | 4.4 | 1.5 | 3.85 | 1.12 | 0.96 | 6.57 | 0.73 | 0.16 |
| w/ change | 4.7 | 1.9 | 3.88 | 1.21 | 0.96 | 6.80 | 0.75 | 0.18 |

Table 26: **Full OOD reconstruction results (Part II).** Evaluation on unseen language (JVS) and accented speech (ERJ). † indicates models trained on Japanese.

| Model | Intelligibility | | Quality | | Speaker | | Prosody | |
|---|---|---|---|---|---|---|---|---|
| | WER↓ | CER↓ | UTMOS↑ | Mel L1↓ | SIM↑ | MCD↓ | F0Corr↑ | F0RMSE↓ |
| JVS (Takamichi et al., 2019) *(unseen language)* | | | | | | | | |
| Ground Truth | 4.6 | 2.5 | 3.63 | – | – | – | – | – |
| FACodec 1:6 | 5.1 | 2.8 | 3.69 | 0.76 | 0.97 | 5.80 | 0.90 | 0.09 |
| FACodec 1:3 | 6.4 | 3.5 | 2.89 | 1.04 | 0.95 | 6.89 | 0.79 | 0.18 |
| PAST 1:8 | 5.2 | 2.8 | 3.62 | 0.84 | 0.98 | 5.98 | 0.88 | 0.09 |
| PAST 1:4 | 7.3 | 4.1 | 2.73 | 1.06 | 0.91 | 7.13 | 0.80 | 0.16 |
| PAST 1:2 | 17.0 | 10.8 | 1.63 | 1.57 | 0.64 | 9.42 | 0.16 | 0.53 |
| ST 1:8 | 5.7 | 3.2 | 3.32 | 0.79 | 0.96 | 6.22 | 0.86 | 0.10 |
| ST 1:4 | 7.8 | 4.6 | 2.87 | 0.96 | 0.90 | 7.06 | 0.82 | 0.12 |
| ST 1:2 | 16.0 | 10.4 | 2.02 | 1.31 | 0.80 | 8.70 | 0.81 | 0.15 |
| TiCodec 1:4 | 5.6 | 3.1 | 3.21 | 0.76 | 0.95 | 5.44 | 0.86 | 0.10 |
| TiCodec 1:2 | 8.5 | 4.8 | 3.06 | 0.89 | 0.92 | 6.34 | 0.85 | 0.10 |
| TiCodec 1:1 | 18.9 | 13.2 | 2.69 | 1.03 | 0.86 | 7.15 | 0.81 | 0.15 |
| Mimi 1:8 | 7.7 | 4.5 | 2.94 | 1.11 | 0.94 | 6.55 | 0.83 | 0.11 |
| Mimi 1:4 | 12.7 | 8.0 | 2.48 | 1.30 | 0.86 | 7.88 | 0.81 | 0.15 |
| Mimi 1:2 | 22.9 | 16.9 | 1.86 | 1.63 | 0.73 | 10.00 | 0.56 | 0.26 |
| DualCodec 1:8† | 5.0 | 2.8 | 3.67 | 0.64 | 0.99 | 4.46 | 0.81 | 0.09 |
| DualCodec 1:4† | 5.5 | 3.1 | 3.64 | 0.77 | 0.97 | 5.40 | 0.83 | 0.10 |
| DualCodec 1:2† | 7.8 | 4.4 | 3.24 | 0.97 | 0.96 | 6.57 | 0.83 | 0.11 |
| X-Codec 2† | 5.4 | 2.9 | 3.59 | 0.76 | 0.98 | 5.28 | 0.89 | 0.10 |
| BiCodec | 5.7 | 3.1 | 3.73 | 1.62 | 0.98 | 7.67 | 0.86 | 0.10 |
| WavTokenizer | 18.2 | 11.3 | 2.92 | 1.01 | 0.88 | 6.92 | 0.82 | 0.14 |
| StableCodec | 25.0 | 16.5 | 3.83 | 1.99 | 0.91 | 10.36 | 0.90 | 0.10 |
| Kanade 12.5Hz | 12.2 | 7.2 | 3.77 | 1.30 | 0.94 | 8.15 | 0.70 | 0.21 |
| Kanade 25Hz | 5.6 | 3.0 | 3.72 | 1.03 | 0.97 | 6.55 | 0.84 | 0.17 |
| ERJ (Nakagawa, 2007) *(accented speech)* | | | | | | | | |
| Ground Truth | 14.9 | 8.0 | 3.73 | – | – | – | – | – |
| FACodec 1:6 | 18.2 | 9.9 | 3.73 | 0.73 | 0.98 | 4.68 | 0.90 | 0.06 |
| FACodec 1:3 | 22.0 | 12.3 | 3.37 | 0.95 | 0.97 | 5.40 | 0.81 | 0.09 |
| PAST 1:8 | 25.3 | 14.1 | 3.65 | 0.90 | 0.97 | 5.40 | 0.85 | 0.07 |
| PAST 1:4 | 33.7 | 19.4 | 3.04 | 1.14 | 0.92 | 6.32 | 0.75 | 0.12 |
| PAST 1:2 | 47.3 | 27.8 | 2.00 | 1.56 | 0.79 | 7.96 | 0.30 | 0.30 |
| ST 1:8 | 19.6 | 10.8 | 3.48 | 0.76 | 0.97 | 5.13 | 0.89 | 0.06 |
| ST 1:4 | 28.5 | 15.7 | 3.11 | 0.93 | 0.94 | 6.12 | 0.82 | 0.09 |
| ST 1:2 | 47.5 | 27.1 | 1.97 | 1.28 | 0.84 | 7.67 | 0.66 | 0.13 |
| TiCodec 1:4 | 18.3 | 10.3 | 3.29 | 0.79 | 0.96 | 5.96 | 0.88 | 0.07 |
| TiCodec 1:2 | 26.7 | 15.8 | 3.13 | 0.95 | 0.94 | 6.90 | 0.82 | 0.08 |
| TiCodec 1:1 | 47.9 | 30.1 | 2.82 | 1.08 | 0.92 | 7.48 | 0.80 | 0.10 |
| Mimi 1:8 | 27.3 | 17.1 | 2.84 | 1.36 | 0.95 | 6.88 | 0.84 | 0.08 |
| Mimi 1:4 | 45.5 | 30.6 | 2.31 | 1.62 | 0.90 | 8.19 | 0.74 | 0.11 |
| Mimi 1:2 | 67.5 | 49.0 | 1.70 | 2.19 | 0.73 | 10.89 | 0.46 | 0.22 |
| DualCodec 1:8† | 17.1 | 9.4 | 3.71 | 0.66 | 0.98 | 4.27 | 0.86 | 0.07 |
| DualCodec 1:4† | 21.5 | 11.9 | 3.66 | 0.80 | 0.96 | 5.18 | 0.83 | 0.08 |
| DualCodec 1:2† | 29.2 | 16.7 | 3.25 | 1.04 | 0.94 | 6.45 | 0.80 | 0.09 |
| X-Codec 2† | 20.7 | 11.3 | 3.69 | 0.78 | 0.97 | 5.16 | 0.86 | 0.08 |
| BiCodec | 21.4 | 11.7 | 3.76 | 2.25 | 0.97 | 8.75 | 0.86 | 0.07 |
| WavTokenizer | 51.7 | 31.6 | 3.06 | 1.06 | 0.91 | 6.45 | 0.82 | 0.08 |
| StableCodec | 51.4 | 29.3 | 4.03 | 2.52 | 0.91 | 10.76 | 0.87 | 0.06 |
| Kanade 12.5Hz | 33.8 | 18.6 | 3.78 | 1.28 | 0.95 | 6.80 | 0.80 | 0.09 |
| Kanade 25Hz | 22.9 | 12.3 | 3.75 | 1.05 | 0.96 | 5.85 | 0.86 | 0.07 |

