# OpenReview forum: "Kanade: Compact Linguistically Rich Speech Tokens for Spoken Language Models"
_ICLR.cc/2026/Conference — Submitted to ICLR 2026_

### Official Review · Reviewer_LiTB · 2025-10-20

**Soundness:** 3
**Presentation:** 3
**Contribution:** 2
**Rating:** 4
**Confidence:** 5

**Summary:**

The authors propose a model for tokenizing speech in discrete space. The work aims to create a compact set of tokens for representing speech audio. In particular, the authors try to separate speaker identity by having a global, frame-static encoder and to make the tokens represent linguistic content beyond phonetic. The resulting model shows improved performance on a clean English audiobook, with demonstrations on some additional downstream tasks including ASR, TTS, SID, ASV, and VC. However, the novelty in approach is limited–SSL distillation and using early layers for speaker encoding–which were readily tried in previous studies without core breakthrough in methodology. Moreover, while including diverse downstream tasks (more than usual codec papers), some key caveats that may be induced in their methods remain unclear. These are more elaborated in the Weakness section.

**Strengths:**

* Better bitrate–intelligibility/quality tradeoff than previous studies in clean English audiobook data (However, this is not fair since, these metrics are ignoring the coding space of global embedding which is not discretized.)

* Improved performance in downstream tasks with

* Diverse and interesting analyses with visualizations. Those analyses provide many interesting insights, which are not usually covered in common Codec papers. It has a value as an analysis paper.

**Weaknesses:**

* Lack of novelty: the main issue of this work is that it severely lacks novelty. The separation of frame-wise and global embedding is dated and included in several previous papers (FACodec, TICodec, LSCodec).

* The coding complexity of global embedding is entirely ignored. The bitrate is calculated only using frame-wise codes. This omission makes several model comparisons significantly unfair, especially in Figure 4, where the other codecs are using all codes participating in the evaluation. This severely disguises the fact that the actual bitrate of Kanade is infinite to fully reconstruct the audio. The suggested reason for this is that the global embedding “would not be used for language modeling”, which is one niche application of codecs. Putting aside whether language modeling is the primary downstream of codec, if the authors would like to claim so, then the spoken language modeling results in Appendix should be presented as primary, main results.

* Lack of fairness in model comparison. For the baselines, the authors are too selective in baseline applications by truncating the codebooks. The rationale behind this decision is not well supported. As stated in Section 3.2, they controlled bitrate of the baselines for fairness of comparison. However, the selected bitrates  in Table 1 are not consistently controlled across models, having non-uniform bitrates. Moreover, it is more fair to use all codebooks since the proposed model is using the infinite bitrate of global embedding. Also, since the baselines are not designed for truncated employment of codes in reconstruction tasks, it is more fair to include a full codebook scores along with the truncated ones in Table 1, like in Figure 4.

* Performance improvement might be driven by potential in-domain overfitting. The training and test data distribution are more matching in Kanade than other baseline models since some core metrics in reconstruction, TTS, voice conversion are done in read English speech. Especially, for lower bitrate (putting aside the fact it is broken), could be achieved by not covering diverse speaking styles in the code space. The evaluation on OOD datasets–Expresso, Japanese and L2 speech–and  in Appendix D.6 doesn’t include other baselines. The F0 reconstruction severely drops compared to the in-domain evaluation in Table 1. And some quality metrics are not informative due to the absence of the baselines. If the baselines, for example the full codebooks of Mimi, are better in those OOD data. The reduction in bitrate is likely induced by being limited in the data domain.

* Lack of support for several key claims. Claim for “linguistically rich.” The conceptual implementation is to merge prosody and phonetic information in a single token stream. However, to claim “richness” in linguistic information, there exists other key aspects like lexical semantics, syntactics, etc. For example, Pasad et al. (20xx, 20xx) show some lexical semantics are represented in speech SSL models. Nonetheless, none of the analyses is supporting evidence for representing those other key linguistic components. ASR and TTS are still operating at the surface level–matching written and spoken formats vice versa. In particular, as shown in Table 13 in Appendix D.4, the proposed tokens are worse in spoken language modeling compared to the previous SSL-distilled Codecs, which imply it lacks those linguistic information. Though the main focus of this paper is targeting the “surface” of the language, which is also denoted well by the authors, being limited in surface level is not compatible with being rich.

**Questions:**

* For ASR, I wonder if all RVQ codes were used for the baselines. If not, that version should be reported as well for a fair comparison.

* Does the model work well with the speech clips with changing voice quality? For example, if a speaker changes his voicing texture within a sentence, how does the model behave? Does it still reconstruct well for both textures?

* I recommend comparing other similar recent models like TiCodec or LSCodec which have the same branching architecture as Kanade.

* The computing complexity is another useful indication of the utility of codec, especially for real-time downstream tasks. I wonder about the real time factor of Kanade in both encoding and decoding, respectively, compared to the baselines.

---

> ### Author Response · Authors · 2025-11-22
>
> We apologize for the late response. We hope that we can address your concerns in full below. If you have any additional comments or have new concerns, please let us know.
>
> **Marginal bitrate**:
> Before making any theoretical arguments, we note that reporting total bitrate is difficult, since it is a function of the audio length. This is not a fixed value, so it would be cumbersome to report. It is also worth noting that prior work, such as FACodec and BiCodec, also uses marginal bitrate.
>
> The motivation for compact representations is not general speech compression, but ease of modeling. In the introduction, we state:
> > The neural networks used in SLMs can learn more efficiently if we provide them with representations that contain only relevant information (Tishby & Zaslavsky, 2015). Just as it is wasteful to learn models of images at the pixel level, which is correlated and noisy (van den Oord et al., 2017), language modeling on verbose representations may be wasteful.
>
> We have updated the introduction to include a more complete motivation and contextualization of disentangled tokenization. It explains that feeding only certain bits to downstream models (almost always the content bits) can indeed simplify downstream modeling:
> > Disentangled speech tokenizers aim to separate information into multiple streams, often time-varying content and acoustic invariants. Since many linguistically irrelevant features like speaker identity and microphone characteristics are constant, this allows the content stream to contain easily-accessible linguistic information, while relegating information necessary for reconstruction to a separate representation.  Martin-Cortinas et al.\[2\]  have shown that using only the content stream can improve the performance of downstream language modeling. The authors hypothesize that this is because the model learns the content distribution, rather than a more complicated joint distribution of speaker information and content. We can avoid the tradeoff described above by preserving reconstruction information, but *not feeding it to models that don't need it*.
>
> We have added the sizes of the global embedding to the paper as a footnote where marginal bitrate is explained (our global embedding is 4096 bits, FACodec's timbre embedding is 8192 bits, BiCodec's global tokens take 384 bits, and TiCodec's time-invariant tokens take 80 bits). We have also updated the section explaining why we use marginal bitrate:
> > (Marginal bitrate) measures the amount of information that downstream models must process, not suitability for general speech compression. All fixed-size embeddings (footnote for sizes) are from global branches of disentangled codecs and are not used in any language modeling tasks like ASR, TTS, or autoregressive SLMs. Though they may be used for discriminative tasks like speaker recognition, these are not the focus of this work.
>
> Furthermore, while speech reconstruction (used as a benchmark to estimate synthesis quality) does require a global embedding, we have updated the introduction to explain:
> > At inference time, it usually suffices to use only the content branch. Kanade’s state-of-the-art disentanglement ensures that the global embedding does not encode content, so it does not need to be calculated for ASR or intent classification. Generative models like SLMs only need to generate a content stream, which can then be decoded to speech using a single baked-in global embedding. This fixed embedding might be considered the “voice” of the model.
>
> Even for speech reconstruction, there is no limit to the amount of speech that can be reconstructed with a single global embedding, since the linguistic content is in the content tokens (as evidenced by VC experiments). We have included a simple experiment with chunked reconstruction using a single global embedding for clips of arbitrary length in the appendix "Chunked encoding and decoding". In this situation, the size of the 128-dim 32-bit floating point global embedding (4096 bits) is amortized over the length of the entire track (which can be of arbitrary size).

---

> ### Author Response · Authors · 2025-11-22
>
> **Novelty**: (Response shares large amount of text with our response to yxWZ) We agree that the dual-branch design is not new, but failed to properly explain this in the main text. We have updated the introduction with the following (citations replaced by model names):
> > Disentangled speech tokenizers aim to separate information into multiple streams, often time-varying content and acoustic invariants. Since many linguistically irrelevant features like speaker identity and microphone characteristics are constant, this allows the content stream to contain easily- accessible linguistic information, while relegating information necessary for reconstruction to a separate representation. Martin-Cortinas et al. (2024) (SSVC) have shown that using only the content stream can improve the performance of downstream language modeling. The authors hypothesize that this is because the model learns the content distribution, rather than a more complicated joint distribution of speaker information and content. We can avoid the tradeoff described above by preserving reconstruction information, but not feeding it to models that don’t need it. Most disentangled speech tokenizers use a multi-branch architecture along with at least one additional method to encourage disentanglement, such time invariance (TiCodec, Single-Codec, SoCodec), data augmentation (LSCodec), supervision (FACodec), or using pretrained models (FreeCodec).
>
> We have also included a discussion section in the conclusion to explicitly point out what we have contributed:
> > Unlike most disentangled tokenizers, Kanade separates global information from content only by the architecture of the model itself. To our knowledge, only BiCodec (Wang et al., 2025a) has attempted this. However, it has a more complicated global branch and, as shown experiments, does not achieve disentanglement... Like RepCodec (Huang et al., 2024), Kanade uses only SSL features as input. But ablations show that end-to-end training with reconstruction loss better preserves prosody and speech quality than RepCodec’s 2-step approach. Using SSL features allows training with only 600 hours of data and 120M unfrozen parameters. Ye et al. (2025b) presented a single-layer codec with FSQ, but we show that disentangling the content yields tokens more suitable for TTS and SLMs. This reduces complexity, since there is no need for an additional step to produce finer tokens, and also allows the model to better attend to suprasegmental features.
>
>
> **SLM in the main text**: We ran additional experiments using the [*slam*](https://pages.cs.huji.ac.il/adiyoss-lab/slamming/) recipe and have moved the results to the main text. The results show that Kanade is competitive with k-means and distilled hybrid codecs. We are still working to compute results for old and new baselines.
>
> **Codebook truncation**: Many RVQ codecs are trained for truncation using layer dropout, following EnCodec. Examples include Mimi, FACodec, and DualCodec. However, we agree that arbitrarily cutting off the codebooks for reconstruction is questionable. For a more principled approach, we recalculated reconstruction results for many codebook layer prefixes and picked the best results for the reconstruction table in the main text. Full results can be found in the last appendix.
>
> **Potential in-domain overfitting**: We have updated the OOD appendix to include baselines. The results show that Kanade performs very well compared to baselines on noisy/conversational speech (Gigaspeech), emotional speech (Salmon), inconsistent emotional style (Salmon w/ style change), unseen languages (JVS), and L2 speech (ERJ).
>
> You were concerned about a drop in F0 correlation for Gigaspeech. Was this for the *w/ clean global emb.* case? If so, we found that the LJSpeech global embedding we used was very different from speech in Gigaspeech (in particular, it is very expressive). We have replaced the global embedding for that experiment by a more standard one from Gigaspeech itself and the F0 correlation is more in line with VC results.
>
> Furthermore, we plan to add ASR OOD results using Switchboard and TTS OOD results using Common Voice.

---

> > ### Author Response · Authors · 2025-11-22
> >
> > **Linguistic richness**: In the introduction, we state:
> > > Just like text, good representations should surface the basic units of lanugages... With low-level phonetic and prosodic information, an SLM can recover higher-level features like morphology or syntax.
> >
> > Choi et al. show in \[1\] that SSL representations mostly encode phonetics: similar-sounding words are encoded closer to each other than words with similar meaning. They also show that for semantic tasks such as intent classification, using a "bag of words" (only word identity) matches or outperforms SSL representations, making it unclear that they encode much more lexical semantics than the word identity. Even so, SSL representations have been successful in spoken language modeling.
> >
> > ---
> >
> > We answer specific questions here:
> >
> > > For ASR, I wonder if all RVQ codes were used for the baselines. If not, that version should be reported as well for a fair comparison.
> >
> > We thought that it would help improve baseline performance if ASR did not include the most detailed acoustic layers. We tested all models again and found that including all RVQ layers yields the best results for all multi-layer models. We apologize for this oversight and have updated the downstream task result table (Table 2) with these new results. The relative rankings of the best models have not changed much.
> >
> > > Does the model work well with the speech clips with changing voice quality? For example, if a speaker changes his voicing texture within a sentence, how does the model behave? Does it still reconstruct well for both textures?
> >
> > We have added reconstruction results to the OOD table (newly updated to include baselines) based on the [Salmon](https://pages.cs.huji.ac.il/adiyoss-lab/salmon/) sentiment consistency dataset. When there is a change in emotion within a sentence, metrics slightly degrade similarly in Kanade and all baselines. This was surprising to us because we expected non-disentangled codecs to be less sensitive. As additional evidence of robustness to within-utterance variation, we have also add a chunked resynthesis experiment in the appendix. It shows that 25-35 minutes of speech can be resynthesized well using a single global embedding.
> >
> > > I recommend comparing other similar recent models like TiCodec or LSCodec which have the same branching architecture as Kanade.
> >
> > BiCodec is the most similar prior work and was included as a baseline. We have also added TiCodec as a new baseline at your suggestion. LSCodec is not open-source and so cannot be compared. VC results show that Kanade achieves much better disentanglement than any baselines.
> >
> > > The computing complexity is another useful indication of the utility of codec, especially for real-time downstream tasks. I wonder about the real time factor of Kanade in both encoding and decoding, respectively, compared to the baselines.
> >
> > Total RTF was reported in the original paper, but we have separated it into encoding and decoding steps. Results are reported in the table "Training budget and inference efficiency". Kanade has excellent inference speech with an encoding RTF of 0.0009 and decoding RTF of 0.002.
> >
> > \[1\] K. Choi, A. Pasad, T. Nakamura, S. Fukayama, K. Livescu, and S. Watanabe, “Self-Supervised Speech Representations are More Phonetic than Semantic,” presented at the Proc. Interspeech 2024, 2024, pp. 4578–4582. doi: 10.21437/Interspeech.2024-1157.

---

> > > ### Comment · Reviewer_LiTB · 2025-11-26
> > >
> > > Thank you for providing the response.
> > >
> > > **Marginal bitrate**
> > > Regarding the marginal bitrate, I do understand the motivation that the only content tokens are used or generated in some downstream tasks. However the motivation does not justify some main figures and tables which are not about those tasks that only use content tokens. Specifically, as I commented in the first review, the Figure 4 is scientifically incorrect. The y-axis is reconstruction quality which is indeed using global embedding. If the motivation is for downstream tasks that are using content tokens only, then the y-axes of those plots should be replaced with those downstream tasks. Otherwise, the only FACodec or BiCodec that can report marginal bitrate should remain. Likewise, in Table 1 is not the right place report marginal BPS if not every baseline is doing so. If we follow the same rule, Mimi's bps can be very small by only calculating the first codebook bits and dumbing other codebooks as just acoustic embedding. The current presentation of results are not scientifically correct and the explanation that authors commented is not apparently visible in the captions. (Especially, for some readers just skim through figures and tables, this will definitely misleads the readers. This is very likely as the title contains the term "Compact".)
> > >
> > > Also, the variable nature of length of speech should not stop reporting the bitrate of global embedding. By assuming various scenario of average utterance lengths, the true bitrate can be reported. For example, if we assume the usecase scenario where the average utterance length is 5 seconds, 4096/5=800 (bps) should be added. For 10 seconds scenario, 400 bps should be added. Ideally, the curve of bitrate-performance in Figure 4 can be drawn in multiple lines with varying average length scenarios.
> > >
> > > I wouldn't complain that much if this is the standard of the field that some dozens of papers are reporting in that way. But that is not the case. Even for FACodec and BiCodec, I respectfully argue that their presentation is not accurate either.

---

> > > > ### Comment · Reviewer_LiTB · 2025-11-26
> > > >
> > > > **Novelty** It would be nice if the authors provide bullet points of the technical novelty in a clearer format. Being "simple" is hard to evaluate, and so far any of the building blocks is not novel in speech processing community. The specific combination  of those ideas in the paper may not be tried, but the effectiveness of the individual component is already shown. (And the combination method itself is not novel.) I would like to review it again if the authors provide the bullet points of the elements that make this work distinguished from other works.
> > > >
> > > > Regarding the statement that the previous models did not achieve disentanglement, I am not very convinced given the voice conversion results. The gain is very marginal from BiCodec. I am also wondering how the source similarity is in generally so high. These numbers seem to suggest none of them is disentangling well. But from my own experience of using those tokenizers, they do pretty reasonable. I am concerned with overall reliability of this evaluation.
> > > >
> > > > For speaker tasks, could authors clarify more on implementation details in Appendix E.3 of using global embedding? For this specific statement: "added to the token embedding sate each time step", why there is time step for global embedding? Doesn't the model just need to model using a single vector per clip? (ECAPA-TDNN should not be adequate model obiously in this case.)
> > > >
> > > > I would love to reevaluate the novelty after having more clarifications.

---

> > > > > ### Comment · Reviewer_LiTB · 2025-11-26
> > > > >
> > > > > **SLM** I appreciate the authors to accept the suggestion. I am wondering being competitive (and some are worse) than previous tokens is aligned with the claim of the paper. I would expect more significant gain in SLM to argue that the suggested new tokens are better in SLM. The results are hard to be regarded impressive given the gravity of SLM applications throughout the narrative of the paper.
> > > > >
> > > > > **Codebook truncation** Thank you for updating the table. By the way, Mimi and some other tokens are omitted in the main table, which seems an accident. However, the discussion in Section 3.4.1 seems not reflecting the change. Kanade is not the best anymore. Also, This statement "Our models achieve the best speaker similarity among low-bitrate codecs." should compare only tokens which uses global embedding, which is more natural grouping than low-bitrate to be fair.
> > > > >
> > > > > **Potential in-domain overfitting** The new results actually corroborate my original concerns. Kanade shows significantly larger degradation of prosody and speaker MCD in whisper and emotional speech. This indeed suggests in-domain overfitting and lack of generalization. Besides, given the main claim is in having better prosody in the tokens, these results are largely dampening the main claim. The numbers are definitely not competitive with the sota baselines...

---

> > > > > > ### Comment · Reviewer_LiTB · 2025-11-26
> > > > > >
> > > > > > **Linguistic richness** The explanation provided means the SLM can be linguistically rich, which does not necessarily mean the tokens used are linguistically rich. As stated by the authors, the richness is achieved by down stream models. We cannot say BPE in text LLM is linguistically rich even LLM can be very linguistically rich.
> > > > > >
> > > > > > I agree SSL models are likely not to have rich semantic contents. But that means linguistic richness is bounded when we solely rely on them. This actually supports my concern that using linguistically not rich frozen SSL features could give arise to linguistically rich tokens. Again, such richness arising from downstream tasks doesn't count. (Besides, even admitting so, the SLM results are not really supporting more richness of Kanade compared to Mimi.)

---

> ### Comment · Reviewer_LiTB · 2025-11-26
>
> Thank you for answering questions. The answers have cleared some of the details I was looking for.
>
> However, as commented above, several key limiting factors remain unsolved. Therefore, I keep the score unchanged. I am open to raise my score based on the authors' answers to my response to their rebuttal.

---

### Official Review · Reviewer_ihXt · 2025-10-31

**Soundness:** 3
**Presentation:** 3
**Contribution:** 2
**Rating:** 2
**Confidence:** 4

**Summary:**

The paper proposes Kanade, a two-branch speech tokenizer that takes SSL features, quantizes a content stream with FSQ, and routes speaker- and environment-related constants through a single global embedding for reconstruction. Training uses a joint SSL feature reconstruction and Mel L1 losses, followed by GAN post-training on the decoder. Models are evaluated for reconstruction and for discriminative (ASR/SID/ER) and generative (TTS, VC) downstream tasks. Reported highlights include strong reconstruction intelligibility at low marginal bitrate, competitive TTS, and strong VC disentanglement.

**Strengths:**

1. Clarity & simplicity of the architecture. A single-stream token sequence, combined with a separate global embedding, is a clean way to keep linguistic content compact and accessible to SLMs while offloading invariants to a non-autoregressive path.

2. Thorough evaluation and ablations. The paper covers reconstruction, discriminative and generative tasks, and includes ablations showing the roles of dual-branch design and loss formulation.

3. Good reconstruction intelligibility and VC disentanglement at low marginal bitrate.

4. Data/computation efficiency claims compared to some prior codecs are appealing.

**Weaknesses:**

# 1. Limited novelty / unclear source of gains

The global/content disentanglement pattern (local vs. global paths) and using SSL features with VQ-VAE/FSQ are all established. The paper itself situates Kanade within prior disentanglement lines and VQ-VAE tokenizers, emphasizing reduced complexity rather than new principles. A sharper analysis isolating what specifically drives gains (FSQ vs. RVQ, layer choices, GAN post-training) would help.

Some prior works with similar ideas
- https://arxiv.org/abs/2402.08093
- https://arxiv.org/abs/2402.03407
- https://arxiv.org/abs/2309.00169 (RepCodec: a baseline directly related to quantizing speech SSL models, but was not compared here)
- https://arxiv.org/abs/2505.19273

# 2. Evaluation on clean data dominates + OOD robustness

Models are trained on LibriTTS and the main reconstruction is on LibriSpeech test-clean. OOD results (Table 14) show noticeable WER/CER degradation on noisy/emotional/unseen-language sets, sometimes with paradoxically higher UTMOS than ground truth. The authors themselves note sensitivity to dynamic background noise. This limits claims about realistic SLM readiness.

# 3. UTMOS and limited human evaluation

In reconstruction, ground-truth UTMOS is 4.07, while Kanade reaches 4.16 (Table 1), which is counterintuitive and suggestive of predictor bias. Also, the paper includes MUSHRA for reconstruction, but TTS quality is reported only with UTMOS, not human MOS/MUSHRA, despite being a core generative downstream task. This weakens quality claims for TTS.

# 4. Bitrate accounting may overstate efficiency for generative use.
The paper repeatedly uses marginal bitrate, explicitly excluding overheads like Kanade's global embedding. Yet the global embedding is required for resynthesis and is used in downstream setups, so an end-to-end bitrate/conditioning cost (including global) would better reflect the trade-offs.

# 5. Potential issues with downstream comparisons

- For ASR, RVQ baselines are evaluated with only the first layer (assumed to carry linguistic content). If lexical information is distributed across layers for some models, this setting may disadvantage them. The paper assumes the first layer suffices.

- For TTS, tokenizers without global embeddings are compensated by prepending reference audio streams, while tokenizers with global vectors use extracted globals. These differences can affect fairness and make it hard to attribute gains purely to token quality.

- GAN post-training likely contributes substantially to *quality* metrics. Ablations show notable changes when removing GAN post-training. However, the paper does not thoroughly quantify how much of the perceived quality/improvements versus baselines stem from this step, which other codecs may not share in this exact form.

# 6. Streaming and deployability limits

The authors acknowledge that tokens are not streamable due to a bidirectional SSL backbone; constant-rate tokens may be redundant; robustness to dynamic background noise is limited. These are important for SLMs in the wild.

**Questions:**

1. **Where exactly do the gains come from?** Please provide controlled ablations that isolate: FSQ vs. RVQ (single-layer RVQ), GAN post-training off vs. on, and precise SSL layer choices for both content and global branches. A compact table summarizing their individual contributions to WER/MUSHRA/UTMOS would help.

2. **Human evaluation for TTS.** Since Table 4 reports only UTMOS, please add a human MOS for TTS to substantiate quality claims and address predictor bias (e.g., your reconstruction UTMOS exceeds ground truth).

3. **Account for global embedding cost.** Provide end-to-end bitrate/conditioning overheads, including the global embedding (e.g., its size, transmission/conditioning cost in a realistic SLM/TTS pipeline) and re-plot rate-distortion curves under this.

4. **Robustness on noisy/emotional/unseen-language speech.** The OOD table reveals notable gaps. Please include targeted experiments with data augmentation/noise-robust training, streamable encoders, or variable-rate tokenization to show mitigation. Also report WER/MUSHRA changes under additive music/babble at different SNRs.

5. **Fairness of downstream baselines.**
    - ASR: Evaluate RVQ baselines with more than one layer (e.g., 1:2) where feasible, to test whether lexical information is indeed concentrated in layer 1 for those models.
    - TTS: Provide an alternative setup where all models use identical prompting/conditioning (e.g., everyone gets a fixed global vector estimated from the same reference frontend) so we can attribute differences to token quality rather than conditioning mechanics.

6. **SSL encoder training.** Kanade uses frozen WavLM features. Please clarify whether any fine-tuning of the SSL encoder was attempted. If not, could light fine-tuning destabilize disentanglement or improve robustness? A small study would answer this open question.

7. **Streaming & latency.** Since non-streamability is a stated limitation, can you prototype a streamable variant (chunked encoding) and report latency/quality trade-offs?

---

> ### Author Response · Authors · 2025-11-22
>
> Thank you for your review. We apologize for the wait, and hope we can address your concerns.
>
> **Comparison with prior work**: Based on your comments and those of other reviewers, we realized that we did not give a proper treatment to prior work on disentangled tokenizers, particularly in the main text. We have since rectified this and explained what is special about our approach in both the introduction:
> > Disentangled speech tokenizers aim to separate information into multiple streams, often time-varying content and acoustic invariants. Since many linguistically irrelevant features like speaker identity and microphone characteristics are constant, this allows the content stream to contain easily- accessible linguistic information, while relegating information necessary for reconstruction to a separate representation. Martin-Cortinas et al. (2024) (SSVC) have shown that using only the content stream can improve the performance of downstream language modeling. The authors hypothesize that this is because the model learns the content distribution, rather than a more complicated joint distribution of speaker information and content. We can avoid the tradeoff described above by preserving reconstruction information, but not feeding it to models that don’t need it. Most disentangled speech tokenizers use a multi-branch architecture along with at least one additional method to encourage disentanglement, such time invariance (TiCodec, Single-Codec, SoCodec), data augmentation (LSCodec), supervision (FACodec), or using pretrained models (FreeCodec).
>
> and in the conclusion:
> > Unlike most disentangled tokenizers, Kanade separates global information from content only by the architecture of the model itself. To our knowledge, only BiCodec (Wang et al., 2025a) has attempted this. However, it has a more complicated global branch and, as shown experiments, does not achieve disentanglement... Like RepCodec (Huang et al., 2024), Kanade uses only SSL features as input. But ablations show that end-to-end training with reconstruction loss better preserves prosody and speech quality than RepCodec’s 2-step approach. Using SSL features allows training with only 600 hours of data and 120M unfrozen parameters. Ye et al. (2025b) presented a single-layer codec with FSQ, but we show that disentangling the content yields tokens more suitable for TTS and SLMs. This reduces complexity, since there is no need for an additional step to produce finer tokens, and also allows the model to better attend to suprasegmental features.
>
> We also would like to comment on the prior work you suggested comparing with:
> - **Base TTS**: We mention Base TTS in passing in the related work section for its use of gradient reversal for disentanglement. It also requires contrastive loss. Our design uses its dual-branch architecture to restrict the flow of information and does not use either of these additional methods to encourage disentanglement. Unfortunately, we cannot compare the performance of Kanade with Base TTS because it is closed-source.
> - **SSVC**: The paper describing SSVC makes the same claim that we do: removing acoustic constants makes downstream modeling easier. We failed to realize this when we wrote the introduction, but have now updated it to reference this work. For disentanglement, SSVC uses gradient reversal and contrastive learning and and produces multiple layers of tokens using RVQ quantization, making it more complicated than our approach and less suitable for SLMs (its effective token rate is 200Hz when used as described in the paper). We could not find an open-source implementation of SSVC, so did not include it as a baseline.
> - **RepCodec**: We cited RepCodec as inspiration for the content branch design. It is not a disentangled tokenizer and does not train end-to-end with a reconstruction task. We showed in our ablations that training end-to-end with the reconstruction task is superior to a RepCodec-like 2-stage process.
> - **Eta-WavLM**: Eta-WavLM describes a way to linearly split WavLM into a speaker-erased embeddings $\eta$ and a single speaker embedding $d$. This is similar to the approach described by Kamper et al. in LinearVC. Given eta-WavLM features, we could optionally downsample and then discretize them for use in downstream tasks. We were excited about this approach, but were unfortunately unable to reproduce the results in the paper, despite the simplicity of the method. There is no open-source implementation.

---

> ### Author Response · Authors · 2025-11-22
>
> **Human Evaluation**: We did not include human evaluation for downstream generative tasks because we found that in our reconstruction results, MUSHRA correlation with UTMOS was quite high (spearman = 0.825).
>
> On UTMOS being higher than the ground truth: we subjectively found that reconstructed speech from Kanade and baselines such as StableCodec (UTMOS = 4.31) have less background noise. The models might be doing some denoising, which may explain cases where reconstruction scores are higher. We also observe that MUSHRA scores for X-Codec 2 are the same as the ground truth, so it is not impossible for model outputs to be rated as well or better than the ground truth.
>
> That being said, we are working to include more MUSHRA scores and will update the paper once we have results.
>
> **OOD Robustness**:  We have updated the OOD appendix to include baselines. These show that Kanade performs relatively well on out-of-domain data. As noted, it is possible for resynthesized audio to have better quality than the ground truth. In fact, the "w/ clean global emb." case was an attempt to use voice conversion as a denoising mechanism. This degrades WER but improves UTMOS.
>
> **Bitrate accounting**:  We have updated the main text to better explain our choice to use marginal bitrate. For a more thorough justification, please see "Marginal bitrate" in our response to LiTB.
>
> **ASR codebook truncation**:  We thought that it would help improve baseline performance if ASR did not include the most detailed acoustic layers. We tested all models again and found that including all RVQ layers yields the best results for all multi-layer models. We apologize for this oversight and have updated the downstream task result table (Table 2) with these new results. The relative rankings of the best models have not changed much.
>
> **Ablations**:
> - **GAN post-training**: We include ablations in the Appendix (GAN post-training ablation results). These show that post-training only slightly improves UTMOS. This is a case where we are slightly suspicious of UTMOS, so we are planning to collect MUSHRA scores for this ablation.
> - **FSQ**: We included an ablation in the appendix for FSQ vs VQ, but have moved it to the main text. We aimed to create a single-layer tokenizer, so did not experiment with RVQ.
> - **SSL layer choices**: (Mostly copy-pasted from response to AUD2) We chose SSL layers to test based on prior work\[1\] on the information available in each layer. While we tested a few layers (in "Additional Ablation Studies"), we did not conduct a systematic sweep because the configuration we used seemed to be sufficient. We agree that a systematic sweep could eke out better metrics, and while interesting, we do not think choosing the best layers would further support the validity of our approach. As the existing ablations show, our approach is not extremely sensitive to the choice of layers.
>
> **Human evaluation for TTS**: We are current retraining all TTS models for consistent conditioning, but will attempt to collect MUSHRA when they are ready.
>
> **Account for the global embedding cost**: For a thorough justification of our choice to use marginal bitrate, please see "Marginal bitrate" in our response to LiTB.
>
> ---
>
> *Edited to remove duplicate information from previous comment.*

---

> > ### Author Response · Authors · 2025-11-22
> >
> > **Robustness on noisy/emotional/unseen-language speech**: We have updated the OOD table to include baselines and updated the discussion. Gigaspeech contains static and dynamic background noise, and we show that Kanade performs better than any low- or medium- bitrate baseline and is not much worse than PAST, the best baseline (11.3% for Kanade vs 10.9% for PAST see Table 18).
> >
> > > Please include targeted experiments with data augmentation/noise-robust training
> >
> > We focus on providing a simple method for creating a speech tokenizer suitable for spoken language modeling. This method works quite well, *even on OOD data* without data augmentation or noise-robust training. However, we plan to experiment with these in the future to build a tokenizer with better denoising capabilities.
> >
> > > streamable encoders
> >
> > See "Streaming" below.
> >
> > > variable-rate tokenization...
> >
> > As mentioned in "Future work", we plan to adapt variable tokenization methods such as those described in Sylber\[4\] to either apply our disentanglement technique to their features or apply their distillation methods to our representations. Adding variable-rate tokenization to this paper (which is already relatively heavy) on top of the main design would confuse readers and provide little value (not to mention there is no space for it).
> >
> > > Also report WER/MUSHRA changes under additive music/babble at different SNRs.
> >
> > Though this might be nice, we already have results (from Gigaspeech) showing that Kanade is no less robust to comparable models to noise. We do not think this additional experiment would add much value.
> >
> > **ASR Fairness**:  We thought that it would help improve baseline performance if ASR did not include the most detailed acoustic layers. We tested all models again and found that including all RVQ layers yields the best results for all multi-layer models. We apologize for this oversight and have updated the downstream task result table (Table 2) with these new results. The relative rankings of the best models have not changed much.
> >
> > **TTS conditioning**: We can see how the difference in conditioning might cloud the results. We are currently retraining all the TTS models to use global conditioning from a pretrained speaker embedding model.
> >
> > **SSL Encoder fine-tuning**: While this could improve performance, we did not find it necessary. This requires more computational resources when training and complicates the training process.
> >
> > **Streaming**: As noted in future work, using the Kanade design for a streamable encoder is possible and desirable. However, methods for doing this are known\[3\] and streaming is not the focus of our work. It is possible to use Kanade in a quasi-streamable way (details are in the new appendix "Chunked encoding and decoding" and include a conservative latency estimate).
> >
> > \[1\] A. Pasad, B. Shi and K. Livescu, "Comparative Layer-Wise Analysis of Self-Supervised Speech Models," _ICASSP 2023 - 2023 IEEE International Conference on Acoustics, Speech and Signal Processing (ICASSP)_, Rhodes Island, Greece, 2023, pp. 1-5.
> >
> > \[2\] Á. Martín-Cortinas _et al._, “Enhancing the Stability of LLM-based Speech Generation Systems through Self-Supervised Representations,” Feb. 05, 2024, _arXiv_: arXiv:2402.03407.
> >
> > \[3\] K. Choi, M. Someki, E. Strubell, and S. Watanabe, “On-device Streaming Discrete Speech Units,” presented at the Proc. Interspeech 2025, 2025, pp. 4423–4427.
> >
> > \[4\]  C. J. Cho _et al._, “Sylber: Syllabic Embedding Representation of Speech from Raw Audio,” presented at the The Thirteenth International Conference on Learning Representations, Oct. 2024.

---

> ### Comment · Reviewer_ihXt · 2025-11-25
>
> Thanks for the response. However, I keep my score of 2 (reject) because the current draft relies on invalid or missing data.
>
> 1. **TTS experiments are incomplete:** You acknowledged that the TTS results in Table 3 are clouded by unfair conditioning and stated you are *currently retraining all the TTS models*. I can't evaluate based on experiments that are still running. Without these corrected results, the performance claims are unsupported.
>
> 2. **Human evaluation is missing:** While you offered to *collect MUSHRA scores for the retrained models*, this data is currently missing. Human evaluation is standard and necessary for this domain to validate perceptual quality.
>
> 3. **Incremental novelty:** The proposed architecture (SSL features + VQ-VAE + Global Branch) is an incremental combination of existing components. Without the validated empirical advantages in TTS (which are currently being retrained), the technical contribution is insufficient for acceptance.
>
> Because key experiments are retraining and human evaluation is still pending, the work is not yet ready. I encourage you to complete these experiments for a future submission.

---

### Official Review · Reviewer_AUD2 · 2025-10-31

**Soundness:** 2
**Presentation:** 2
**Contribution:** 3
**Rating:** 4
**Confidence:** 3

**Summary:**

This paper introduces Kanade, which is a speech tokenizer. Kanade contains a content branch and a global branch that are aimed at capturing phonetic features and speaker characteristics, respectively. Trained on just 586 hours of LibriTTS data using WavLM features, Kanade achieves strong performance on downstream tasks, while maintaining competitive reconstruction quality at low bitrates.

**Strengths:**

The paper addresses a timely and important problem as spoken language models become more prominent. Efficient speech tokenizers will be important for next-generation multimodal language models. The work has clear practical relevance given the growing interest in speech-based LLMs for applications such as voice assistants. The proposed two-branch architecture presents a practical solution that achieves competitive performance across multiple metrics while using only modest amounts of training data.

**Weaknesses:**

The introduction could benefit from a more direct problem statement. While it establishes the trade-offs current tokenizers face, a clearer explanation of why existing solutions are insufficient before presenting Kanade would strengthen the narrative.

The ablation study for SSL feature layer selection appears non-systematic, as it tests only some layers for content, and fails to comprehensively evaluate layer combinations for the global branch. A systematic sweep would be necessary to validate the chosen layers. The ablation results should be in the main text rather than appendices, as they directly impact core architectural decisions.

The paper frequently references specific experimental results in Section 2 before introducing the experimental setup in Section 3. This forward-referencing forces readers to jump ahead to understand the evidence behind design choices. The method section should either avoid citing specific results or preferably be reorganized for better readability.

Training only on LibriTTS raises questions about generalization. Also, the main evaluation uses mostly read speech datasets, which may not reflect real-world performance on spontaneous speech that spoken language models would need to handle.

Claiming SoTA performance seems not to be substantiated by all presented results, as the scores on many performance metrics appear close between models. Consider adding significance tests or soften claims.

**Questions:**

Could the authors motivate the choice for WAVLM as the SSL encoder instead of other available SSL encoders?

Could the authors provide more information about the SSL feature layer selection? Specifically, could the authors report a grid/sweep over candidate layers?

The goal of the global branch is to capture speaker characteristics and the authors write that the global branch captures information about the audio that does not change over time. However, many speaker characteristics (pitch, style, emotion) do vary within and across utterances. Could the authors clarify this design decision and contextualize the effect of producing a global embedding only?

To validate that the results generalize beyond read speech, could the authors evaluate their method (e.g., on ASR) from any established corpus containing spontaneous conversational speech? This would strengthen claims about suitability for spoken language modeling, which must handle diverse real-world speech.

In the ASR evaluation, RVQ-based models use only their first layer while FACodec uses both of its content layers. Could the authors clarify this methodological choice and provide additional results with a per layer RVQ ablation?

---

> ### Author Response · Authors · 2025-11-22
>
> Thank you for your thoughtful review. We apologize for taking so long to reply.
>
> ---
>
> Below, we address your concerns.
>
> **Problem statement**: We agree that our introduction was vague about how our model is different from existing work. We have updated the introduction and added the following discussion to the conclusion:
> > Unlike most disentangled tokenizers, Kanade separates global information from content only by the architecture of the model itself. To our knowledge, only BiCodec (Wang et al., 2025a) has attempted this. However, it has a more complicated global branch and, as shown experiments, does not achieve disentanglement... Like RepCodec (Huang et al., 2024), Kanade uses only SSL features as input. But ablations show that end-to-end training with reconstruction loss better preserves prosody and speech quality than RepCodec’s 2-step approach. Using SSL features allows training with only 600 hours of data and 120M unfrozen parameters. Ye et al. (2025b) presented a single-layer codec with FSQ, but we show that disentangling the content yields tokens more suitable for TTS and SLMs. This reduces complexity, since there is no need for an additional step to produce finer tokens, and also allows the model to better attend to suprasegmental features.
>
> **Missing layer ablations**: We chose SSL layers to test based on prior work\[1\] on the information available in each layer. While we tested a few layers as you noted, we did not conduct a systematic sweep because the configuration we used seemed to be sufficient. We agree that a systematic sweep could eke out better metrics, and while interesting, we do not think choosing the best layers would further support the validity of our approach. As the existing ablations show, our approach is not extremely sensitive to layer selection.
>
> **Ablations in the main text**: We included some ablations in the main text, but decided to only include those that highlight architecture choices that were particularly important for the success of our approach, but are not necessarily intuitive. We chose to use deep layers for content and shallow layers for global content based on prior work, rather than experimentation. Since the results were consistent with the literature, we do not think including layer ablations in the main text is necessary. However, due to your comment, we reconsidered the set of main ablations and decided to move the FSQ/VQ ablation into the main table, since it shows that FSQ greatly contributed to the success of our method.
>
> **Forward-referencing in the method**: There was one instance where the results were used to justify the completeness of WavLM features, but we have updated the text to reference the literature instead.
>
> The method section also includes many references to ablations. One aim of our paper is to motivate both the specific disentanglement approach we took and the architectural decisions we made, with the goal of providing a good foundation for future work. Each decision was based on theory or experimentation, and we point to ablations in all cases of the latter. We think interleaving the reasons we made choices with the method is the best way to teach readers about our thought process.
>
> **Concerns about generalization**: Kanade is trained on only about 600 hours of English read speech. We have expanded the section on OOD reconstruction and included baselines. These experiments show that Kanade performs competitively with other models trained on much more data. This paper is a recipe for a simple speech tokenizer that achieves good disentanglement and performance with a very small amount of data and compute. It can, and should, be trained on a more diverse dataset for practical use.
>
> > To validate that the results generalize beyond read speech, could the authors evaluate their method (e.g., on ASR) from any established corpus containing spontaneous conversational speech?
>
> We plan to include OOD ASR using the Switchboard corpus, but have not yet retrained the relevant downstream models to compute the metrics. We ask for your patience while we do so.
>
> **SOTA claims**: We claim state-of-the-art performance when Kanade performs better or *as well as* the best baselines. This is consistent with its usage in our experience reading other works. We do not claim to "set a new state of the art" or "surpass the state of the art". The objective metrics are calculated on large number of samples and aggressively rounded, so we do not think it is necessary to include significance tests for them. However, we do think that the subjective listening test needs confidence bounds. Unfortunately, it looks like getting this right is non-trivial\[3\], so we ask for some time to figure out exactly how to report this correctly.

---

> ### Author Response · Authors · 2025-11-22
>
> We also want to answer your questions (where they do not overlap with the concerns outline above):
>
> > Could the authors motivate the choice for WAVLM as the SSL encoder instead of other available SSL encoders?
>
> We chose WavLM because\[1\] showed that layer 9 had very high levels of similarity with phone labels and \[2\] showed that WavLM features are easily separable into speaker and non-speaker components. HuBERT is tested in the "Backbone and SSL encoder ablation results" table in the appendix, but we suspect that Wav2Vec2-BERT (used in X-Codec 2) or other SSL models may perform better, particularly in multilingual settings.  Similarly to our layer choices (see "Missing layer ablations" above), we do not think that the exact SSL encoder was an important part of our design, so relegated it to the (single) ablation to the appendix.
>
> > The goal of the global branch is to capture speaker characteristics and the authors write that the global branch captures information about the audio that does not change over time. However, many speaker characteristics (pitch, style, emotion) do vary within and across utterances. Could the authors clarify this design decision and contextualize the effect of producing a global embedding only?
>
> As you indicate, pitch, style, and emotion can vary within utterances. Therefore, they should be encoded as content. We know that relative pitch is encoded in the content based on the voice conversion results. However, as you point out above, training on a limited dataset of read English speech may have allowed Kanade to encode some elements of style, emotion, and dynamic range in the global embedding.
>
> We have done new OOD experiments on the [Salmon](https://pages.cs.huji.ac.il/adiyoss-lab/salmon/) sentiment consistency dataset (full results in the updated manuscript) to show what happens when the emotion changes within an utterance. The results surprisingly show that both disentangled tokenizers like Kanade and codecs without a global embedding slightly degrade over almost all metrics. Based on this result, Kanade is not more susceptible to changes in style within utterances than non-disentangled tokenizers. As additional evidence of robustness to within-utterance variation, we have also add a chunked resynthesis experiment in the appendix. It shows that 25-35 minutes of speech can be resynthesized well using a single global embedding.
>
> > In the ASR evaluation, RVQ-based models use only their first layer while FACodec uses both of its content layers. Could the authors clarify this methodological choice and provide additional results with a per layer RVQ ablation?
>
> We thought that it would help improve baseline performance if ASR did not include the most detailed acoustic layers. We tested all models again and found that including all RVQ layers yields the best results for all multi-layer models. We apologize for this oversight and have updated the downstream task result table (Table 2) with these new results. The relative rankings of the best models have not changed much.
>
> \[1\] A. Pasad, B. Shi and K. Livescu, "Comparative Layer-Wise Analysis of Self-Supervised Speech Models," _ICASSP 2023 - 2023 IEEE International Conference on Acoustics, Speech and Signal Processing (ICASSP)_, Rhodes Island, Greece, 2023, pp. 1-5.
>
> \[2\]
> H. Kamper, B. van Niekerk, J. Zaïdi, and M.-A. Carbonneau, “LinearVC: Linear transformations of self-supervised features through the lens of voice conversion,” June 02, 2025, _arXiv_: arXiv:2506.01510. doi: [10.48550/arXiv.2506.01510](https://doi.org/10.48550/arXiv.2506.01510).
>
> \[3\] C. Mendonça and S. Delikaris-Manias, “Statistical tests with MUSHRA data,” in _Audio engineering society convention 144_, Audio Engineering Society, May 2018.
>
> ---
>
> *Edited section on OOD for clearer wording*

---

### Official Review · Reviewer_AYrk · 2025-11-01

**Soundness:** 3
**Presentation:** 4
**Contribution:** 2
**Rating:** 4
**Confidence:** 4

**Summary:**

This work present a new speech tokenizer named Kanade designed for speech language models. They use a SSL encoder (WavLM) to extract features and partition the features in different layers into two branches, one for content and the other for speaker. They use a SSL reconstruction and a Mel reconstruction loss to train the model and add a GAN post training to get fine-grained spectrogram. They compare Kanade with many speech codecs, and evaluate generations on WER, quality, speaker identity and prosody. Reconstruction results show that Kanade can achieve comparable or better results with lower bitrate. They also have ASR, speaker and emotion recognition, TTS and VC as downstream tasks to further show the strong ability of Kanade.

**Strengths:**

1. The experiments are thorough and extensive. Besides the model and speech token itself, this paper presents a comprehensive evaluation protocol of speech tokenizers, including intelligibility, quality, speaker identity, prosody, and some discriminative and generative tasks. Ablation studies and analysis provide clear evidence for the design choices of the speech tokenizer.
2.  The proposed speech tokenizer can achieve high-quality speech generation with just small amount of data and lower bitrate compared to most existing methods.

**Weaknesses:**

1. In the abstract and introduction of the paper, the authors present the proposed speech tokenizer as an important advance for spoken language modeling results. However, SLMs results are put in appendix and show Kanade underperforms SSL-distilled codecs (SpeechTokenizer, Mimi) on both sWUGGY and sBLIMP, which are important metrics for SLM evaluations. And there are a lot of other SLM evaluation metrics which are not mentioned or considered here.

2. Missing comparisons with some existing work speech tokenization such as dmel (https://arxiv.org/abs/2407.15835) on basic generative and discriminative tasks. Also, while the author mentioned PAST (https://www.arxiv.org/pdf/2505.14470), Sylber (https://arxiv.org/abs/2410.07168) and SyllableLM (https://arxiv.org/abs/2410.04029) in the related work, there is no comparison with them. They achieve impressive results in SLMs.

3. The proposed speech tokenizer is heavily dependent on some pretrained models, and is a relative straightforward fusion of existing ideas. With the incremental model side innovation, I expect better results in SLMs as the author claimed.

**Questions:**

1. How do you prove features from different layers contain content and speaker information? Table 3 why using both representations leads to worse performance than using just the global embedding? As the way you partition the features is pretty hard, will there be any leakage of the linguistic information?

2. Can you provide more concrete experiments/evidences why Kanade can lead to better SLMs despite underperforming on sWUGGY/sBLIMP? What model sizes did you train for SLMs? Maybe larger models can lead to more clear gaps?

---

> ### Author Response · Authors · 2025-11-22
>
> Thank you for your thoughtful review and suggestions. We apologize for the relatively late response.
>
> ---
>
> First, we would like to address your questions:
> > How do you prove features from different layers contain content and speaker information?
>
> We're not sure whether you mean the input SSL features or the encoded representations from Kanade. If you mean SSL features, the first paragraph in the method provides the motivation. While all layers contain some speaker and content information, Pasad et al. (2023)\[2\] among others, show while shallow layers encode more acoustic features, deeper layers encode more content.  If you mean Kanade's representations, the voice conversion, length generalization, chunked resynthesis (new, in the appendix), and speaker discrimination results all show that we have achieved good disentanglement.
>
> >  Table 3 why using both representations leads to worse performance than using just the global embedding?
>
> In Kanade, we aimed for excellent disentanglement so that downstream models could ingest only the information relevant to their tasks. The models that we used for speaker and emotion recognition are not particularly strong, so likely benefitted from seeing only the global embedding, which has few bits but is high in relevant information. We have updated the results to include global-only results from other disentangling tokenizers. BiCodec as Kanade exhibits the same pattern with all metrics improving when using only the global embedding.
>
> Emotion recognition by its nature should be based on content, but previous work\[1\] shows that low-level acoustic features are surprisingly effective for this task. I would suspect that, in contrast with our results, a very strong model for emotion recognition would be able to make use of both representations in a dataset that reflects real-world emotions in speech.
>
> > As the way you partition the features is pretty hard, will there be any leakage of the linguistic information?
>
> We're not sure what you mean by this. Could you elaborate?
>
> > Can you provide more concrete experiments/evidences why Kanade can lead to better SLMs despite underperforming on sWUGGY/sBLIMP? What model sizes did you train for SLMs? Maybe larger models can lead to more clear gaps?
>
> We trained a larger model using the [*slam*](https://pages.cs.huji.ac.il/adiyoss-lab/slamming/) recipe using one epoch of LibriLight and have moved the results to the main text. They show that Kanade closes the gap we saw with a weaker model, but not that it outperforms. We would like to train an even larger model, but do not currently have a larger dataset to do this (and don't want to muddy things by including synthetic data or preference optimization).
>
> ---
>
> Below we also address some of your concerns.
>
> **Missing SLM metrics**: We have added tStoryCloze and sStoryCloze, which are common in recent works. Is there another metric that you think would be enlightening, or is this enough?
>
> **Missing baselines**:
> - dMel: We did consider including dMel, but it has very different goals. It makes no attempt to compress the speech stream to make downstream modeling easier and does not attempt disentanglement. While its discriminative results (ASR WER 9.0%, ASR CER 4.6%, SID Acc 69.6%, ASV EER 10.7%, and ER Acc 55.3%) good (and ASV EER was better than Kanade when content tokens were included), its unique design makes it difficult to fairly compare in the generative modeling experiments (it has 80 layers of tokens).
> - PAST: Not including PAST, which is also meant for SLMs, was an oversight. We have added reconstruction and discriminative metrics, and are working to calculate results for spoken language modeling and TTS.
> - Sylber / SyllableLM: These methods take advantage of the fact that speech is inherently variable-bitrate (for example, vowels are much longer than short consonants like a flapped r), but are based on existing fixed-rate representations. Therefore, we think that variable-rate tokenization is complementary to our work. In fact, as mentioned in "Future work", we plan to adapt these methods to either apply our disentanglement technique to their features or apply their distillation methods to our representations. Therefore, we opted to only compare fixed-rate tokenizers.
>
> \[1\] Y. Terraf and Y. Iraqi, “LANCET: Lightweight Attention-Enhanced Network for Robust Speech Emotion Recognition,” in _2025 33rd European Signal Processing Conference (EUSIPCO)_, Sept. 2025, pp. 371–375.
>
> \[2\] A. Pasad, B. Shi and K. Livescu, "Comparative Layer-Wise Analysis of Self-Supervised Speech Models," _ICASSP 2023 - 2023 IEEE International Conference on Acoustics, Speech and Signal Processing (ICASSP)_, Rhodes Island, Greece, 2023, pp. 1-5.

---

### Official Review · Reviewer_xExe · 2025-11-04

**Soundness:** 2
**Presentation:** 3
**Contribution:** 2
**Rating:** 2
**Confidence:** 4

**Summary:**

This paper introduces **Kanade**, a speech tokenizer designed for Spoken Language Models (SLMs). Kanade uses a two-branch architecture that separates linguistic content (phonetic + prosody) from time-invariant acoustic characteristics (speaker identity, recording conditions). The content branch quantizes deep SSL features (WavLM) into discrete tokens at 25Hz using FSQ, while the global branch extracts a single continuous embedding from shallow layers for reconstruction. The paper demonstrates SOTA performance in speaker disentanglement and linguistic richness while maintaining low bitrate (250 bps) and high reconstruction quality.

**Strengths:**

1. **Well-motivated architecture**: The two-branch design elegantly addresses the fundamental challenge of separating linguistic content from acoustic characteristics. Using deep vs shallow SSL layers is principled and grounded in prior understanding of SSL representations.

2. **Comprehensive evaluation**: The paper includes extensive experiments covering reconstruction quality, speaker disentanglement, linguistic availability (phoneme and prosody), and cross-corpus generalization. The metrics are well-chosen and appropriate.

3. **Compact representation**: Achieving single-stream discrete tokens at 25Hz is a significant advantage for downstream LM efficiency compared to multi-stream RVQ codecs.

**Weaknesses:**

1. **Insufficient SLM experiments**: This is the most critical weakness. The paper is explicitly motivated for "Spoken Language Models" and positions Kanade as a tokenizer designed FOR SLMs, yet provides minimal experimental validation for actual language modeling. The main paper includes no SLM experiments at all. The only language modeling results appear in Appendix Table 13, which shows preliminary next-token prediction experiments on a small LibriSpeech dataset. Critically, these results demonstrate significantly worse performance compared to existing published speech LMs such as Moshi and Kimi-Audio, raising serious concerns about whether Kanade tokens are actually suitable for the proposed use case. Without convincing evidence that these tokens work well for autoregressive language modeling, the paper's core contribution remains unvalidated.

2. **Questionable SOTA claims and insufficient experimental details in Table 4**: The TTS experiments in Table 4 lack transparency and fairness, making it difficult to assess the validity of the reported results:
	1. **Missing training details**: The paper provides no information about what data were used to train the models in Table 4, nor details about model architecture, parameter counts, or training procedures.
	2. **Unfair comparison across tokenizers**: Different speech tokens are optimized for different modeling architectures. Using a single architecture optimized for Kanade to evaluate all baselines creates an unfair comparison. For example, Mimi uses a depth transformer specifically designed for RVQ decoding—was this architecture used in Table 4, or were all baselines forced into Kanade's decoder?
	3. **Lack of baseline context**: Table 4 includes no comparison with published or publicly released TTS models. Without established benchmarks, it is difficult to interpret whether 4.7% WER represents strong performance, especially given that the test set appears to be newly created by this submission rather than using widely adopted benchmarks like seed-tts-eval (https://github.com/BytedanceSpeech/seed-tts-eval).

3. **Missing critical related work and insufficient contextualization**: The paper fails to cite and compare with several highly relevant recent works, and tables lack proper contextualization with SOTA baselines:
	1. **SparkTTS** (https://arxiv.org/pdf/2503.01710): Introduces a new speech tokenizer and demonstrates how to adapt a text-only LLM with speech tokens to enable speech generation—directly relevant to the SLM motivation.
	2. **dMel** (https://arxiv.org/abs/2407.15835): Proposes a tokenizer-free discrete speech representation that can be jointly modeled with text tokens in transformer-decoder architectures for both TTS and ASR.
4. **Insufficient baseline contextualization**: All downstream evaluation tables (Tables 2, 4, 5) omit comparisons with SOTA task-specific models. For instance:
		- Table 2 should report the latest SOTA ASR WER, speaker verification results, and emotion recognition accuracy to help readers understand where Kanade's performance stands relative to specialized models (not necessarily to claim superiority, but to provide context).
		- Table 4 should include results from existing TTS models for reference.
		- Table 5 should include voice conversion results from open-source models to establish performance context.


Overall, my concern is about the experiments:
- use the submission's own test set and their own model, without using standard benchmark and opensourced models
- no experiments about SLM in the main body. Only find minimal exploration in appendix. This is not aligned with the abstract and the introduction of this submission

**Questions:**

- How robust the proposed speech token is? For example, can it reconstructed speech with noise background as described in dMel paper(https://arxiv.org/abs/2407.15835)?

- Can you evaluate the TTS model with seed-tts-eval benchmark (https://github.com/BytedanceSpeech/seed-tts-eval) or Emergent-TTS-Benchmark (https://arxiv.org/abs/2505.23009)? otherwise hard to understand the results with your own test set.

---

> ### Author Response · Authors · 2025-11-22
>
> Thank you for your thoughtful review and suggestions. We apologize for the relatively late response.
>
> ---
>
> **SLM experiments**: We ran additional experiments using the [*slam*](https://pages.cs.huji.ac.il/adiyoss-lab/slamming/) recipe and have moved the results to the main text. The results show that Kanade is competitive with k-means and distilled hybrid codecs. We are still working to compute results for old and new baselines.
>
> > Critically, these results demonstrate significantly worse performance compared to existing published speech LMs such as Moshi and Kimi-Audio
>
> Even the new results will not compare favorably with published results for highly fine-tuned SLMs. It is not our goal to report SOTA metrics on SLMs, but to compare tokenizers.
>
> **TTS experiments**
> - **Training details** were provided in the main text with further details in Appendix E.3 Downstream model configurations. The TTS models are 12-layer, 12-head, 768 LLaMA-style transformers with 85M parameters (excluding projection for input/output) trained on LibriTTS. For tokenizers with multiple layers, these layers are flattened and embedded into a single vocabulary. Inputs and outputs are detailed in the appendix.
> - **Were all baselines forced into Kanade's Decoder?** No, RVQ tokens are generated by autoregressive modeling of the flattened token stream and then decoded using their decoders.
> - **Model architecture (why not depth transformer?)**: This is reasonable to question, and our explanation was vague. We have updated the manuscript to be clearer about why flattening is a fair way to compare with RVQ-based tokenizers (citations replaced by model names):
>   > To generate audio for RVQ-based decoders, we need to generate multiple layers of dependent tokens. Autoregressive modeling produces the highest quality results when tokens are flattened into a single token stream (MusicGen). While some works cite scalability and performance as reasons to use additional modeling step instead (VALL-E, Moshi), we chose to standardize on flattening for its topline synthesis quality and simplicity.
> - **Lack of baseline context**: See **Context from specialized models** below.
>
> **Missing critical related work**:
> - SparkTTS describes BiCodec, which is included as a baseline and compared with Kanade. We have also updated the manuscript with additional contextualization in the introduction and conclusion.
> - dMel: (The following is copied from our response to AYrk, who also wondered about dMel.) We did consider including dMel, but it has very different goals. It makes no attempt to compress the speech stream to make downstream modeling easier and does not attempt disentanglement. While its discriminative results (ASR WER 9.0%, ASR CER 4.6%, SID Acc 69.6%, ASV EER 10.7%, and ER Acc 55.3%) good (and ASV EER was better than Kanade when content tokens were included), its unique design makes it difficult to fairly compare in the generative modeling experiments (it has 80 layers of tokens).
>
> **Robustness**
> > How robust the proposed speech token is? For example, can it reconstructed speech with noise background as described in dMel paper
>
> The Gigaspeech dataset used in OOD experiments includes a variety of background noise conditions, including static and dynamic background noise. We have also expanded the appendix with these experiments to include baselines to contextualize the OOD results.
>
> **Context from specialized models**: We have updated the downstream task result table with metrics from specialized models as reported in their papers. The VC table already had metrics from specialized methods including LinearVC, FreeVC, and CosyVoice 2.
>
> > Can you evaluate the TTS model with seed-tts-eval benchmark ([https://github.com/BytedanceSpeech/seed-tts-eval](https://github.com/BytedanceSpeech/seed-tts-eval)) or Emergent-TTS-Benchmark ([https://arxiv.org/abs/2505.23009](https://arxiv.org/abs/2505.23009))? otherwise hard to understand the results with your own test set.
>
> It looks like seed-tts-eval provides a subset of Common Voice and uses Whisper-large-v3 to calculate WER. We use LibriTTS test-clean because it is in-domain for Kanade and all baselines. While many works (for example, VALL-E, WER 5.9%) use LibriSpeech test-clean, we chose to use LibriTTS because of its higher 24 kHz sampling rate. LibriTTS test-clean has been used for evaluation in prior work (for example, CosyVoice, 3.17%) . We also use parakeet to evaluate WER across all metrics since it has weaker language modeling capabilities and so should be more sensitive to poor synthesis.
>
> However, it is not difficult to report numbers for Seed-TTS. We are currently retraining all TTS models to use consistent global conditioning, so will report back when we have updated results.
>
> ---
>
> *This post was edited to correct test-clean-100 to LibriTTS test-clean and for clarity in the section justifying its use.*

---

### Official Review · Reviewer_yxWZ · 2025-11-08

**Soundness:** 3
**Presentation:** 2
**Contribution:** 3
**Rating:** 6
**Confidence:** 4

**Summary:**

The paper presents Kanade, a speech tokenizer which aims to produce compact, linguistically rich, and speaker-disentangled speech representations. It achieves this through a dual-branch encoder comprising a content branch which captures linguistic and prosodic information and a global branch which captures suprasegmental features such as speaker style and channel characteristics. The content branch would be responsible for tokenizing in the manner similar to a text encoder, while the global branch would separate the speech-specific context. Content branch employs a zero-mean, unit-variance normalization, a transformer encoder with local-window attention and strided convolution for downsampling followed by quantization. Global branch uses ECAPA-TDNN with ConvNeXt and then Adaptive stats pooling for a single global vector. The model is trained with a joint mel-spectrogram reconstruction loss and an SSL feature reconstruction loss to balance prosody, acoustic detail, and linguistic fidelity. The paper additionally employs some design choices that strengthen the work, including Finite Scalar Quantization (FSQ) for quantization to prevent collapse, conditioning the mel spectrogram generation on the global embedding via AdaLN-Zero, and GAN post training to ensure the reconstructed speech quality is unaffected.

The results show positive performance on both generative and discriminative speech tasks. The paper also includes an extensive suite of ablations in the supplementary section. Overall, the paper presents a clean architecture combining dual-branch encoding, scalar quantization, and joint Mel+SSL objectives into a stable and compact tokenizer suitable for spoken-language modeling.

**Strengths:**

* The paper presents a clean separation between linguistic (content) and acoustic/speaker (global) factors. This design makes both empirical and conceptual sense, as demonstrated by the experimental results.
* The identification of Finite Scalar Quantization (FSQ) as an efficient and stable compression alternative is a notable contribution. FSQ avoids codebook collapse while achieving extremely low bitrates and maintaining high reconstruction quality. This is arguably the paper’s strongest technical contribution.
* The inclusion of a GAN-based post-training phase enhances perceptual quality. While not novel, it demonstrates a high level of engineering completeness.
* The paper employs creative and well-designed visualizations to illustrate evaluation metrics (Figures 1 and 4), effectively supporting its empirical claims.
* Paper includes an extensive supplementary section include extensive ablations, design rationales, and component analyses, reflecting thoroughness and careful empirical validation.

**Weaknesses:**

* The dual-branch architecture (content + global) is claimed to be the first and hence major contribution. However, this approach is not new - prior works such as FACodec (Chen et al., 2023) already separate linguistic and speaker/prosodic factors as content and global branches.
* The paper does not clearly describe whether the content branch alone functions as the tokenizer at inference time. This obscures the model’s practical usage and efficiency trade-offs.
* The value for loss balancing coefficient α between the mel-spectrogram and SSL feature losses is not reported. Since α directly governs the trade-off between prosody and linguistic fidelity, including a sensitivity study on α would benefit the paper.
* The paper suffers from over-reliance on supplementary material. Most of the essential information such as related work and baseline definition is deferred to the supplementary when they should appear in the main paper.
* The paper would benefit from examples or discussions of failure cases or degradation patterns, providing insight into when and why Kanade struggles (e.g., prosodically rich or noisy speech).

**Questions:**

Questions:
1. The paper mentions “superficial”  features but does not define what counts as “superficial.” Could you clarify this?
2. During inference, is the content branch alone used as the tokenizer? What does the full tokenization inference loop look like?

Suggestions:
1. The work would be significantly strengthened by explicitly juxtaposing its design and motivations with earlier dual-branch and global-conditioning frameworks, clarifying what is architecturally or functionally distinct about Kanade's formulation.
2. Figure (2) shows other contemporary and prior approaches. The figure makes it seem like Kanade is the only approach with a global branching. Please consider expanding the image to include prior works with similar branching so you can highlight key differences with your method.

---

> ### Author Response · Authors · 2025-11-22
>
> Thank you for your thoughtful review and suggestions. We apologize for the late response.
>
> ---
>
> First, we'd like to answer your questions:
> > The paper mentions “superficial” features but does not define what counts as “superficial.” Could you clarify this?
>
> This wording was used in the introduction to indicate features such as channel or environment properties that are irrelevant to the linguistic content of the track, such as microphone properties, speaker characteristics, and background noise. This vague wording is unhelpful, so we have updated the manuscript to avoid it.
>
> > During inference, is the content branch alone used as the tokenizer? What does the full tokenization inference loop look like?
>
> We did a poor job explaining how a disentangled tokenizer like Kanade is used for downstream tasks. The introduction has been updated with the following content:
> > At inference time, it usually suffices to use only the content branch. Kanade’s state-of-the-art disentanglement ensures that the global embedding does not encode content, so it does not need to be calculated for ASR or intent classification. Generative models like SLMs only need to generate a content stream, which can then be decoded to speech using a single baked-in global embedding. This fixed embedding might be considered the “voice” of the model.
>
> We have also updated the speaker-related discriminative tasks section to more clearly show that leaving out the right kinds of information improves downstream modeling.
>
> ---
>
> We'd also like to address some of the weaknesses you pointed out:
>
> **Novelty of the dual-branch design**: We agree that the dual-branch design is not new, but failed to properly explain this in the main text. We have updated the introduction with the following (citations replaced by model names):
> > Disentangled speech tokenizers aim to separate information into multiple streams, often time-varying content and acoustic invariants. Since many linguistically irrelevant features like speaker identity and microphone characteristics are constant, this allows the content stream to contain easily- accessible linguistic information, while relegating information necessary for reconstruction to a separate representation. Martin-Cortinas et al. (2024) (SSVC) have shown that using only the content stream can improve the performance of downstream language modeling. The authors hypothesize that this is because the model learns the content distribution, rather than a more complicated joint distribution of speaker information and content. We can avoid the tradeoff described above by preserving reconstruction information, but not feeding it to models that don’t need it. Most disentangled speech tokenizers use a multi-branch architecture along with at least one additional method to encourage disentanglement, such time invariance (TiCodec, Single-Codec, SoCodec), data augmentation (LSCodec), supervision (FACodec), or using pretrained models (FreeCodec).
>
> The data distribution, which Figure 2 shows, is not completely unique to Kanade. All speech tokenizers that use a fixed length encoding for global features will look roughly like this. The original figure was misleading. We apologize for this and have updated it.
>
> A large part of our argument is that disentanglement is important: for linguistic tasks, you use linguistic representations and for non-linguistic tasks, you can ignore those same representations. As we show in the experiments, Kanade achieves the best disentanglement among disentangling baselines.
>
> We have also included a discussion section in the conclusion to explicitly point out what we have contributed.
>
> **Loss coefficient $\mathbf{\alpha}$**: This is given under **Training Details** in section E.1:
> > The SSL feature and mel-spectrogram reconstruction losses are weighted equally ($\alpha = 1$).
>
> This value worked well enough for earlier experiments with other architectures, and we did not adjust it in experiments with Kande as presented here. However, as you point out, it might be possible to adjust the performance characteristics of the model by tuning alpha. I think such an experiment would be helpful, but given the limited time in the rebuttal period, we've decided not to run this particular experiment at this time.
>
> **Too much supplementary material**: We agree that there is a lot of supplementary material. The introduction contains a brief summary of related work, but almost completely ignored the related work in disentangled speech tokenizers. This has since been rectified with the inclusion of the paragraph on other dual-branch designs and by updating Figure 2. SLM results have also been updated and moved to the main text.
>
> **Failure cases**: We have expanded the OOD analysis in the appendix to include baselines and and experiment with utterances with inconsistent style. Please let us know if you have any lingering concerns.

---

> > ### Comment · Reviewer_yxWZ · 2025-11-23
> >
> > I thank the authors for the response and the updates. The paper is now somewhat more linear in content organization, clearer in its writing, and the empirical additions, particularly the global vs. content breakdown for Kanade and the baselines in Table 2, help clarify the behavior of the disentanglement.
> >
> > However, despite these improvements, some core issues that motivated my concerns remain unresolved, particularly around overall coherence, structural quality, and the concreteness of the contribution framing. In my initial review, I noted that the dual-branch design was presented as the primary novelty, even though similar architectures exist in prior work (e.g., FACodec). The revised text makes it clearer that the actual distinguishing factor is the use of a lightweight local-global architectural split instead of the multi-stage or multi-branch disentanglement mechanisms seen in prior work. While this clarification is welcome, it remains implicit and is not presented in a coherent, concrete, or prominent way. The paper still requires the reader to infer the true contribution rather than having it stated clearly.
> >
> > The introduction and contributions section, which should guide the reader, are still comparatively weak, vague, and structurally muddled. Although the conclusion now articulates the contribution more clearly, this explanation arrives too late. The narrative continues to lack the crispness, clarity, and rigor expected for a top-tier publication. Ultimately, the framing still feels incoherent, and the paper does not communicate its novelty or significance as well as it could.
> >
> > Given this, *my recommendation remains unchanged*. It is clear that some contribution meaningfully distinguishes this work from previous closely related disentangled tokenizers. While the contribution could be viewed as incremental, it is still innovative in its simplicity and design, and there is value in demonstrating that a minimal architecture can achieve disentanglement on par with or better than more complex counterparts. However, given the persistent issues with coherence, structure, clarity, and overall paper quality, I would not strongly vouch for it in its current form - The paper would benefit from further refinement to be presented in a top-tier venue.
> >
> > Recommendations for Revision:
> > 1. Make the contributions concrete and technical: The current phrasing is still high-level and indirect. The paper should explicitly and precisely state what Kanade’s architectural novelty is, how it differs from prior disentanglement approaches, and what it replaces or simplifies.
> > 2. Improve coherence of the framing: The introduction should be rewritten to lead with a clear articulation of the problem, the tradeoff Kanade addresses, and the exact architectural insight behind the method.
> > 3. Ensure the narrative does not require reader inference: The current presentation forces the reader to infer the novelty. The paper should explicitly state it to prevent misunderstanding.
> > 4. Typo: Fix typo in L117 (“such *[as]* time invariance (TiCodec, Single-Codec, SoCodec), data augmentation (LSCodec), supervision (FACodec), or using pretrained models (FreeCodec)”).

---

### Author Response · Authors · 2025-12-03

We have updated the manuscript based on the feedback we received from the reviewers.

In particular, the main changes we made were:
- Updating the discussion and related work to make clear our contributions.
- Updated the abstract and title to better fit the new narrative
- Reorganized the paper to decrease reliance on supplemental material
- Updated main text to include SLM results from training using the slam recipe. Added additional metrics.
- Expanded OOD reconstruction results, adding baseline results and replacing Expresso with Salmon to determine the effect of large changes in sentiment within an utterance
- Updated TTS for consistent speaker conditioning
- Addition of human evaluation in TTS (quality and prosody) and voice conversion (similarity), with confidence intervals reported in the Appendix.
- Added Seed-TTS-Eval for comparison with other work
- Added ASR evaluation on Switchboard for performance on spontaneous English speech
- Added chunked reconstruction results to show that it is possible to encode arbitrarily long audio using Kanade and that linguistic content is not leaked into the global embedding
- Removed discussion of marginal bitrate and relevant figures because it was divisive.
- Updated all reconstruction and downstream results to use all RVQ layers.

We hope that our responsiveness to reviewer feedback and the robustness of our model in further experiments can provide the confidence to accept our work.

---

### Meta-Review · Area_Chair_yMbZ · 2026-01-06

**Summary:**

There are several issues raised by the reviewers. Even though there are 6 reviewers, the concerns are overwhelmingly consistent. The novelty is one of the main issue, raised by reviewer yxWZ, ihXt, and LiTB. The second problem is the insufficient SLM experiments, raised by reviewer xExe, AYrk, and AUD2. There are also concerns about generalization beyond read speech, raised by reviewer AUD2, ihXt, and LiTB. In addition, there are numerous issues in experimental design, such as unfair comparison and a lack of baselines. There are also various writing issues, raised by reviewer yxWZ and AUD2.

None of the issues can be easily addressed without significantly changing the paper.

**Reviewer Concerns:**

The rebuttal tries to address all the concerns, to the point of re-running many experiments. However, based on the discussion, the reviewers are not convinced by the rebuttal.

**Reviewer Scores:**

The rebuttal does not seem to change the reviewers' opinions, as reviewer yxWZ, ihXt, and LiTB, all explicitly state that they are keeping their scores.

---

### Decision · Program_Chairs · 2026-01-26

Reject